# Calibrating estimates of ionospheric long-term change

Christopher Scott[1,2], Matthew Wild[2], Luke Barnard[1], Bingkun Yu[3], Tatsuhiro Yokoyama[5], Michael Lockwood[1], Cathryn Mitchel[4], John Coxon[6], and Andrew Kavanagh[7]

[1]Department of Meteorology, University of Reading, Berkshire, RG6 6BB, UK
[2]RAL Space, Rutherford Appleton Laboratory, Chilton, Oxfordshire, OX11 0QX, UK
[3]Institute of Deep Space Sciences, Deep Space Exploration Laboratory, Hefei, 230088, China
[4]Department of Electronic Electrical Engineering, University of Bath, BA2 7AY, UK
[5]Research Institute for Sustainable Humanosphere, Kyoto University, Uji, Kyoto 611-0011, Japan
[6]Department of Mathematics, Physics and Electrical Engineering, Northumbria University, UK.
[7]British Antarctic Survey, High Cross, Madingley Road, Cambridge, CB3 0ET, UK

**Correspondence:** Christopher Scott (chris.scott@reading.ac.uk)

**Abstract.** Long-term reduction ($\sim 20km$) in the height of the ionospheric F2 layer, $hmF2$, is predicted to result from increased levels of tropospheric greenhouse gases. Sufficiently long sequences of ionospheric data exist to investigate this long-term change, recorded by a global network of ionosondes. However, direct measurements of ionospheric layer height with these instruments is not possible. As a result, most estimates of $hmF2$ rely on empirical formulae based on parameters routinely scaled from ionograms. Estimates of trends in $hmF2$ using these formulae show no global consensus. We present an analysis in which data from the Japanese ionosonde station at Kokubunji were used to estimate monthly median values of $hmF2$ using an empirical formula. These were then compared with direct measurements of the F2 layer height determined from Incoherent Scatter measurements made at the Shigaraki MU observatory, Japan. Our results reveal that the formula introduces diurnal, seasonal and long-term biases in the estimates of $hmF2$ of $\approx \pm 10\%$ ($\pm 25km$ an altitude of $250km$). These are of similar magnitude to layer height changes anticipated as a result of climate change. The biases in the formula can be explained by changes in thermospheric composition that simultaneously reduce the peak density of the F2 layer and modulate the underlying F1 layer ionisation. The presence of an F1 layer is not accounted for in the empirical formula. We demonstrate, that for Kokobunji, the ratios of $F2/E$ and $F2/F1$ critical frequencies are strongly controlled by changes in geomagnetic activity represented by the $am$ index. Changes in thermospheric composition in response to geomagnetic activity have previously been shown to be highly localised. We conclude that localised changes in thermospheric composition modulate the F2/E and F2/F1 peak ratios, leading to differences in $hmF2$ trends. We further conclude that the influence of thermospheric composition on the underlying ionosphere needs to be accounted for in these empirical formulae if they are to be applied to studies of long-term ionospheric change.

## 1 Introduction

The concept of long-term change in the upper atmosphere and ionosphere due to anthropogenic production of $CO_2$ and $CH_4$ (popularly called the 'greenhouse effect') was first considered by Roble and Dickinson (1989). Using a coupled mesosphere,

thermosphere and ionosphere model, they concluded that the thermosphere would be expected to cool by around 50K as a result of a doubling of the $CO_2$ and $CH_4$ in the lower atmosphere, thereby trapping more heat in the troposphere. Roble and Dickinson (1989) note that the modelled cooling is caused primarily by enhanced $CO_2$ emissions. Rishbeth (1990) examined

the consequences of such a cooling on the ionosphere and concluded that the height of the ionospheric F2 peak, $hmF2$, would be reduced by around 20 km as a result, though changes to the peak density of the F2 layer, $nmF2$, would be small.

Despite the existence of long-term ionospheric data sets, extracting this signal is challenging. In addition to any contraction of the thermosphere due to greenhouse forcing, there are other mechanisms that will change the height of the ionosphere, on a range of timescales.

The behaviour of the upper atmosphere is largely controlled by variations in solar activity, both through changes in ionising solar radiation and through interaction of the solar wind (in the form of fast solar wind streams and coronal mass ejections, CMEs) with the Earth's magnetosphere which influences the ionosphere and thermosphere by driving currents that cause heating. Changes in solar ionising radiation follow the eleven-year activity cycle, with more Extreme Ultra-Violet (EUV) and X-ray radiation incident on the Earth's upper atmosphere at solar maximum, leading to greater plasma production in the

ionosphere (see e.g Rishbeth (1988)). Superimposed on this trend are transient enhancements due to solar flare activity. The solar wind consists of a magnetised plasma flowing supersonically from the Sun and filling interplanetary space. Here, the magnetic field becomes known as the Heliospheric Magnetic Field (HMF). The constant outflow contains fast ($\sim 750kms^{-1}$) and slow ($\sim 400kms^{-1}$) streams, with transient CMEs producing localised regions of modulated solar wind density, speed and magnetic field for periods of a few days. If the HMF and geomagnetic fields are oppositely aligned, the two fields can

'reconnect', (Dungey, 1961) enhancing the flow of energetic particles into the upper atmosphere at high latitudes, and increasing electric fields, both of which cause localised heating (e.g. McCrea et al. (1991)). The resulting expansion and convection in the thermosphere brings molecular-rich neutral gasses to higher altitudes where it enhances the loss rate of ionisation, causing a depletion of the ionosphere (King, 1963). This molecular-rich air is subsequently transported equatorward by general circulation, extending the influence of the geomagnetic activity globally, with the magnitude of the response decreasing with

decreasing geomagnetic latitude (e.g. Rishbeth et al. (1985)). Seasonal changes in thermospheric circulation coupled with seasonal changes in geomagnetic activity produce seasonal variability in thermospheric composition that varies with location. While this is a necessarily brief summary of solar-terrestrial interactions, a detailed review of the subject is given by Pulkkinen (2007) and Schwenn (2006).

Above a given ionospheric monitoring station therefore, changes to the height of the ionosphere are a superposition of several

different effects, occurring across a wide range of temporal scales from long-term ($\sim$ multi-decadal) through solar cycle ($\sim$ decadal) to individual space weather events $\sim$ hours. These can be grouped into several interrelated categories;

1. Geomagnetic activity causes heating of the thermosphere, as described above. This also increases the height of ionospheric layers, which tend to lie at constant pressure levels.

2. Solar irradiance modulates the energy input to the upper atmosphere (thermosphere, ionosphere, mesosphere), increasing

the electron production and changing the height of the ionospheric layers due to the thermal expansion of the upper atmosphere, and hence raising of pressure levels.

3. Changes in thermospheric composition alter the shape of the ionisation profile, with such changes becoming apparent on an ionogram through the visibility of the F1 layer and a weakening of the F2 peak due to molecular species enhancing the loss rate of ionisation at greater altitudes. Changes to the shape of the ionisation profile can alter the altitude at which the peak

electron concentration is established. Such compositional changes can result from thermospheric circulation both on a seasonal timescale and as a result of space-weather related events raising the molecular content of the upper thermosphere which is then transported equatorward via thermospheric circulation.

4. At the higher altitudes of the F2 layer, the relatively long lifetime of individual ions and electrons means that they can be transported through collision with the neutral winds. Therefore changes to the wind pattern influence the height of the F2 layer,

with poleward winds blowing ionisation down the field-lines to lower altitudes and equatorward winds blowing ionisation up the field-lines to higher altitudes.

5. Of a smaller magnitude, there are influences from the lower atmosphere, including contraction of the thermosphere due to the presence of enhanced quantities of greenhouse gasses in the lower atmosphere. This latter effect is predicted to alter the height of ionospheric layers over a multi-decadal timescale.

Absent from most previous literature has been consideration of the ionisation profile changes in (3). This effect may seem irrelevant, as it mainly affects the electron concentration of given features in the profile, rather than their true height. However, such profile changes can have an indirect effect on estimates of this true height using ionosonde measurements. As discussed in the following section, this can potentially introduce bias into reconstructions of true height estimates needed to extract the target climate signal from (5).

Further detailed discussion of the range of potential contributions to ionospheric long-term change has be presented by Rishbeth and Clilverd (1999).

Since long-term sequences of data exist from the global network of ionospheric monitoring stations, covering an epoch of more than 90 years, many studies have attempted to detect the predicted contraction of the ionosphere in response to enhanced greenhouse gasses. One popular technique is to fit proxies for geomagnetic activity and solar irradiance to the data. Variations

in F2 layer height introduced by (1) and (2) are accounted for by subtracting the fit from the data, with any residual long-term drift widely assumed to be dominated by the contraction of the thermosphere (5). When such a technique is applied globally, no consistent pattern has yet emerged from a global analysis of these data (e.g. Bremer et al. (2004)) which has been attributed (e.g. Jarvis et al. (1998), Bremer (1998)) to phenomena that are unaccounted for in the analysis such as localised changes to thermospheric circulation (4).

## 1.1 Measuring the ionosphere using ground-based radar

Free electrons in the Earth's ionosphere resonate at a frequency related to the local electron concentration such that $f = 8.98\sqrt{N}$ where $f$ is the frequency in $Hz$ (known as the plasma frequency) and $N$ is the electron concentration in $m^{-3}$. For the plasma concentrations present in the Earth's ionosphere, this relates to frequencies in the High Frequency (HF) range of the radio spectrum, typically between $1 - 20MHz$. A radio signal (of the so-called 'ordinary wave' with left-handed circular

polarisation, see Rishbeth and Garriott (1969)) launched vertically will be returned from the ionosphere when it reaches an

altitude at which the radio frequency matches the local plasma frequency. By transmitting a series of ordinary wave radio pulses across a range of frequencies and measuring the time for each to be returned from the ionosphere, a vertical profile of the ionospheric electron concentration can be obtained. Data from such a sounding is usually presented as a plot of time-of-flight against radio frequency known as an ionogram. By assuming the pulse is travelling at the speed of light in a vacuum, the time-of-flight can be converted to a height in $km$, though since the presence of plasma delays the pulse, these heights become exaggerated and are known as 'virtual' heights, $h'$. The virtual height of the F2 layer, for example would be expressed as $h'F2$. In contrast, the peak frequency returned from each layer is an absolute value, denoted by $fo$, so, for example, the peaks of the E and F2 layers would be represented as foE and foF2 respectively. The 'o' represents the 'ordinary' ray path since the presence of the Earth's magnetic field makes the ionosphere birefringent, creating an alternative 'extraordinary' ray path for radio pulses propagating with the opposite polarisation. Tabulating the entire ionospheric height profile was not routinely carried out in the early days of routine ionospheric monitoring. Instead, international standards were established for the identification and recording of key features on each ionogram, such as the peak frequencies of each layer and their virtual heights (Piggott and Rawer, 1978). This task is referred to as ionogram 'scaling' or 'reduction' and was usually carried out by a skilled individual or small team from each ionospheric sounding station to ensure consistency of the data.

It is possible to 'invert' an ionogram to obtain the true height profile by integrating along the virtual height profile, accounting for the presence of ionisation at each height step, with assumptions made about the ionisation within the unobserved 'valley' between the E and F regions (e.g. Rishbeth and Garriott (1969)). This process was time-consuming and so was not carried out routinely in the early days of ionospheric research. With the advent of digital sounders and the ability to automatically scale and invert ionograms, true-height analysis is now far more readily available but, alas, does not yet cover sufficiently long time intervals to allow meaningful estimates of trends in ionospheric layer heights. When derived, the true height of each layer is denoted by $hm$ such that the true height of the F2 layer peak is $hmF2$.

The shape of the electron concentration profile is influenced by the composition of the neutral thermosphere, through the loss rate of ionisation. At equilibrium, the electron concentration, $N$, can be expressed as (Rishbeth and Garriott, 1969);

$$N = \left(\frac{q}{2\beta}\right)\left[1 + \left(1 + \frac{4\beta^2}{\alpha q}\right)^{1/2}\right] \tag{1}$$

where $q$ is the ion production rate $(s^{-1})$, $\alpha$ is the loss rate of molecular ions $(s^{-1})$ and $\beta$ is the loss rate of atomic ions $(s^{-1})$. Between the $E$ and $F2$ layers, there is often an additional layer, the $F1$ layer, visible on ionograms. This layer forms between $160 - 200km$ where the loss rate of ionisation transitions from being dominated by the loss of molecular ions at the lower altitudes in the $E$ layer where;

$$N = N_\alpha = \left(\frac{q}{\alpha}\right)^{1/2} \tag{2}$$

to a loss process dominated by atomic ions at the higher altitudes of the $F2$ layer where;

$$N = N_\beta = \frac{q}{\beta} \tag{3}$$

Ratcliffe (1956) first suggested that the transition between molecular to atomic loss processes may account for the splitting of the F region into two distinct layers, with the parameter $\beta^2/\alpha q$ determining the shape of the electron distribution with height. Denoting this quantity as $G$ at the level of peak production;

$$G = \frac{\beta^2}{\alpha q} = \frac{N_\alpha^2}{N_\beta^2} \tag{4}$$

Figure 28 in Rishbeth and Garriott (1969), reproduced as figure 1 of Scott et al. (2021), presents the vertical profile of electron concentration for a range of values of $G$. When $G$ is small $(< 1)$, the presence of the $F1$ layer is barely visible in the profile, becoming a much more pronounced inflexion for larger values of $G$. In this way, the prominence (and thus visibility of the $F1$ layer on an ionogram) is a function of the ratio of molecular and ion loss rates, with the ion composition itself being
controlled by the composition of the neutral thermosphere. F1 layers are a daytime phenomenon, prominent during summer months. While the presence of $F1$ layers in historic ionograms is tabulated (in terms of peak frequency $foF1$ and virtual height, $h'F1$), these do not record the prominence of the layer, from which a more detailed understanding of the ionospheric and thermospheric composition profiles could be gleaned.

Similarly, changes in neutral composition affect the peak electron concentration of the $F2$ layer. At noon, where the $F2$
layer approaches a steady-state condition, production and loss are in equilibrium. If the loss rate of ionisation is enhanced by the presence of molecular ions, the peak electron concentration of the layer will be reduced. Since the electron concentration is proportional to the peak frequency squared, comparison can be made between measurements at similar dates but different times in the solar cycle by scaling these values by the ion production rate, $q$. A good proxy for $q$ is the $F10.7cm$ solar radio flux. Thus;

$$N \propto \frac{f^2}{q} \equiv \frac{foF2^2}{F10.7} \tag{5}$$

Using noon values of $foF2$ to track qualitative changes in thermospheric composition was suggested by Rishbeth et al. (1995) with Wright and Conkright (2001) comparing the efficacy of this simple index (their $FFD$ index, averaged over 5 hours around noon) with other more complex indices derived from the rate of change of the ionosphere at sunrise.

### 1.2   Investigating long-term trends in the height of the upper atmosphere

Temperature trends in the troposphere and stratosphere have been defined by re-analysis of data, for example from the GNSS RO, ERA5, MERRA2, and ERA-I satellites Shangguan et al. (2019). These authors show that tropospheric warming is accompanied by stratospheric cooling - an unique signature of the greenhouse gas effect. However, this can only be observed for the interval of regular global stratospheric temperature data. Shangguan et al. (2019) studied the interval 2002-2017 and this

was recently extended to 1986-2022 by Santer et al. (2023). The trend is also detected in balloon radiosonde data extending back to 1978 (Philipona et al., 2018) but these data also show that the trend slowed, stopped or even reversed, depending on location, around 2000 because of the recovery of the ozone layer. In comparing these studies it is important to remember that these effects in the upper atmosphere are altitude-dependent, which will lead to unhelpfully differing results, depending on the altitudes of the specific observations in each study. By contrast, searching for a descent in ionospheric layers is valuable because it would be the integrated effect of upper atmosphere cooling over all altitudes. Furthermore, of the upper atmosphere regions, the ionosphere is unique as it can be observed remotely with relative ease. Due to the ionosphere's importance for long-distance radio communication, such observations have been made routinely since the early 20th century. The resulting longevity of ionospheric data series offers the potential to extend observations of stratospheric cooling back by a further five decades, provided the cooling effect trend can be extracted from the data, with all other potentially-confounding effects compensated for.

This potential has motivated similar re-analysis of ionospheric data, seeking evidence of climate-driven trends. The first published analysis (Bremer, 1992) of long-term trends in $hmF2$ was for the mid-latitude station at Juliusruh ($54.6°N, 13.4°E$) and provided evidence for a decrease of the peak height of the ionospheric F2-layer. The associated long-term variations of the peak electron concentration were small, consistent with the modelling work by Rishbeth (1990). Subsequent work (Bremer, 1998) repeated the analysis for 31 stations in the European sector for which long-term ionospheric records exist. Bremer concluded that in the F2 region, there was no consistent trend, with stations west of 30° E showing negative trends in $hmF2$ and peak electron concentration (inferred from $foF2$) whereas positive trends in both parameters dominated in data from stations to the east of 30° E. Bremer further remarked that these longitudinal differences probably resulted from dynamical effects in the F2 layer.

Jarvis et al. (1998) presented an analysis of long-term trends in $hmF2$ observed in two Southern-Hemisphere stations. They reported long-term changes in altitude, which showed seasonal and diurnal variation, at both sites. The magnitude of the long-term trend was altitude-dependent which, they argued, could be interpreted either as a constant decrease in altitude combined with a decreasing thermospheric wind effect or as a constant decrease in altitude which is altitude-dependent.

Of particular relevance to the current paper is the work of Xu et al. (2004) who conducted an analysis of data from the ionosonde station at Kokubunji in Japan (the same station examined here) with monthly medians of ionosonde observations taken over a period of more than four solar cycles. Using a linear regression model to eliminate solar and geomagnetic effects, they determined a decreasing trend in $hmF2$ of $0.398km/year$ at noon and $0.505km/year$ at midnight. In addition, they analysed seasonal and diurnal trend variations. They found that the seasonal variations of $hmF2$ at noon and midnight were opposite to each other, though the long-term trends at both times remained negative. The data indicated that the effect of geomagnetic activity was not significant in regression models applied to data recorded at this station.

Bremer et al. (2004) presented an analysis of global trends in a number of ionospheric parameters, including $hmF2$. They concluded that, in the F2 layer, the scatter of trends for the different stations was high and no significant mean global trends could be estimated.

There have subsequently been many studies made of $hmF2$ trends, the summarisation of which lies beyond the scope of this paper. A useful review of more recent investigations into long-term ionospheric trends has been presented by Lastovicka (2013).

While details of the analysis technique differ between studies, long-term trends in $hmF2$ are usually determined in the following way. 1. An empirical formula based on standard ionosonde scaled parameters (usually monthly medians of $foF2$, $foE$ and $M(3000)F2$ - see section 2 for a definition of the latter quantity) is used to estimate $hmF2$ over an extended time period (preferably several decades). 2. Having determined the long-term trend in $hmF2$, the influence of variability in geomagnetic activity and solar irradiance is estimated by fitting proxies for these (usually the $Ap$ index and solar $f10.7cm$ flux respectively) to the $hmF2$ data. 3. This two-parameter fit is then subtracted from the original data and any difference is considered to be due to local environmental change, such as via greenhouse forcing.

When analysing such trends in residuals however, it is important not to assume any local environment changes which may underlie this can be attributed to greenhouse forcing alone. The previously-discussed effects altering thermospheric composition (3) and wind patterns (4) risk being confounding factors, as they can potentially also lead to trends on long timescales. Mikhailov and Marin (2001) argued that the observed F2 trends were strongly dependent on the long-term variations in geomagnetic activity through changes in composition of the neutral thermosphere, thermospheric temperature and changes to the neutral wind. They subdivided the time-series to demonstrate that the observed trend in $F2$ parameters was dependent on the rate of change in the geomagnetic activity. Subsequent work (Mikhailov, 2006) proposed that the difference in $hmF2$ trends seen across Europe could be explained by differences in thermospheric winds.

Scott et al. (2014) presented long-term changes in the relative strength of the annual and semiannual variability in the $foF2$ critical frequencies at Slough/Chilton in the UK which were highly anticorrelated with those recorded at Stanley in the Falkland Islands. The dominance of annual or semi-annual variations in $foF2$ is a function of thermospheric composition and so they argued that the observed long-term changes are due to changes in thermospheric composition driven by geomagnetic activity. Since the response was so different at the two stations, Scott et al. (2014) also suggested that this could account for the differences in long-term trends in $hmF2$ observed at different locations.

Subsequent analysis (Scott and Stamper, 2015) was conducted to investigate the long-term trends in annual/semiannual variability in $foF2$ from 77 ionospheric monitoring stations around the world. By using Slough as a reference station, and correlating the long-term trends from other stations with it, strong regional variations were revealed in the data, which bore a striking similarity to the regional variation observed in long-term changes to the height of the ionospheric F2 layer presented by Bremer et al. (2004). Scott and Stamper (2015) argued that since both the height and peak electron concentration of the ionospheric $F2$ region are influenced by changes to thermospheric circulation and composition, the observed long-term and regional variability can be explained by such changes.

Rishbeth (1999) considered the results in $hmF2$ long-term trends presented up to that date and discussed the challenges in extracting a reliable signal of the long-term ionospheric change induced by greenhouse warming. He argued that long-term sequences of ionosonde data are needed to address the question but that any data analysis must be "accurate and painstaking", with thought given to the subsequent analysis and interpretation of the data. Ulich et al. (2003) went further in considering

some of the problems with identifying long-term trends in ionospheric data. They highlight the lack of consistency between results from different locations, the quality control of the data, the significance of any resulting trends, the reliance on empirical formulae for calculating $hmF2$ (including how they account for the presence of underlying ionisation) and the presence of other dominant signals in the data that lead to diurnal, seasonal and solar cycle variations.

The purpose of the current paper is to investigate the potential pitfalls in deriving long-term ionospheric trends in $hmF2$ using empirical formulae and to demonstrate that such effects may have potential for reconciling the differences in trends derived from the global network of ionospheric monitoring stations. Section 2, contains a summary of various methods used to derive $hmF2$ estimates from routinely scaled ionospheric parameters, and the assumptions made in doing so. Thereafter, section 3 will examine the accuracy of such estimates through comparison with ionospheric heights measured by Incoherent Scatter Radar.

## 2 Estimating $hmF2$ from empirical formulae

While in more recent decades automatic scaling and inversion of ionograms have produced routine estimates of $hmF2$, for historical data, this was not always the case. Even for those stations where the original analogue ionograms survive, retrospectively scaling and inverting these data would be prohibitively labour-intensive and time-consuming. Early on in ionospheric science, thought was given as to how to estimate $hmF2$ values from existing standard ionospheric parameters (e.g. $foF2$, $foE$). Determining these from an ionogram only required a scaling process (not inversion), hence these were routinely calculated from ionograms at the time of measurement.

A first simple approach to the problem (Booker and Seaton, 1940; Appleton and Beynon, 1940) was to assume that the F2 layer electron concentration was parabolic with height (a so-called "parabolic model") and that collisions and the effects of the Earth's magnetic field were ignored. From this, a relation between the true height, $h$, and the virtual height, $h'$, could then be derived.

Appleton and Beynon (1940) considered the relationship between the critical frequency for vertical incidence, $foF2$, and the maximum usable frequency, $MUF$, that can be reflected from the layer over a given distance. This relationship depends on the height of the layer, the thickness of the layer and, to a lesser extent, the presence of underlying ionisation. By international agreement, for standard communications purposes, the $MUF$ is considered over a distance of $3000km$. The ratio $MUF/foF2$ is referred to as the $M(3000)F2$ factor and is calculated according to a semi-empirical relation (e.g. Lockwood (1983)).

For a thin layer and a curved Earth, Appleton and Beynon (1940) derived the relationship;

$$hmF2 = \frac{1500}{((M(3000)F2^2 - 1)^{1/2})} - 176 \tag{6}$$

For a thick layer and curved Earth, Appleton and Beynon (1940) derived the following relation to estimate the true height of the layer peak, $hp$;

$$hp = \frac{1153}{((M^2-1)^{1/2})} - 100 \qquad (7)$$

where $M$ represents the $M(3000)$ factor of an undefined layer.

Shimazaki (1955) made simplifications to the theory, since Snell's law is invalid for a thick layer, Bouguer's rule (that the path of the ray is irrelevant) must be employed together with Martyn's equivalence theorem, applied to a curved Earth. The following relation was derived as a result;

$$hmF2 = \frac{1490}{M(3000)F2} - 176 \qquad (8)$$

Dudeney (1974) showed that these assumptions should lead to an overestimate of $hmF2$ of between $10$ and $15km$. However, Shimazaki (1955) used data from a selection of stations representing a wide range of global locations and compared values obtained from his formula with those of Booker and Seaton (1940) and found no systematic offset. Dudeney (1974) suggested that this could be due to offsetting assumptions involving a simple parabolic layer, the lack of a magnetic field when deriving $M(3000)F2$ (thus introducing a dependence on dip angle) and differences in the methods used of deriving $M(3000)F2$ from actual ionograms. Dudeney (1974) concluded that the Shimazaki (1955) formulation is fundamentally inaccurate but that similar inaccuracies in the accepted method of determining $M(3000)F2$ tend to compensate. Meanwhile, though the Appleton and Beynon (1940) formula was inherently more exact (with the $1/(M^2-1)$ formulation being more accurate), it was of no practical use generally due to globally varying factors (such as the magnetic dip angle).

In their publication, Booker and Seaton (1940) recognised the need to correct for underlying ionisation in the $E$ and $F1$ layers. Vickers (1959) proposed a method that accounted for the $F1$ layer ionisation but it was limited in that it could only be used when a scalable $foF1$ parameter was visible on the ionogram (most often during the day in summer months) and the coefficients were strongly dependent on sunspot number, leading to further complex analysis.

Bradley and Dudeney (1973) found the simplest way to account for underlying ionisation was to use parabolic models for the $E$ and $F2$ layers and represent the interim ionisation with a linear increase in electron concentration. The height of the $E$ layer peak was fixed at $120km$, with a thickness of $20km$. Trial and error showed that the best agreement occurred (with values of $hmF2$ derived from ionogram inversion analysis) when the linear portion of the assumed underlying ionisation profile intersected the $F2$ parabola where it equalled 2.89 times the peak E-layer electron concentration (equivalent to $1.7foE$). They noted that the majority of $foF1$ values on ionograms were scaled from minor fluctuations in electron concentration (indicating that the presence of an F1 layer did not represent a significant increase in electron concentration). They argued that it is the ionisation between the $E$ and $F2$ regions that contribute most to the group retardation of signals returned from the $F2$ region, irrespective of the prominence of the $F1$ ledge. In this way, they suggested $F1$ ionisation could be accounted for by using the the more ubiquitously-recorded parameters $foE$ and $foF2$.

Using synthetic ionograms that neglected the influence of the Earth's magnetic field, they found that their results were consistent with the following for $x_E > 1.7$ (where $x_E = foF2/foE$)

$$hmF2 = a(M(3000)F2)^b \tag{9}$$

where $a = 1890 - 355/(x_E - 1.4)$ and $b = (2.5x_E - 3)^{-2.35} - 1.6$

Dudeney (1974) noted that $x_E > 1.7$ is equivalent to about $x_F \approx 1.2$ (where $x_F = foF2/foF1$), sufficiently close to the layer critical frequency to make a significant contribution to the total group delay of the radio pulse. However, Bradley and Dudeney (1973) compared estimates of $hmF2$ with heights determined from ionogram inversion for a number of different locations, over all seasons and for solar cycle extremes. These results supported their conclusion that the presence of an $F1$ ledge had minimal effect on $hmF2$ estimates calculated using their empirical formula.

While powerful, this method cannot be used when $X_E < 1.7$, which frequently occurs during the daytime summer at mid to high latitudes (Dudeney, 1974). Here the E-layer is strongest due to increased ion production while the F2-layer is often weakened through an enhanced loss rate resulting from a higher fraction of molecular species in the upper thermosphere. In addition, this formula does not account for the effects of the Earth's magnetic field.

Dudeney (1974) concluded that the best way to estimate $hmF2$ is via ionogram inversion, though this is a slow and expensive process. The Bradley-Dudeney model (Bradley and Dudeney, 1973) is generally applicable over wide areas, though a model selected and calibrated to fit the ionosphere at a particular location is capable of higher accuracy (such as demonstrated by Vickers (1959)).

Dudeney (1974) considered a method that followed Shimizaki's original equation (Shimazaki, 1955) but applied a correction for the underlying ionisation using a value that assumes the contribution of $foF1$ is negligible. To do this he differentiated Shimazaki's equation to obtain the relation;

$$\Delta h = \frac{(1490 \Delta M)}{M_o^2} \tag{10}$$

where $M_o = MUF/foF2$.

In this way, correcting for the underlying ionisation by considering the difference between modelled and observed heights as a function of $\Delta M$, means $\Delta M$ can be considered a function of $x_E$, whereas $\Delta h$ is an inverse function of $M$ and hence a direct function of $hmF2$. In this way, Dudeney (1974) established an empirical function of $x_E$, creating a single equation for $hmF2$ applicable to all epochs of the solar cycle. (but still not accounting for local variations in the Earth's magnetic field).

From his analysis, Dudeney (1974) derived two formulae;

$$hmF2 = \frac{1490}{(M(3000)F2 + \Delta M)} - 176 \tag{11}$$

where $\Delta M = \frac{(0.280 \pm 0.009)}{(x_E - 1.200)} - (0.028 \pm 0.010)$

or, more comprehensively;

$$hmF2 = \frac{(1490.MF)}{(M(3000)F2 + \Delta M)} - 176 \tag{12}$$

where $MF = M(3000)F2 \left( \frac{(0.0196M(3000)F2^2 + 1)}{(1.2967M(3000)F2^2 - 1)} \right)^{1/2}$ and $\Delta M = \frac{(0.253 \pm 0.008)}{(x_E - 1.215)} - (0.012 \pm 0.009)$

Dudeney (1974) state that the differences between these two equations are barely significant for most values of $M(3000)F2$

but become more important when $M(3000)F2$ is small. Therefore for studies including solar cycle variations in $hmF2$, where extreme values of $M(3000)F2$ are expected, the more complex relationships must be used.

For these relations, Dudeney (1974) estimates the uncertainty in $hmF2$ to be $\pm 89/M(3000)F2^2 km$.

Dudeney (1974) concludes that calibration of this equation should be carried out for each individual station as it is probable that the $\Delta M$ relation is a function of the Earth magnetic dip angle and and plasma gyro frequency. Through a comparison with

the Bradley and Dudeney (1973) equation, Dudeney (1974) states that it should be possible to use the same coefficients with confidence over quite wide zones.

Subsequently, further refinements of similar formulae have been carried out, though a comprehensive review will not be given here. One popular formulation is that of Bilitza et al. (1979) which attempts to take account of geographic sensitivity to geomagnetic activity by using sunspot number as a proxy.

Bilitza et al. (1979) compare a wide variety of empirical $hmF2$ formulae with incoherent scatter radar data (over periods of around 4 years from the 1960s and early 1970s) for Millstone Hill, Arecibo and Jicamarca. They conclude that the global ionosphere is best represented using either the Bradley and Dudeney (1973) model, or that of Bilitza and Eyfrig (1978).

McNamara (2008) used the international reference ionosphere (IRI) to generate ionospheric profiles against which the efficacy of the Dudeney (1974) and Bilitza et al. (1979) empirical $hmF2$ formulae were tested. From these profiles they generated

artificial ionograms for different times, seasons and points in the solar cycle. By scaling the necessary parameters from these (including $M(3000)F2$) they were able to use them to estimate $hmF2$ using a variety of formulae. They concluded that the best agreement was found when considering the midnight $F2$ layer using the simple approximation that $hmF2$ was found at a virtual height where the plasma frequency was $0.834 \times foF2$, since at midnight, the layer approximates best to the assumption of a parabolic $F2$ layer. However, this is not easily applied to the study of long-term change in the ionosphere since this

parameter ($hpF2$) was not routinely recorded and would require scaling from the original ionograms.

In the absence of $hpF2$ values, McNamara (2008) concluded that the Dudeney (1974) model is better than the Bilitza et al. (1979) model for midnight ionograms. The scatter in the model errors is smallest at midnight, and is smaller for the Dudeney (1974) model (because the errors have a smaller solar-cycle variation). During the day, the Bilitza et al. (1979) formula gave the smaller range of errors because of the inclusion of a solar cycle term.

McNamara (2008) was also able to investigate the uncertainty in the values of $M(3000)F2$, which should be expected to be at least as large as the standard scaling accuracy of $\pm 0.05$, with a superimposed random component. An uncertainty in $M(3000)F2$ of $\pm 0.1$ would lead to an uncertainty in $hmF2$ of $\pm 15km$, though if these uncertainties were indeed random, this could be accounted for by considering monthly median values. McNamara (2008) cautions that the conclusions presented

in their work are predicated on the assumption that the version of the IRI used was a better representation of the subpeak
ionosphere than the empirical models of $hmF2$.

While analysis of long-term change in $hmF2$ has been presented for many stations, no standard formula has been used to calculate $hmF2$. For the purposes of our analysis, which aims to investigate the presence of any long-term bias in empirical estimates of $hmF2$ through comparison with heights determined by an extended sequence of data from Incoherent Scatter Radar, we will compare with the relation of Bradley and Dudeney (1973), presented in equation 4. Many authors have used this formulation, in particular Jarvis et al. (1998). It is useful for us to use this as a starting point for such comparisons with an ISR since we wish to reproduce the analysis of Jarvis et al. (1998) here, in order to determine which elements can be interpreted as physical change within the ionosphere, and which are biases introduced by the assumptions used in formulating the empirical relationship between ionospheric parameters and $hmF2$. In keeping with Jarvis et al. (1998) we will estimate values of $hmF2$ where $foE$ is below the detection threshold of the ionosonde at night by assuming the low value of $foE = 0.4MHz$.

## 2.1 The Kokobunji ionosonde data

Routine observations of the ionosphere have been made using an ionosonde at Kokobunji, Japan (35.71 N, 139.49 E) since the International Geophysical Year in 1957. Scaled parameters from these hourly ionospheric soundings have been digitised and made available via the UK Solar System Data Centre (www.ukssdc.ac.uk). To estimate $hmF2$, scaled critical frequency parameters for the $E$, $F1$, and $F2$ layers ($foE$, $foF1$, and $foF2$), as well as the $M(3000)F2$ factor were downloaded. Monthly averages were then calculated for these data, to protect against outliers caused by short-lived space weather events that are not representative of the data on monthly timescales. Specifically, hourly monthly medians were used - medians calculated across corresponding hours (in local time) within a given month. For a given year, this yields 288 median values (bins of 12 months, and 24 local time hours). Such hourly monthly medians of $foE$, $foF2$ and $M(3000)F2$ were then used to estimate corresponding hourly values of $hmF2$, the true height of the $F2$ layer, using the formula of Bradley and Dudeney (1973) as given in equation 9. Following the analysis of Jarvis et al. (1998) it was initially assumed that $foE = 0.4MHz$ at night. Where $x_E < 1.7$, no value of $hmF2$ was calculated.

## 2.2 The Middle and Upper Atmosphere (MU) Radar

The Middle and Upper Atmospheric (MU) radar is located at Shigaraki MU observatory, Shigaraki, Japan. Being located at latitude $34.85°$ N and longitude of $136.12°$ E, this is at a similar latitude to Kokobunji and about 310km to the west. For 2004 (the centre of the interval over which data from the two stations are compared) the International Geomagnetic Reference Field (IGRF) gives the geomagnetic coordinates of the Kokobunji ionosonde to be $26.78°$ N and $208.22°$ E and of the MU radar to be $25.65°$ N and $205.24°$ E. Designed for both middle and upper atmospheric studies, it has been routinely making observations of the ionosphere using incoherent scatter since 1986. True incoherent scatter occurs when an electromagnetic wave excites electrons within a plasma. Each electron acts as an antenna that re-radiates the wave, with the thermal and bulk motion of the plasma Doppler shifting the original signal. The received signal is a superposition of the re-radiated waves from all the electrons in the line-of-sight of the incoming wave in the range "gate" set by the pulse delay range that the received signals

are integrated over. In the ionosphere, while the heavier positive ions within the plasma are not excited directly by the radio wave, they influence the motion of the electrons, thereby modifying the received signal spectrum. If the transmitted frequency corresponds to wavelengths significantly greater than the Debye length of the plasma, the scatter is not truly incoherent, but rather occurs preferentially from ion-acoustic waves within the plasma, resulting in a characteristic 'double-humped' spectrum corresponding to the upwards and downwards propagating ion-acoustic waves of the appropriate wavelength. In this way, incoherent scatter enables routine measurement of electron concentration, the bulk motion of the plasma and the ion and electron temperatures.

The MU radar transmits in the VHF radio spectrum at a frequency of $46.5 MHz$ ($3.5 MHz$ bandwidth and $1 MW$ peak output power). The antenna field consists of 475 antennas arranged in a circular array with a diameter of $103\ metres$. Fast beam steering enables various observation configurations. Ionospheric observations are routinely made with the radar in Incoherent Scatter Radar (ISR) mode. These consist of a sequence of four beam directions, with the azimuth and zenith angles of the beams of $(355.0, 20.0)$, $(85.0, 20.0)$, $(175.0, 20.0)$, and $(265.0, 20.0)$, degrees respectively. When operating in ISR mode, the radar can make measurements of electron and ion temperatures, plasma velocity and echo power density. The echo power data show the intensity of electromagnetic waves reflected from the ionosphere between $80$ and $1,200 km$. The heights recorded by the ISR are not subject to the same delays as with the ionosonde data since the transmitted frequencies are far greater than ionospheric plasma frequencies. However, at $46.5 MHz$ there will still be small delays that will result in the systematic increase in measurements of F2 layer height of the same order as the height resolution of the radar ($\approx 4.5 km$). For clarity, such delays are not considered in the main analysis of the current paper, since their inclusion does not significantly affect the results. Modified results that include an estimate of the expected delay are discussed in the conclusions.

The ISR mode is run on a campaign basis, with a typical run lasting from several hours to over a day. The data are made available as hourly averages of height versus received power (in decibels) for the four antenna positions. For the purposes of our analysis, data from the four beams were first converted from decibels to a received power of arbitrary units following the method detailed by Sato et al. (1989). These height profiles were then averaged over the four antenna positions to reduce any random errors. The system noise for each combined height profile was then estimated from the average power returned from heights above $700 km$ (which are considered to contain no signal). This noise was then subtracted from the received powers which were then range corrected. The resulting power profiles can be converted to absolute electron concentration through calibration with a measure of absolute electron concentration (such as from an ionosonde) but this was not necessary for the present analysis since it was only the height of the ionospheric $F2$ layer that was of interest, not the electron concentration. The $F2$ peak in each profile was then identified as the largest range-corrected power in each profile occurring between altitudes of $180$ and $500 km$. This window was selected to be as wide as possible without potential contamination from strong sporadic E layers. In order to suppress estimates from noisy profiles, data points with a signal-to-noise-ratio (SNR) below $5\%$ were excluded from the analysis.

For comparison with the $hmF2$ values estimated from the ionosonde data, corresponding hourly monthly means were calculated for each hour of ISR data. The radar is not run in ISR mode as routinely as the ionosonde generates ionograms, but over the 35 years of ISR data used in this study, ISR observations have been made in $48\%$ of the $10,080$ bins (monthly means,

in bins for each local time hour and month, over 35 years). While monthly median values are calculated for the ionosonde parameters, for the ISR data, measuring $hmF2$ directly, the number of data points per bin is small and so the median is inappropriate. That having been said, the mean and median values were significantly different in only 17 out of the 4853 bins
containing data. When the ISR data are averaged annually, there is no significant difference between the arithmetic mean and median values.

## 3 Results

### 3.1 Seasonal and diurnal comparison

Hourly monthly median $hmF2$ values derived from the Kokubunji ionosonde data using the model of Bradley and Dudeney
(1973) are presented in figure 1 (top panel). These are compared with hourly mean $hmF2$ values derived from the MU radar (middle panel). Both data sequences show a clear solar cycle trend, as well as a decreasing trend visible in both sequences from the start of routine MU radar observations in 1986. The difference between these two datasets, $\delta hmF2$ (lower panel) is calculated by subtracting the ionosonde-derived $hmF2$ values from the ISR measurements of $hmF2$. From this comparison it can be seen that the ISR data are noisier, due to there being relatively fewer data points from which the mean values are
calculated. Also at solar minimum years some values are close to the $180km$ floor of the altitude window in which the F2 peaks were identified.

The $hmF2$ values from the two datasets are compared in figure 2 for the 35 years when the two datasets overlap (1986-2020). The top left panel shows all the data overplotted, with daytime data (for which the solar zenith angle, $sza < 90°$ ) shown as black points, twilight data ($90° \geq sza \leq 100°$) shown in magenta and nighttime data ($sza > 100°$) shown as cyan points.
For clarity, these populations are also plotted separately. As expected, there is a strong similarity between the two datasets. Conducting a robust linear fit to all the data (to minimise the influence of outliers) results in a best-fit line with a gradient of $0.71 \pm 0.01$ and an offset of $86.15 \pm 1.82km$. Restricting the fit to consider just the daytime points, the relationship improves considerably, with a gradient of $0.86 \pm 0.1$ and offset of $37.63 \pm 2.01$. There is much more scatter in the twilight and night-time $hmF2$ populations, with fit gradients of $0.56 \pm 0.02$ and $0.41 \pm 0.01$ respectively.

From equation 9, over-estimating $foE$ will lead to an underestimate of the $foF2/foE$ ratio and an underestimate of $hmF2$. In addition, assuming that the value of $foE$ is a constant (nighttime E-region ionisation results from cosmic ray and astronomical x-ray sources that will vary throughout the night) likely introduces scatter around this underestimate. Added to this is the fact that a typical ionosonde is insensitive to frequencies below $\sim 1MHz$.

With reference to figure 2, that the night-time estimates result in a lower gradient than the twilight population indicates that
the assumed value of $0.4MHz$ is an over estimate at night, being more applicable to (though still an over-estimate of) the E-region critical frequency at twilight (at least for this location).

In order to investigate whether this difference was due to seasonal or diurnal biases between the two data sets, monthly averages were calculated for each hour, averaging the aforementioned 10,080 bins over the year axis, for the 35 years for which there was common data. The results are presented in figure 3. While the broad distributions are largely similar in both

data sets (higher $hmF2$ at night), there are differences. The ISR (top right panel) shows more distinct peaks in night-time $hmF2$ at the equinoxes (months 3 and 9) while the ionosonde-derived $hmF2$ values (top left panel) show an unexpected stronger peak around mid-day in the summer months. The difference between these two datasets (lower left panel) confirms that the ionosonde estimates of $hmF2$ exceed those measured by the ISR in summer at noon (and midnight). The number of years of ISR data contributing to the mean value in each bin is plotted in the lower right hand panel. This confirms that there is an adequate number of data points in each bin with which to calculate these means (minimum 14, maximum 31, median 25). The dotted and solid white lines overlaid on these plots represents the times at which the solar zenith angle is $90°$ and $100°$ respectively. These values were used to differentiate between the 'daytime', 'twilight' and 'nighttime' populations. It is interesting to note that there is a clear change in $hmF2$ at the dawn boundary which is less apparent at dusk. This is likely due to variability in the F2 layer at dusk which is 'reset' by the decay of this layer overnight.

The maximum bias in the empirical $hmF2$ formula occurs around noon in the summer months, coincident with the presence of foF1 layers in ionogram data. We therefore next investigated whether the assumptions made about the underlying ionisation in the empirical calculation of $hmF2$ could be the source of this midday summer bias. To do this, we plotted the difference between these two data sets against the ratio of $foF2/foF1$ values (the parameter $xF$ intruduced when discussing equation 9). The results are presented in figure 4. It was expected that equation 9 should be valid for values of $foF2/foF1$ above 1.2. While there is no significant difference between the two distributions for values of the $foF2/foF1$ ratio above 1.6, below this, it appears that the presence of an $foF1$ layer significantly affects the ionosonde-derived $hmF2$ values due to the presence of underlying ionisation that is unaccounted for in the empirical formula. A line was fitted to the values with a ratio below 1.6 (gradient $231 \pm 21$, offset $-356 \pm 31km$) and this relationship was used to correct for the presence of $foF1$ in the ionosonde-derived estimates of $hmF2$. For each bin in figure 3 where $foF2/fof1 \leq 1.6$, the liner fit was used to calculate the bias in $hmF2$ and these biases were used to correct the ionosonde-derived values of $hmF2$. The revised seasonal and diurnal distribution is presented in figure 5. Comparing with the distribution of ISR-derived $hmF2$ values presented in the lower panel of figure 3, it can be seen that corrected ionosonde-derived $hmF2$ daytime summer values (where $foF1$ is most likely to be observed) are now in much closer agreement with the ISR values. While the presence of $foF1$ values during the day are an indication of changes in thermospheric composition, this has not corrected for the difference in nighttime $hmF2$ values since $foF1$ is only visible during daylight hours. Nevertheless, changes to the thermospheric composition will still be present at night, affecting the loss-rate of ionisation which could potentially introduce a bias into the derivation of $hmF2$ values through changes to the distribution of underlying ionisation. Additionally, it has been shown that there is a greater uncertainty in the empirical equation via the approximation of a fixed $foE$ value of $0.4MHz$ at night.

While the presence of the F1 layer is not accounted for in the empirical formula, changes in thermospheric composition could also cause bias through modulation of the loss rate in the F region during times of enhanced molecular composition. This would in turn influence the $foF2/foE$ ratio (the 'x' term in the empirical relation). Both $foE$ and $foF1$ are Chapman layers that are only present during the daytime. As a result they are naturally highly correlated.

Figure 6 presents the seasonal and diurnal variation in both $foF2/foE$ and $foF2/foF1$ ratios. Both ratios show distinct minima around noon in the summer months. While $foF1$ and $foE$ peak around these times, the ratio of both quantities is

reduced due the the reduction of $foF2$ during the summer months (average seasonal/diurnal plots of the individual parameters are presented in Appendix A).

In order to test for the presence of bias in the $foF2/foE$ ratio, we next repeat the above analysis, this time binning $\delta hmf2$, by the corresponding value of the $foF2/foE$ ratio.

Figure 7 reproduces figure 4 with $\delta foF2$ this time being plotted against $foF2/foE$. It can be seen that there is a similar
bias which affects $hmF2$ values where the $foF2/foE$ ratio falls below $\sim 2.5$. It should be noted that if night-time data are included using the approximation of $foE = 0.4MHz$, that this results in a population with much larger values of $foF2/foE$ which have a broad range of $hmF2$ estimates. This is further evidence that such an approximation is not applicable for such long-term studies. Correcting for the bias introduced by $foF2/foE < 2.5$ values and applying this correction to the seasonal data (figure 8) once again results in a reduction of the summertime noon bias.

Whether or not the biases in the $foF2/foE$ and $foF2/foF1$ ratios are independent of each other, both are most dominant during the summer months around noon where the reduction in $foF2$ and the presence of $foF1$ are both characteristic signatures of compositional change in the thermosphere, with a larger fraction of molecular species enhancing the loss rate of ionisation in the upper thermosphere.

## 3.2  Long-term bias

The above analysis has established that there are biases in the empirical $hmf2$ formula for values of $foF2/foE$ and $foF2/foF1$ below thresholds of 2.5 and 1.6 respectively. It has been shown that these introduce systematic errors on a seasonal and diurnal basis. It is therefore pertinent to the discussion of long-term change in the ionosphere to now consider how such biases may influence the long-term trends in $hmF2$ values derived via an empirical formula. Figure 9 presents the $foF2/foF1$ and $foF2/foE$ ratios against year for hourly monthly median values (top panels), annual average (middle panels) and the per-
centage of observations (where a ratio can be calculated), for which the ratio is less than the threshold below which a bias is introduced into the empirical $hmF2$ equation ($\leq 1.6$ for $foF2/foF1$, as identified in Figure 4 and $\leq 2.5$ for $foF2/foE$, as identified in figure 7). It can be seen that there is a strong solar cycle dependence in both of these ratios, together with longer-term changes, particularly the apparent step change since the year 2000. The result of this is that some years will be far more susceptible to the systematic errors introduced into the empirical formula. The lower panels demonstrate that the percentage of
data (for which ratios can be calculated) falling below these thresholds can vary from around 10 to 100%.

The relationship between the bias in $hmF2$ and the $foF2/foF1$ ratio established above was used to correct affected $hmF2$ values (where $foF2/foF1 \leq 1.6$) before averaging by year. When daytime values are plotted against the ISR $hmF2$ values (figure 10), the correction results in a revised gradient of $0.89 \pm 0.01$ with an offset of $30.32 \pm 2.28km$.

Applying the $foF2/foE$ correction to data results in an even closer fit between these datasets (figure 11). The gradient of
the fit is $0.97 \pm 0.01$ with an offset of $0.68 \pm 2.67$. This improvement over the correction due to the presence of an $F1$ layer is likely due to the fact that $foF2/foE$ values exist for a larger proportion of the daytime data points ($\sim 64\%$) than $foF2/foF1$ ($\sim 42\%$).

While both these corrections have improved the relationship between the empirically derived and directly measured $hmF2$ values, the remaining gradient is not 1:1. This is unsurprising since there are other approximations that have been made when determining the coefficients within the empirical relation (which are likely specific to the dip angle of the local magnetic field) and in deriving $M(3000)F2$ values from the ionograms (which does not account for the presence of the magnetic field). In addition, we have not corrected for the small but systematic bias in $hmF2$ introduced by the signal delay in the ISR data.

It can be seen from figure 5 that the night-time ionosonde-derived $hmF2$ values still show more scatter when compared with those measured by the ISR. As shown in figure 2, using a value of $foE = 0.4MHz$ in the empirical formula tends introduces more uncertainty into the $hmF2$ estimates which results in an underestimate of the F2 layer height on average.

## 3.3   Influence of the Earth's magnetic field on long-term $hmF2$ estimates

Standard calculations of $M(3000)F2$ do not take account of the influence of the Earth's magnetic field on radio propagation and it has long been known that this can introduce a bias into the estimation of this parameter from ionograms (Davies, 1959). More recently, Elias et al. (2017) have modelled this bias and quantified the subsequent error introduced into estimates of $hmf2$. In order to estimate the influence of the magnetic field on $hmF2$ calculations for the location used in this study, the international geomagnetic reference field (IGRF, Thébault et al. (2015)) was used to determine long-term magnetic field variations at an altitude of $250km$ above Kokubunji. Throughout the epoch of this study. the inclination has remained relatively stable, declining from $48.7°$ to $48.4°$ between 1957 and 1980, subsequently rising to $49.6°$ by 2020. Using the figure presented in Elias et al. (2017), this would result in a systematic offset in $hmF2$ of $\leq 1km$, well within the uncertainties of the measurements. Further to this, modelling work by Cnossen and Richmond (2008) and Elias (2009) indicates that changes to the Earth's magnetic field over Kokobunji would not be expected to affect the observed values of $hmF2$ through thermospheric dynamics for the epoch covered by this study. It is therefore assumed that there is no measurable bias caused by magnetic field changes in the long-term variation in $hmF2$.

## 3.4   The impact of $foF2/foF1$ and $foF2/foE$ biases on the long-term drift in estimates of $hmF2$

The supression of $foF2$ values during the summer months, together with the seasonal variation in the presence of the $F1$ layer are both indications of a change in thermospheric composition. Having shown that these can lead to a systematic bias when using an empirical formula to estimate $hmF2$, this raises the question as to whether such a bias would be introduced into the study of long-term change in the height of the F2 layer.

In order to investigate this, the relationship between the ISR and ionosonde-derived $hmF2$ values was determined for each of the 35 years of common data. For each year, (uncorrected) monthly median ionosonde-derived $hmF2$ values were plotted against mean ISR measurements for all hours and months where there were data from both instruments. For each year, a linear fit was made between ionosonde-derived and ISR $hmF2$ values. The resulting gradient and offset of each fit were used to derive a modelled height for an arbitrary ISR height of $250km$. The differences between these two values were used to reconstruct the percentage error in $hmF2$. The results of this analysis are presented in Figure 12 (a). It can be seen that there are solar cycle variations in the model error with an amplitude of $\pm10\%$ ($\pm25km$ at $250km$. This is of the order of decrease

expected from climate change). Moreover, there is a long-term drift in this error which would undoubtedly introduce a bias into any estimates of long-term change in the height of the $F2$ layer. The question then arises as to what could be the cause of this long-term change in the formula error. Figure 12 (c) presents the annual average of the geophysical $am$ index (Lockwood et al., 2019). We here choose to use the $am$ index rather than the $Ap$ index used in previous studies. The response patterns of the individual magnetic observatories used to compile such indexes depend strongly on the level of geomagnetic activity. At low activity levels the effect of solar zenith angle on ionospheric conductivity dominates over the effect of station proximity to the midnight-sector auroral oval, whereas the converse applies at high activity levels. It has been shown (Lockwood et al., 2019) that these biases are far smaller for the $am$ index than for $Ap$.

There is a strong and significant correlation (0.77, $p \ll 0.0001$), between the am index and the model error, suggesting that it is geomagnetic activity that is driving this variation in the accuracy of the empirical formula. If this long-term bias is consistent with the seasonal and diurnal bias of $hmF2$ estimates demonstrated in the earlier section of this paper, it would be reasonable to assume that the formula is being affected by changes to the underlying ionisation profile, introduced by long-term changes in thermospheric composition arising in turn from long-term changes in geomagnetic activity. Figure 12 (b) presents a qualitative proxy for the annual average thermospheric composition calculated from the square of monthly median noon $foF2$ values scaled by the solar $f10.7cm$ flux (Wright and Conkright, 2001). It can be seen that this proxy reveals very similar characteristics of a solar-cycle variation combined with a long-term decline (the correlation between these data and the model error is 0.835, $p \ll 0.0001$). More directly related to the earlier result that a bias is introduced into the empirical $hmF2$ formula by the presence of an $F1$ layer, figure 12 (d) presents the annual average $foF2/foE$ and $foF2/foF1$ ratios. These too demonstrate similar solar cycle variations combined with, for $foF2/foF1$ in particular, a long term decrease ($foF2/foF1$ correlation with model error is $0.86.p \ll 0.0001$; $foF2/foE$ correlation with model error is $0.70p \ll 0.0001$).

The sensitivity of thermospheric composition changes to geomagnetic activity varies with geomagnetic latitude (e.g. Zuzic et al. (1997)), with a station at low geomagnetic latitude being less prone to changes in molecular species at F-region altitudes than a station at a high geomagnetic latitude.

For example, Slough/Chilton, is a mid-latitude station in a geographic longitude sector near to the geomagnetic pole (at a geomagnetic latitude during this epoch of $\sim 48 - 50$ N). Here there is an annual variation in ionisation, with ionospheric densities being greatest in the winter. In the summer, the greater concentration of molecular species in the thermosphere increases the ionospheric loss rate, resulting in lower F-region ionospheric densities in the summer months where the proportion of molecular species is relatively high. In the winter months, down-welling of the meridional thermospheric circulation results in a thermospheric composition dominated by atomic species which have a lower loss rate. This seasonal change in composition exceeds the variation in ion production due to the seasonal change in solar zenith angle over the same period.

In contrast, Stanley in the Falkland Islands (at a geomagnetic latitude of $\sim 35 - 39S$ during this epoch) is a station that is far enough from the magnetic pole that compositional changes between equinox and winter months are relatively small compared with the associated change in solar zenith angle, resulting in a semiannual variation in $foF2$ (Millward et al., 1996). The relative magnitudes of the annual and semi-annual variations at a given station vary depending on geomagnetic activity, resulting in the long-term trends identified by Scott et al. (2014).

In contrast, the influence of compositional change on the peak concentrations of the $E$ and $F1$ layers is much smaller, since molecular ions exist in much greater proportions at these altitudes and loss rates are higher due to the comparatively high thermospheric densities.

Such differences are also likely to influence the relative values of the $foF2/foE$ and $foF2/foF1$ ratios at these stations. For example, the ratios at Slough/Chilton will be lower during the summer when compositional change suppresses $foF2$ while $foE$ and $foF1$ are at their peak. Figure 13 presents the mean annual $foF2/foE$ and $foF2/foF1$ ratios calculated for Slough/Chilton (Left hand column) and Stanely (right hand panel). With 2.5 and 1.6 representing the critical values below which a bias is introduced into the empirical formula used to calculate $hmF2$ (via $foF2/foE$ and $foF2/foF1$ respectively), it can be seen that Slough/Chilton will be far more susceptible to these biases than Stanley, where the mean $foF2/foE$ and $foF2/foF1$ ratios are higher, and a greater proportion of the values lie above these thresholds (shown in the figure as a dash-dot lines in the first and third rows). In addition, both stations exhibit some long-term change in these ratios which would introduce further bias into any estimates of long-term trends in $hmF2$. Such regional differences will need to be accounted for in any global analysis of $hmF2$ trends.

### 3.5 Accounting for signal delay in the estimate of $hmF2$ in the MU radar ISR data

As discussed previously, the above analysis has assumed that the propagation of the ISR radar pulses was not delayed by the presence of underlying ionisation. Which, for the frequency of the MU radar ($46.5MHz$) is expected to introduce a small but systematic offset. Since it is the bias in the height of the $F2$ peak we are interested in, the signal delay needs to be integrated along the path of the radio wave between the ground and the F2 peak, accounting for the upwards and downwards path of the signal.

By modelling the delay introduced to the time of flight by the signal interacting with the underlying ionisation, and comparing this with the known (modelled) height of the layer, an estimate can be made of this bias over a range of diurnal, seasonal and solar cycle conditions. While this is not an absolute measure of the delay occurring in the real-world data, it is sufficient to estimate the relative change in bias across a representative range of conditions.

In order to estimate this offset, the simplified Appleton-Hartree equation was applied, where;

$$\mu^2 = 1 - \frac{kN}{f^2} \tag{13}$$

where $k = 80.5$, $N$ is the electron concentration per $m^3$ and $f$ is the radar frequency. Applying the binomial expansion, this can be approximated to;

$$\mu = 1 - \frac{40.3N}{f^2} \tag{14}$$

Here, the second term on the right-hand side of the equation represents the range bias introduced as the radio wave passes through a plasma. In this way, $1\ TEC$ unit ($1e16$ electrons per $m^2$) delays the ISR signal by approximately $(1e16 * 40.3)/(46.5e6)^2 = 186m$.

In order to estimate the likely impact of such a bias in the data being considered, the 2016 International Reference Ionosphere (IRI2016) was used to generate electron concentration profiles at the location of the MU radar for dates between 1986 and 2020, corresponding to each month and hour considered in the study. Each ionospheric profile was then integrated from an altitude of 80 km up to the height of the maximum electron concentration in order to estimate the integrated range bias (doubled to account for the two way travel of the radio pulse).

The resulting values vary between $0.6$ and $11.5km$ with a median value of $1.47km$ varying as a function of peak electron concentration which, as expected, varies with time of day, season and solar cycle.

The matrix of height offsets was then subtracted from the $hmF2$ hourly monthly means derived from the ISR data and the analysis was repeated. The results (see Appendix A) showed similar biases, with the coefficients of the linear fit exhibiting slight changes (gradient 79.6, offset $-196.5$ for the $foF2/foE$ correction and a gradient of $46.6$ with an offset of $-74.1$ for the $foF|2/foF1$ correction). While it is not appropriate to apply these corrections to the individual points in the 35 year time series (since these have not been range-corrected), as these are linear fits, the small changes to the coefficients will result in similarly small changes to the corrected values. The underlying conclusions concerning the impact of the $foF2/foE$ and $foF2/foF1$ ratios on the empirical $hmF2$ formula are unaffected.

## 4   Conclusions

Empirical formulae used to estimate the height of the ionospheric F2 layer from standard parameters, scaled from ionograms, have necessarily had to make some assumptions about the underlying ionisation profile. We have shown that, for at least one of the established empirical formulae, that diurnal, seasonal and long-term biases are introduced into estimates of $hmF2$ that are of similar, if not greater, magnitude than those expected to be introduced by the long-term cooling resulting from increased levels of $CO2$ and $CH4$ in the lower atmosphere. While in the case of the Kokubunji station, the long-term bias is well correlated with long-term changes in geomagnetic activity, the physical mechanism is via changes to the underlying ionisation, driven by variation in thermospheric composition. This leads to diurnal, seasonal, and long-term variation in both the $foF2/foE$ and $foF2/foF1$ ratios that are not accounted for in the empirical formula.

When conducting their analysis of the Kokubunji data, Xu et al. (2004) used the formula of Bilitza et al. (1979). While a direct comparison cannot be made with the current analysis, the variability in long-term trends observed by Xu et al. (2004) (difference between long-term trends in noon and midnight $hmF2$, with the seasonal variation at these two times being opposite to each other) is consistent with a maximum bias occurring around noon in the summer months. Xu et al. (2004) also conclude that geomagnetic activity was not significant in the regression model used to remove the effects of geomagnetic and solar variability. This could be attributed to their use of a different empirical model. Nevertheless, our analysis indicates that variations in the bias of $hmF2$ estimates is likely driven by geomagnetic activity.

As noted in the introduction, geomagnetic activity may also induce change in global thermospheric circulation with changes in the meridional wind modulating the height of F2-layer. Titheridge (1995) reviews the magnitude of these effects. A poleward wind would move ionisation to lower altitudes, where the loss rate is higher. This would lead to a decrease in the peak $F2$ ionisation. Under such circumstances, the $foF2/foE$ ratio could potentially become sufficiently small that, in addition to the genuine change in layer height, the empirical formula would start to underestimate $hmF2$. More work is needed to deconvolve the relative magnitude of these effects but whether driven by changes in the wind-field or local changes in composition, geomagnetic activity can lead to long-term bias in estimates of $hmF2$ when using an empirical formula.

While the wider family of empirical formulae have not been tested in this work, there is evidence (McNamara, 2008) that at least some of these empirical formulae exhibit seasonal bias. Furthermore, there is evidence that, while being driven by geomagnetic activity, long-term change in ionospheric composition can be geographically localised, with individual stations exhibiting a wide range of responses to geomagnetic activity (Scott et al., 2014; Scott and Stamper, 2015). We conclude that the lack of consistency in global estimates of long-term changes in $hmF2$ results from the localised nature of the long-term changes in thermospheric composition not accounted for in the empirical formula used.

Jarvis et al. (1998) reported an altitude-dependence in their estimates of long-term change in $hmF2$ and hypothesised physical mechanisms that would explain this. Our results indicate that such mechanisms do not need to be invoked, rather that the bias introduced into the formula affects the percentage uncertainty in the estimate of $hmF2$, which would lead to a bias that would also be altitude dependant.

It may be possible to use the relationship between the $foF2/foE$ and/or $foF2/foF1$ ratios and the formula bias to correct for long-term changes in thermospheric composition for this station, and it is also likely that the equivalent ratios at other stations could be used to account for the global variations in this bias to produce a unified estimate of the rate of long-term change in $hmF2$. Caution should be used in such an exercise however, since the bias in the formula varies with season and time of day. In addition, the formula used was calibrated for a specific station and the sensitivity to these biases may vary with location. Other variations on the formula should be tested in this way to determine their relative sensitivities to compositional effects. It would be interesting to see if the biases determined in the present study vary with location by conducting similar calibrations using other ISR stations.

With the potential for biases within these empirical $hmF2$ formulae, the ideal approach would be to determine such trends from alternative sources such as directly from ISR measurements (which, as pointed out by Rishbeth (1999) will require a few more decades of measurements before any trends can be considered significant) or through the labour-intensive process of inverting ionogram profiles. While this latter suggestion is theoretically possible for stations such as Slough/Chilton for which the original ionograms still exist, such a task is currently beyond the scope of this analysis, requiring careful digitisation, scaling and verification across many generations of instruments and data formats.

While this work has not addressed any potential bias introduced by long-term changes to thermospheric circulation or geomagnetic field, it has nonetheless demonstrated a bias in the formula that, through long-term changes in thermospheric composition, can lead to localised biases in the estimates of $hmF2$ which in turn can explain the lack of global consensus in long-term changes in the height of the ionosphere. Importantly, the results from this paper show that diurnal, seasonal and

long-term biases are introduced into estimates of trends in ionospheric heights that are of similar, if not greater, magnitude than those expected to be introduced by the long-term thermospheric cooling. These analysis issues must be addressed before ionospheric observations can be correctly interpreted in relation to long-term climate models.

*Code and data availability.* Software used in the analysis of these data is available via https://github.com/cscott42/hmF2CalibrationCode (This will be deposited on Zenodo on acceptance of the paper). MU radar data was provided by the Research Institute for Sustainable Humanosphere of Kyoto University can be obtained from their website at https://www.rish.kyoto-u.ac.jp/mu/isdata/. Ionospheric data used in this analysis are available via the UK Solar System Data Centre at https://www.ukssdc.ac.uk.

## Appendix A: Appendix A: Additional Figures

In this section, additional plots are presented of the average seasonal and diurnal variation in individual ionospheric parameters used to estimate $hmF2$ values via the empirical formula, and how the estimated range correction for the ISR (as detailed in section 3.5) affects the observed biases between ISR and ionosonde-derived estimates of $hmF2$. As with the analysis described in the main body of the paper, monthly median values were calculated for each hour of data and these monthly medians have then been averaged over the 35 years of common ionosonde/ISR data.

*Author contributions.* C. J. Scott lead on the data analysis and interpretation. M. N. Wild advised on and provided the ionosonde data, L. A. Barnard and B. Yu contributed to the data analysis, T. Yokoyama advised on the analysis of MU radar data, M. Lockwood advised on analysis and geomagnetic indices, C. Mitchel conducted analysis on the range correction of MU radar data, J. Coxon provided early insight into data analysis methods and A. Kavanagh provided access to important reference material.

*Competing interests.* At least one of the (co-)authors is a member of the editorial board of Annales Geophysicae.

*Acknowledgements.* The authors would like to thank the Research Institute for Sustainable Humanosphere of Kyoto University for providing the MU radar data, WDC for Ionosphere and Space Weather, Tokyo, National Institute of Information and Communications Technology for providing the Kokubunji ionosonde data, which was also made available through the UK Solar System Data Centre. This work was funded under the UKRI grant NE/W003384/1.

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

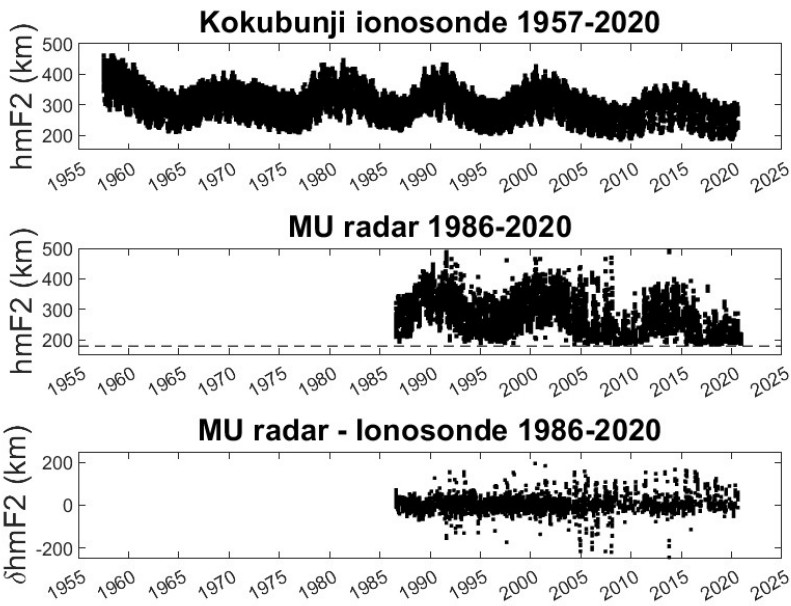

**Figure 1.** Hourly monthly median $hmF2$ values derived from the Kokubunji ionosonde data (top panel). compared with hourly mean $hmF2$ values derived from the MU radar (middle panel). The difference (ISR - ionosonde) between common hmF2 values is shown in the lower panel. It can be seen that the ISR data are noisier, due to there being relatively fewer data points from which the mean values are calculated. Also at solar minimum years some values are close to the $180km$ floor of the altitude window in which the F2 peaks were identified.

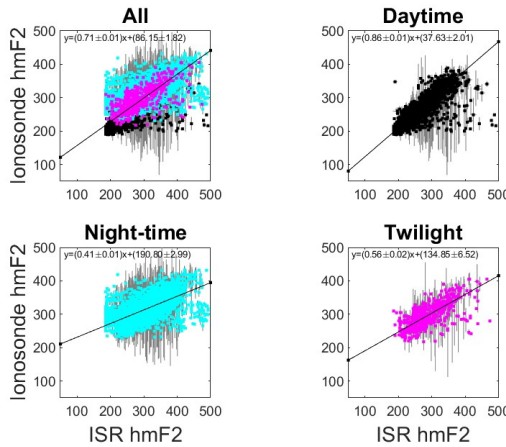

**Figure 2.** Hourly monthly median $hmF2$ values derived from the Kokubunji ionosonde data versus hourly mean $hmF2$ values derived from the MU radar. The four panels show all data (top left), daytime data only (top right), nighttime only (bottom left) and twilight only (bottom right). A linear fit between the data sets for all data (black line, top right panel) has a gradient of $0.71 \pm 0.01$ ($R^2 = 0.51$) while for daytime only (solar zenith angle $< 900°$, black line, top left panel) the gradient is $0.86 \pm 0.01$ ($R^2 = 0.65$). A linear fit to the twilight values (lower right panel) for which solar zenith angle is between $90°$ and $100°$) has a reduced gradient of $0.56 \pm 0.02$ ($R^2 = 0.44$) while nighttime values (solar zenith angle $> 100°$) show much greater scatter with a fit gradient of $0.41 \pm 0.01$ ($R^2 = 0.33$).

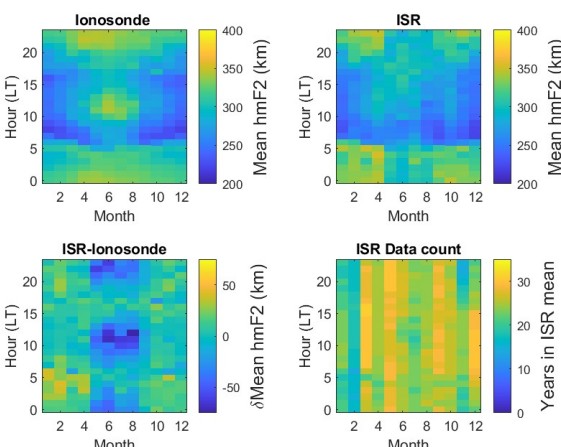

**Figure 3.** Monthly and hourly $hmF2$ values averaged over the 35 years of common data (1986-2020) used in this study. The dotted and solid white lines overlaid on these plots represents the times at which the solar zenith angle is $90°$ and $100°$ respectively.Both ionosonde (top left) and ISR (top right) derived $hmF2$ values show clear seasonal and diurnal variations, with higher $hmF2$ values at night. The difference between these two datasets, $\delta hmF2$ (ISR-ionosonde, lower left panel) highlights that this difference is greatest in the summer around noon (and midnight) with the ionosonde estimates of $hmF2$ exceeding the ISR measurements. Since the ISR is run on a campaign basis, the data contributing to the mean values in each bin could be affected by low data counts. The lower right panel presents the number of ISR data contributing to the mean value in each bin. the minimum number of ISR data available in a bin was 14, the maximum number was 31 and the median was 25.

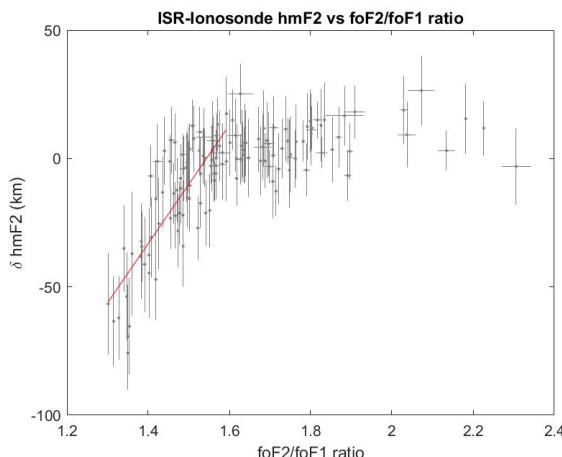

**Figure 4.** The difference between the ISR and ionosonde distrbutions shown in figure 3 plotted against the $foF2/foF1$ ratio (for hours and months where such values exist). There is no significant difference between the two distributions for values of the $foF2/foF1$ ratio above 1.6 but below this ratio, the presence of an $foF1$ layer significantly affects the ionosonde-derived $hmF2$ values due to the presence of underlying ionisation that is unaccounted for in the empirical formula. The red line represents a fit to the values corresponding to an $foF2/foF1$ ratio below 1.6 (gradient $231 \pm 21$, offset $-356 \pm 31km$

)

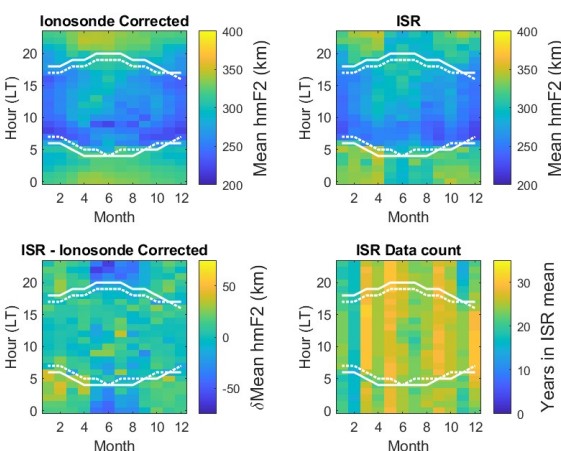

**Figure 5.** The same as for figure 3 but now with correction applied for bins where $foF2/foF1 \leq 1.6$. This has brought the summer daytime hmF2 values into closer agreement with those observed by the ISR (top right panel)

.

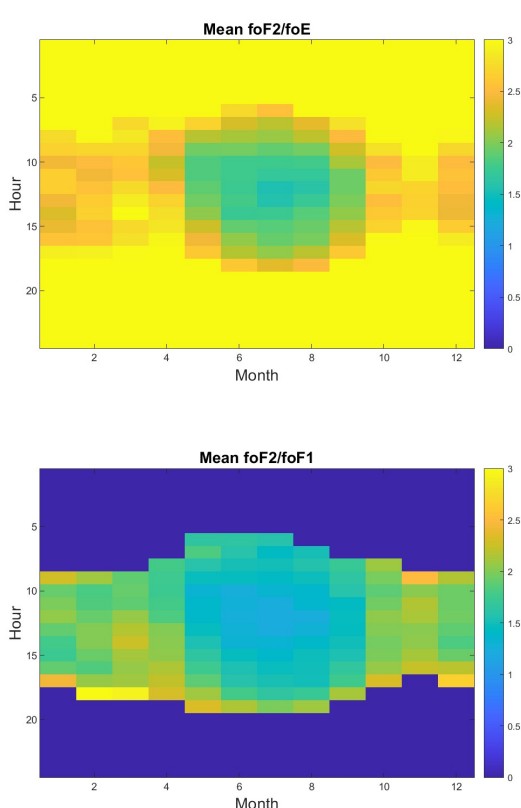

**Figure 6.** A comparison between the seasonal and diurnal variation in the ratios $foF2/foE$ (top panel) and $foF2/foF1$ (lower panel). It can be seen that both ratios follow the same trends, with distinct minima around noon in the summer months.

.

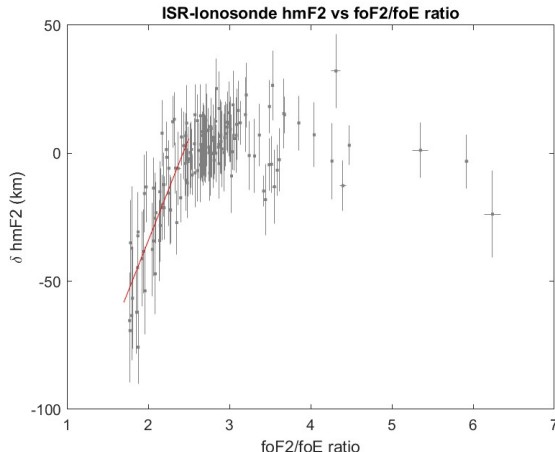

**Figure 7.** Similar to 4 but with $\delta hmF2$ binned by the associated value of the $foF2/foE$ ratio. A significant bias is introduced for values of this ratio below $\sim 2.5$. A robust linear fit to these values has a gradient of $80 \pm 9$ and an intercept value of $-194 \pm 19$.

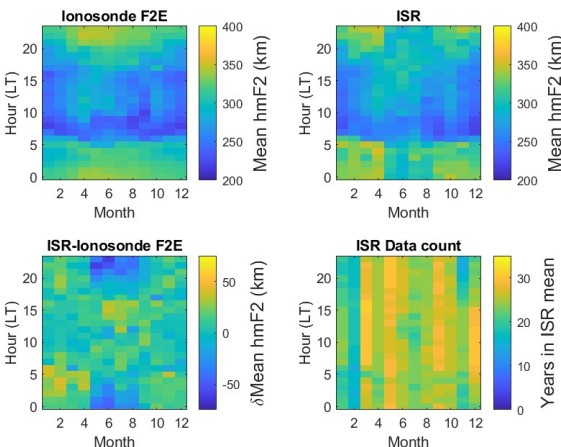

**Figure 8.** The same as for figure 3 but this time with the ionosonde-derived $hmF2$ values corrected for the bias introduced by low values of the $foF2/foE$ ratio. Corrected ionosonde (top left) and ISR (top right) derived $hmF2$ values show close agreement, with the summer daytime peak in hmF2 now supressed. This difference between these two datasets, $\delta hmF2$ (ISR-ionosonde, lower left panel) highlights that the difference in the summer around noon is much reduced.

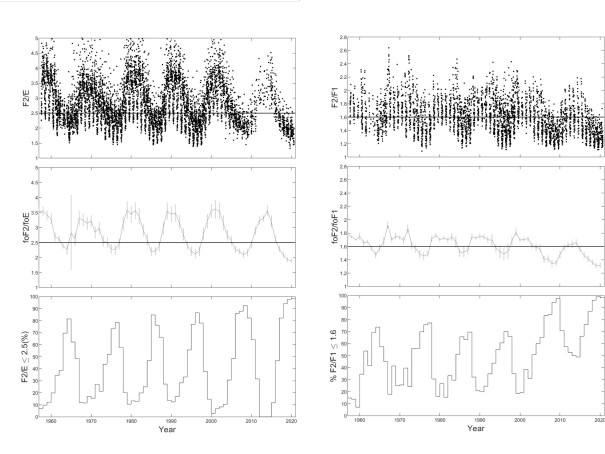

**Figure 9.** $foF2/foF1$ (left hand column) and $foF2/foE$ (right hand column) ratios against year for hourly monthly median values (top panels), annual averages (middle panels) and (lower panels) the percentage of observations (where ratios can be calculated) for which the ratios fall below the bias threshold ($\leq 1.6$ for $foF2/foE$ and 2.5 $foF2/foF1$. It can be seen that there is a strong solar cycle dependence in both of these ratios, together with longer-term changes, particularly the apparent step change since the year 2000.

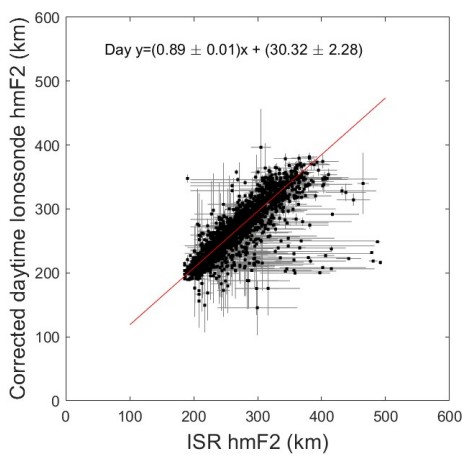

**Figure 10.** Hourly monthly median ionosonde-derved $hmF2$ values, corrected for the presence of $foF1$, plotted against monthly mean $hmF2$ values determined from the ISR, the correction results in a revised gradient of $0.89 \pm 0.01$ with an offset of $30.32 \pm 2.28km$ (given by the red line).

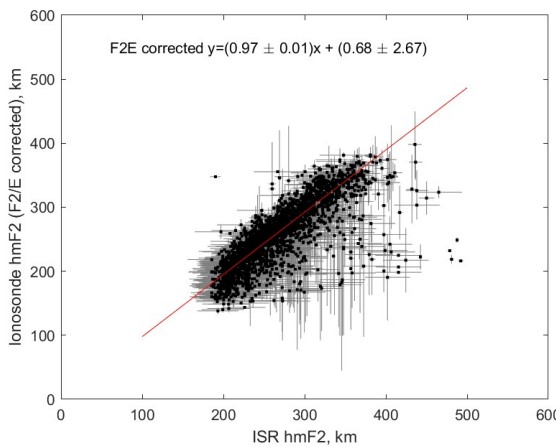

**Figure 11.** The same as figure 10 but this time corrected for the bias introduced for values of the $foF2/foE$ ratio below $2.5$. The correction results in a revised gradient of $0.97 \pm 0.01$ with an offset of $0.68 \pm 2.67 km$ (given by the red line).

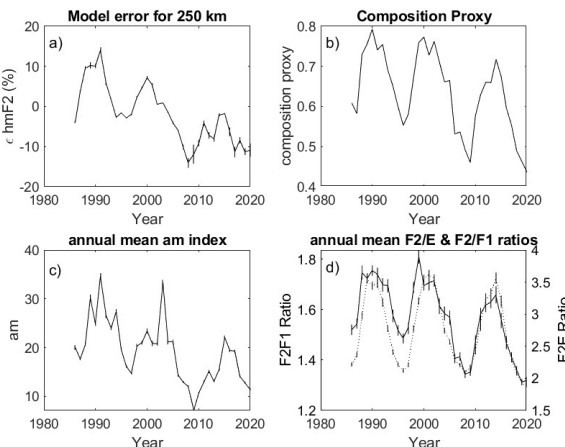

**Figure 12.** a) Percentage error in the empirical $hmF2$ model, $\epsilon hmF2$ after calibration with ISR data for each year in the range 1986-2020. b) Qualitative composition proxy based on annual average noon foF2 value scaled by solar $f10.7cm$ flux. c) Annual average values of the am index or the same epoch. d) The annual average ratios in $foF2/foF1$ (solid line) and $foF2/foE$ (dotted line) for the same epoch.

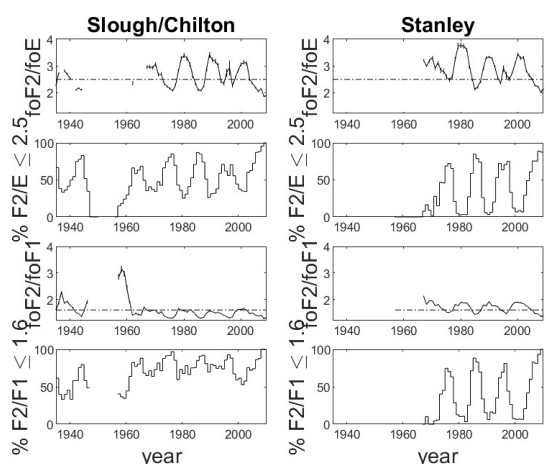

**Figure 13.** Annual mean $foF2/foE$ (top row) and $foF2 : foF1$ (third row) ratios calculated from monthly hourly median data for two stations Slough/Chilton in the UK (left hand column) and Stanley in the Falkland Islands (right hand column). Ratios less than 2.5 (for $foF2/foE$ and 1.6 (for $foF2/foF1$) have been shown to introduce bias into the empirical formula used to calculate $hmF2$ from ionosonde data. There are marked differences in the sensitivity to these biases between the two stations. The $foF2/foE$ and $foF2/foF1$ ratios remains below the respective thresholds for most of the data in the Slough sequence, making it far more sensitive to bias in $hmF2$ calculations than Stanley, where the ratio is above the thresholds for a far greater proportion of the data. The fraction of the data (for which each ratio can be calculated) also varies with time (second and fourth rows), which will lead to a bias in any long-term trend in $hmF2$ calculated using the empirical formula. The sensitivity to this long-term bias will also differ between stations.

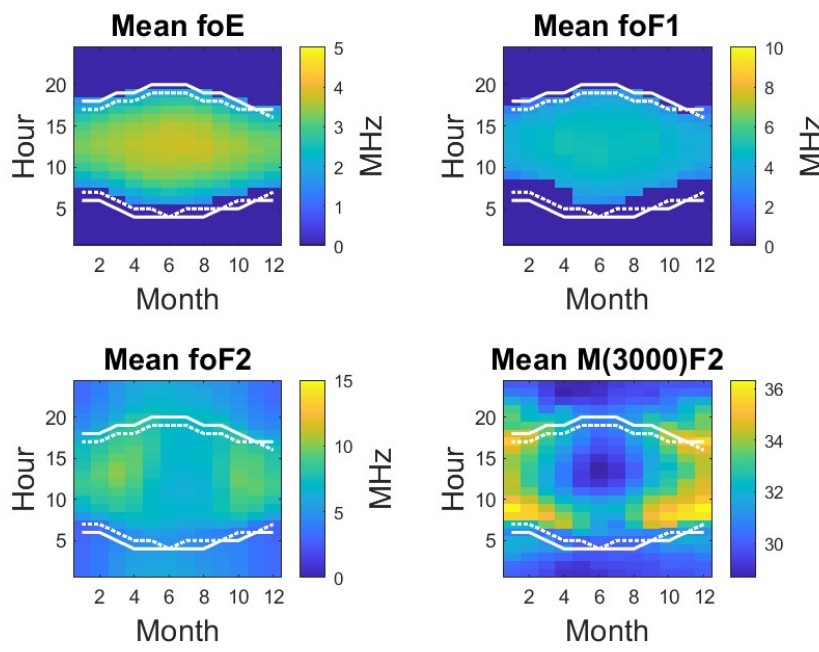

**Figure A1.** Hourly monthly median values of individual ionospheric parameters, averaged over the 35 years of common ionosonde and ISR data. the E-layer peak frequency, $foF2$ (top left, in $MHz$) peaks around noon in the summer months, corresponding to peak ion production (smallest zenith angle) and is absent at night. This same variation is seen for the F1-layer peak frequency, $foF1$ (top right, in $MHz$), which is greatest around noon in the summer months. In contrast, the F2-layer peak frequency (lower left panel, in $MHz$) shows a minima around noon in the summer months. The $M(3000)F2$ factor (lower right panel) is calculated from the ratio of the Maximum Usable Frequency ($MUF$) and $foF2$. It too has a distinct minima around noon in the summer months.

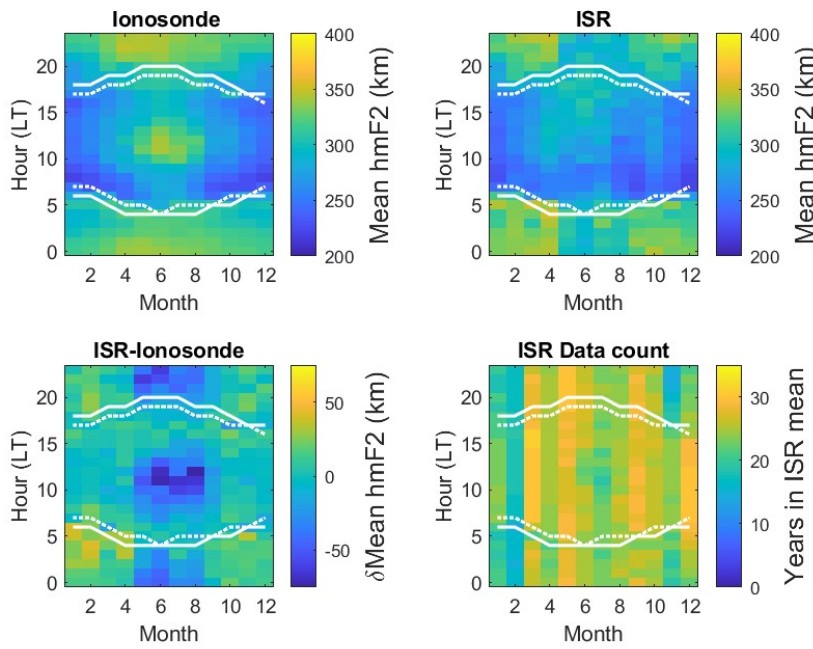

**Figure A2.** The same as for figure 3 but with the estimated ISR range corrections applied to each of the 12x24 bins in the seasonal/diurnal average data. Application of the estimated range correction has not significantly changed the observed seasonal/diurnal variations.

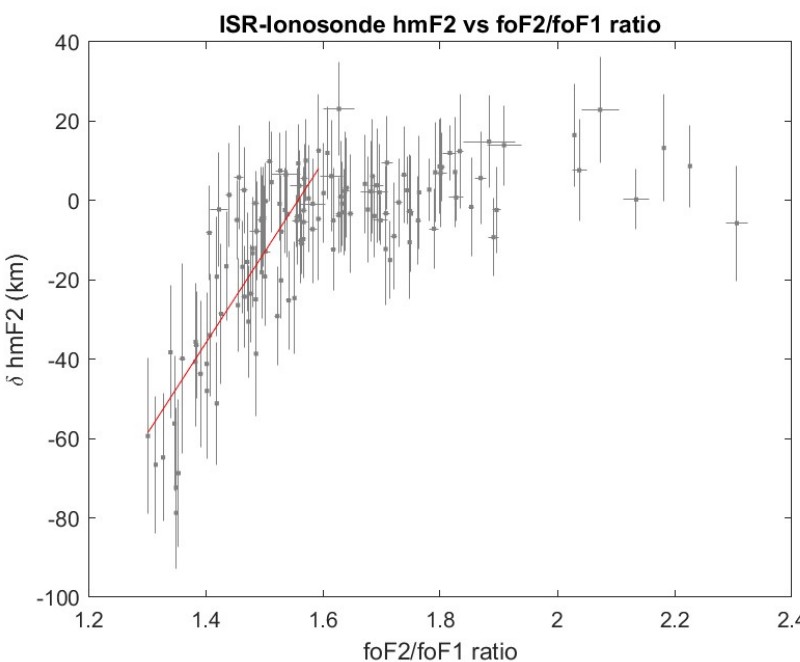

**Figure A3.** The same as for figure 4 but with the estimated ISR range corrections applied to each of the 12x24 bins in the seasonal/diurnal average ISR data. Application of the estimated range correction does not remove the observed bias below the threshold of $foF2/foF1 = 1.6$, though the resulting fitted gradient changes to $46.6$ with an offset of $-74.1$.

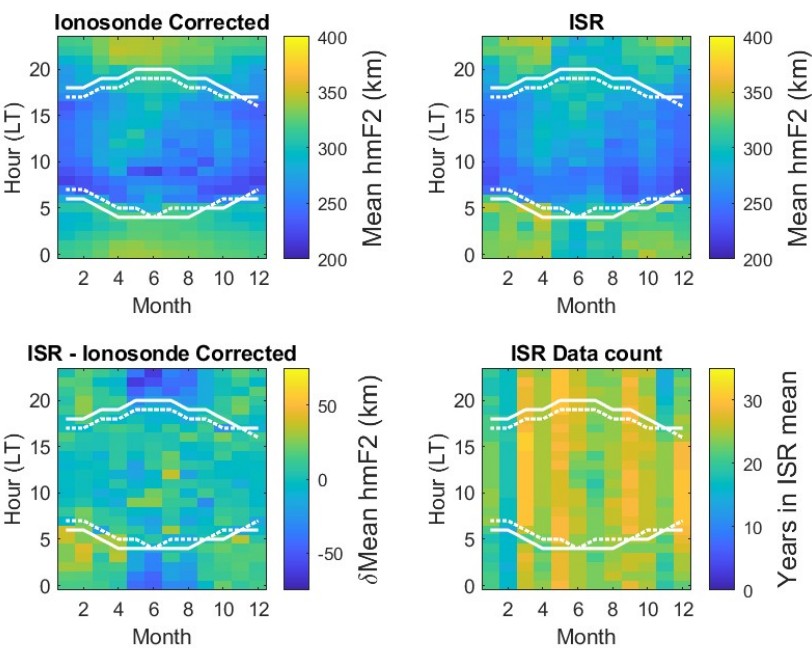

**Figure A4.** The same as for figure 5 but with the estimated ISR range corrections applied to each of the 12x24 bins in the seasonal/diurnal average ISR data. The corrected data is very similar in form, with the anomalous peak around noon in the summer months now reduced, which better matches the ISR data.

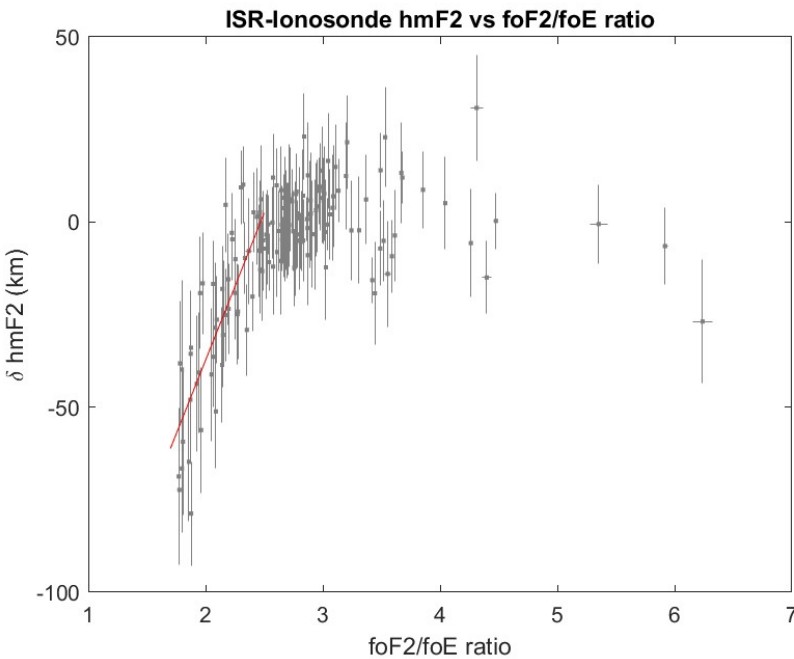

**Figure A5.** The same as for figure 7 but with the estimated ISR range corrections applied to each of the 12x24 bins in the seasonal/diurnal average ISR data. Application of the estimated range correction does not remove the observed bias below the threshold of $foF2/foE = 2.5$, though the resulting fitted gradient changes to 79.6 with an offset of $-196.5$.

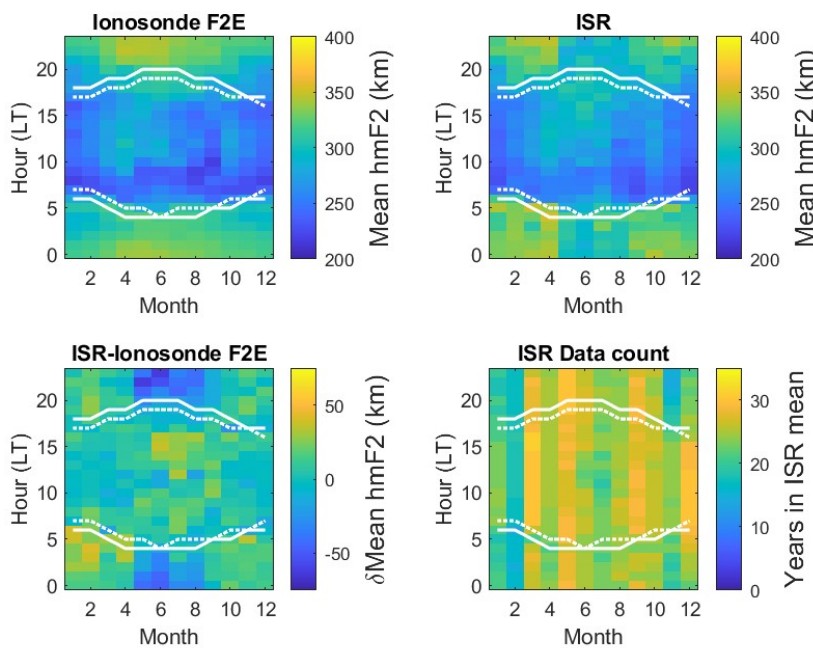

**Figure A6.** The same as for figure 8 but with the estimated ISR range corrections applied to each of the 12x24 bins in the seasonal/diurnal average ISR data. The corrected data is very similar in form, with the anomalous peak around noon in the summer months now reduced, which better matches the ISR data.