# Peer review of "Calibrating estimates of ionospheric long-term change"

_EGUsphere, 2023_

## Referee Comment (RC2)

**Review of Scott+ 2023 "Calibrating estimates of ionospheric long-term change" (egusphere-2023-2599)**

**Contents**

**General comments**

Scott et al. consider the derivation of ionospheric F2 layer heights from historic scaled ionosonde parameters. These parameters have been recorded since the early 20th century, and in principle provide a very valuable long-term record in which to seek an expected long-term ~20 km decrease in this F2 layer arising from climate change. Similar expected decreases are harder to discern in other upper atmosphere layers, where records are also shorter. Consequently, many previous authors have examined this ionospheric data, trying to identify a secular long-term trend, but this has turned out to be difficult, with inconsistent, countervailing trends emerging from different ionospheric observations worldwide.

Scott et al. provide a valuable synthesis of the state of the art, and make a compelling, novel, and lucidly-written case that previous analyses have relied on semi-empirical equations which incorrectly assume any F1 layer present has an insignificant effect. Taking ionosonde data through such an equation, and comparing the resulting imputed F2 layer heights with more reliable estimates from an incoherent scatter radar, they show F1 layers can cause biases in height estimates a little larger than the climate trend being sought, on various timescales. Hence risking masking any climate trends, or causing misattributions. They attribute this F1 effect to thermospheric composition effects, driven by geomagnetic changes affecting energy deposition into the thermosphere, and resulting impacts on thermospheric heating and circulation.

The quality of the analysis and manuscript is generally very good, and nearly ready for publication. However to ensure the work reaches its full potential, becoming an extremely valuable reference for such efforts, there are a few areas which need addressing. Some where the authors seem to have gone a little fast in their analysis, and where their case needs to be evidenced a little better, to ensure their conclusions are as robust as possible. Or where the authors can make their reasoning more explicit. The remarks below are quite fussy, holding the authors to a very high standard. They are invited to push back as appropriate!

**Specific comments**

*Note B&D abbreviation used throughout = Bradley & Dudeney.*

**Attribution of effect to F1: tighten up composition link**

You've made a fairly compelling case that F1 can contaminate hmF2 values:

Pretty strong: evidence

- Fig 4, showing clear break in ISR-ionosonde hmF2 bias, ordered by xF = foF2/foF1
- Fig 5, showing you can use the trend from smaller xF in Fig 4 to successfully remove the summer daytime hmF2 bias
- Fig 8a, showing (modelled) hmF2 bias has a trend and timescales which could lead to confounding with climate signal

Slightly weaker however: foF1 <-> composition link

- Physical rationale (introduced ~L364, invoked ~L405) linking presence of foF1 to composition changes (may be a priori knowledge I lack - make this more explicit if so!)
- Qualitative support from composition proxy in 8b showing similar trend to 8a Am not familiar with this Wright & Conkright proxy, and I note comment CC1 queries it...
- Discussion of annual/semi-annual circulation-induced changes to composition in L430- L447 seems to suggest foF2 is more affected than foF1: L441 "foF2 is suppressed... when foF1 is at its peak"

So it would be good if the foF1 <-> composition link rationale could be tightened up a little

**Attribution of effect to F1: exclude E as a confounder**

While the F1 evidence *seems* good, I'm a bit concerned that the apparent "smoking gun" in the derived parameters means you may not have looked at the raw data, to see if there are other potential causes for the patterns you've discerned.

*Apologies - perfectly possible you've done this already, discovered it's not a concern, and decided not to clutter paper with this. If this has happened though, I'd recommend you say so explicitly, along lines of "An examination of ABC (not shown) excludes XYZ"*

Specific areas of concern:

- the domain of validity for B&D model is for xE (= foF2/foE) > 1.7
- both xF and xE share a common numerator, so I'd expect them to be somewhat correlated

Data-wise:

- What happens if you create an equivalent to Fig 4, but order the hmF2 bias by xE?
  - Is there a similar "break" partway along?
  - Shouldn't be, if the B&D model for hmF2(xE) is decent along its validity range...
  - but worth double-checking, in case of any nasty non-linearities?
- Likewise does any of your data have xE <= 1.7 (i.e. outside of B&D validity range)?
  - What happens to bias then?

Physically:

- Might composition changes affecting foF1 also affect foE?

I think it's worth being certain that the attribution to foF1 contamination is rock-solid.

Not least as a foE effect might be good news: given foE scaled more frequently than foF1, if any effect can be characterised, *might* give more data for looking at trends globally using already-manually scaled parameters, without data-rescue-based inversions?

See my suggestion below on a few supplementary material plots for probing foF1 attribution.

- there may be other ways of excluding potential confounders of foF1 attribution

- not least simply telling me if I'm wide of mark here!

**Enhance existing plots**

There are various places where the existing plots could be enhanced, to reveal details which are currently hard to see, or which aren't fully probed.

A few suggestions below. From your code, I think all should be ~easy to add.

**Figure 1: flag 180 km limit, plot delta_hmF2**

On ISR subpanel, worth adding a horizontal line / note in caption that there's a 180 km threshold getting imposed to avoid sporadic E layer?

- This to help readers spot this in subsequent plots
- And consider this when ISR is acting as source of truth

Add a 3rd subpanel below current two, plotting `delta_hmF2 = hmF2_ISR - hmF2_ionosonde`

This will help:

- act as a hook for you to define delta_hmf2 in text (not currently done - see pedantry below on why useful)
- make the bias (higher ionosonde hmF2) clearer
    - I can see this by concentrating hard, but much easier to spot on a difference plot
- allow upfront spotting of any temporal patterns in raw delta_hmf2 (later plots are processed)
- help readers reason about intercept in Fig 2, 7
- act as hook for decimated delta_hmF2 in Fig 4, modelled delta_hmF2 in 8a

Pedantic quibble - no need to change sense, just why IMO useful to define delta_hmF2:

- above seems to be the definition for delta_hmF2 you use in 4, 8a
- but in this paper, you treat hmF2 from ISR as the "truth"/reference value
- so definition is ~backward cf the usual definition of change = value - reference
- a little counterintuitive, so adds to why worth explicitly defining delta_hmF2 in text

*Note if you **did** change sense, delta_hmF2 values would be biased to positive values, so making the sign consistent with the positive offsets in Figs 2, 7*

**Figure 2: break out the day/night/terminator populations**

Recommend changing this to a 2x2 subpanel plot

- keep current plot as an "all populations" subpanel
- but use other 3 subpanels to plot each day/night/terminator population separately

I say this as you've gone to some trouble to stratify the three populations, and I think it's worth getting all the benefit out!

Current plot is good for general intercomparison of the three populations, and reasoning about the "all" fit. But overplotting means it's hard to see & assess any detailed features in each population.

E.g. there seem to be ~two distinct clumps in the nighttime population, above/below the main trendline, at low/high ISR hmF2. But as everything is overplotted, it's hard to assess if there's anything similar in the twilight / day populations, e.g.

- do they have similar clumps?
- if so, do their "upper" clumps also contain ISR hmF2s which hit the 180 km floor?

Likewise hard to see scatter cores - is anything interesting lurking in there?

Overall makes it harder to assess validity of the linear fits. These look OK, but good to reassure reader before these get used (Fig 5, 8a)

**Figure 3: plot the delta_hmF2; bin count / 180 km count**

These plots are great for probing the seasonal and local time behaviour, but I think they could be made even more revealing.

Again I'd recommend inflating this to a 2x2 plot. Keep the current two, but add:

- 3: subpanel of delta_hmF2
- 4: subpanel of number of year bins [0 - 35] which have contributed to the ISR mean value in current lower panel

3: to help readers spot the differences, and quantify these. Very hard to compare colours by eye! Might also reveal more subtle differences.

4: to check there's no concerns about quality of ISR averages in any bins I *hope* that the 48% occupancy you report for the 35 years means there's ~16-17 counts per ISR month-localtime bin. But I think worth exposing bin count to be sure:

- as you're treating ISR as ~truth-ier than ionosonde
- as ISR is a campaign-based instrument, and campaigns might clump in season-localtime space!
- as this ISR climatology seems to be showing a few interesting features (e.g. dips in summertime night hmF2) - worth checking these are robust!
    - are these summertime night dips likely to be real?
    - if not, could they be contributing to bias?

alternative 4: fraction of bin with ISR hmF2 = 180 km If the ISR bins all turn out to have OK-ish occupancy, flat enough that you don't think there's an issue, I think you can simply report this in text, and use the subpanel to look at other minor concern instead:

- The 180 km threshold on ISR will skew the bin mean value
- and in worst case (all "true" ISR hmF2 values in bin across years < 180 km) set it to 180

This might be affecting some of the blue-er bins. You could quantify this by calculating what fraction of the counts in each bin have values == 180 km.

**Figure 5: plot delta_hmF2**

The daytime summer peak has clearly gone, but it's hard to compare colours here to Fig 3b, to get anything more out of this.

Again, recommend adding a subpanel for delta_hmF2 - help readers compare with the previous delta_hmF2, see if there's anything more here.

Will also support your discussion of the nighttime hmF2 differences remaining

- in L365
- in L388 - let you be a bit more nuanced - your "generally lower" point may apply across all months, but doesn't apply in summer months.

**Create supplementary material plots**

The plots you present are examining the hmF2 or foF2/foF1 ratio. You don't present plots for individual components which go into these derived values.

I've suggested a few supplementary material plots below, to do a deeper dive into the underlying ionosonde data you've retrieved and processed, to:

- ensure that the attribution of hmF2 bias to F1 contamination is correct - check that none of the other parameters (e.g. foE) yield similar behaviour, i.e. act as confounding factors
- check that there aren't any compensating errors occurring

Seems worth extracting all you can out of this - may even give some extra insights!

Looking at your code, I think should be ~easy to generate such plots. Realise this deeper dive is extra work, and on margins of "out of scope for this paper"! Will leave you to judge merit & feasibility!

Specific plots I'm considering you create:

1. For long-term trends: stackplot of hourly monthly median timeseries, like Fig 1, but for the individual foE, foF1, foF2, and M(3000)F2 components. Should show if there's any striking long-term trends in m/any of these individually

2. For seasonal and diurnal patterns: 3x2 subpanels of month-localtime bins, meaned over ionosonde data years. So like 3a, but showing:

   - 2a: foE
   - 2b: foF1
   - 2c: foF2
   - 2d: M(3000)F2
   - 2e: xE i.e. foF2/foE
   - 2f: xF i.e. foF2/foF1

Will help test validity of foF1 attribution, and more, by giving more insights into:

- xF: L479 discusses diurnal & seasonal variation - don't think you've shown this?
  - with this plot, you would!
- xE > 1.7 domain of validity for B&D eqn 4 you're using
  - are there any times & seasons where this is breached
- probing any correlation between xE and xF
  - do both show a summer midday peak, or only xF?
- foE 0.5 MHz assumption for low foE
  - what local times / seasons does this affect most? Is this what you expect?
- M(3000)F2 not accounting for mag field
  - what are values here, how does it compare to Elias?
  - probing conclusions in L404
  - Even do you see Russell-McPheron type variation signatures?
- foF1: help illustrate climatology
  - quantify point in Fig 4 caption "for hours and months where such values exist"
  - similar for L365 "only visible during daylight hours"

**Better justify attribution of scatter to foE floor at 0.4 MHz**

In discussion of Fig 2 ~L345, you tentatively attribute the scatter to the imposed floor of foE = 0.4 MHz.

Later, in L368 & L389, you strengthen this conclusion - I'm concerned too much. Especially as in L390 you link scatter to ionosonde underestimating (? see typos) hmF2. I assume on basis of offset? Is this scatter though? Or gradient?

As you'll see in technical section, I played with B&D equation in Excel. I set "true" foE in range 0.001 to 0.399, set the M(3000)F2 and foF2 to fixed values, and compared resulting hmF2 to the hmF2 value I got with imposed foE = 0.4 MHz. The hmF2 values different by ~3 to 0 km.

Such small OoM ~1 km sizes, compared to the ~50 km (rough width of "core" scatter) in Fig 2 makes me query if this attribution holds.

But maybe I'm not using appropriate M(3000)F2 and foF2 values, or thinking about this in the right way.

Can you dig into this a little more please?

**Check/discuss error bars in Figs 2, 7**

Can you check the error bars are implemented correctly? And if correct, briefly discuss relative sizes.

By eye, looks like they're often much larger in vertical cf horizontal direction

- I may be mistaken, as for Fig 2 this is dominated by what I can see in outliers - subpanels would help!
- Looks slightly more equal in Fig 4

If true, seems slightly surprising to me, as your 48% figure above means ionosonde errors are nominally being constrained by considerably more data than your ISR errors. So if errors of equal magnitude, on standard error basis, I'd expect smaller (~70%) error bars for ionosonde.

But in Fig 2, bigger ionosonde error bars suggest ionosonde errors are inherently larger.

- But this doesn't seem to square up with Figure 1, where ISR looks noisier?

I've not looked at your code here, but worth a quick check that error bar axes are consistent with point axes, that things haven't been swapped accidentally?

**Check unexpected striations in time series scatter plots**

Can you double-check your plotting code in Fig 6 (likely applies to Fig 1 too)? And if all is coded OK, explain how the striations in panel a are arising?

The striations in panel a seem a bit weird - at this zoom level I'd not expect to see these.

I think you're plotting your binned results here, and if I've understood your binning code correctly, bin values for a given month would be would be 24 vertically-aligned points (all local time bins for that month), and a year 12 such vertical lines (all month bins).

So I'd expect to see striations if the plot was on timescales of a few years.

But on the multi-decades timescale, being examined here, you should have 120 lines / decade. Which shouldn't I think give rise to any visible stripes. Whereas I seem to see stripes on ~annual cadence.

I guess striations *could* arise if there's something in orig data / your post-processing decimating F2/F1 bin entries to NaNs preferentially in one half of year - say summer.

But from discussion so far, don't think there is anything like this? (Again the supplementary plots requested might help diagnose anything like this here!)

In absence of something like decimation, I wonder if there's a mistake in plotting code? e.g. month going into a date plotting routine in a day index position?

Wild guess - I've not checked code to see if there's anything like this!

**Flag up more clearly where your findings differ from others'**

There are a few places throughout intro sections where you flag up ~detailed points from previous literature - often I think as these run counter to what you find.

But you don't refer back to these when you present your results / conclusions.

Upshot is that it's not easy for the reader to spot these - requires multiple passes.

I'd recommend flagging any such disagreements up more explicitly somewhere: either in results as you go, or aggregating somewhere just before/in conclusions.

Ones I've spotted (may well be more!)

- L119: Xu not finding significant geomag effect (cf your Fig 8)
- L215: B&D claiming F1 insignificant. OK, "claim", and it's ~clear that your whole paper is contesting this general point! But if B&D / derivatives get used as much as L246 implies, I think worth being blunt somewhere "B&D's claim is wrong"!
- L487: the juxtaposition is a little too subtle for me - think you could make if clearer that that you're ~disagreeing with Jarvis. How about something like following?

```
@@ L488 @@
-explain this. Our results indicate that
+explain this. Our results indicate such mechanisms do not need to be invoked, rather that
```

**Minor points**

L212: "They noted that `xE > 1.7` is equivalent to about `xF ≈ 1.2`"

- Where are you getting 1.2 from? Looking at B&D I can't see this.

Closest match their sentence below, but I don't think this yields this equivalence. "(In the limit for x = 1.7, f1 = foF2 and we have an entirely linear FI/F2-layer.)"

L164: discussion of routine estimates of hmF2 from digisondes

Likely ~irrelevant/tangential cf the manually-scaled parameters considered in this paper (so not necessarily suggesting you cite it) but in case you missed it, worth sharing Themens+' interesting demonstration that ARTIST's confidence scores for auto-scaled features (incl hmF2) need to be treated carefully.

Possibly of more relevance to here: they also show ARTIST misses F1 layer quite often! Themens+ 2022, ARTIST Ionogram Autoscaling Confidence Scores: Best Practices, https://doi.org/10.46620/22-0001

**Technical corrections**

**Improve equations: typesetting, M(3000)F2 consistency**

I found maths in section 2 a little hard to follow due to:

- M(3000)F2 factor changing form a lot
- equations having inline fraction form

Great if you could tighten this up a little.

**M(3000)F2**

Not your fault this ratio's name has such a convoluted form. *Grr, not a single letter, no subscripts/superscripts, just one long set of inline chars! Guess a fine ionospheric tradition - hmF2 etc - and historic typesetting constraints.*

But this convoluted form doesn't help readers parse these equations, twig it's just a factor. Even harder as the form in this section changes quite a bit:

- Same thing: specific case: M(3000)F2, M3000F2
- Not the same thing: general case: M (looks like eqn 2 is for a general layer)

Realise that for sake of for sake of clarity/historic continuity, in specific case you'll have to use something like M(3000)F2. But there:

- can you use the M(3000)F2 form consistently please - not M3000F2
- great if there's anything you can do with typography to make it clearer this is just a factor!
  - `M_{3000}^{F2}` say?
  - ignore me if this won't wash!

**Inline equations**

Generally, there are various divisions going on in these equations. With the current inline fraction form it's much harder to follow them.

- for eqns 6 and 7 I ended up downloading Dudeney to check what was going on!

Could you change these to non-inline fractions please? Assuming you're using latex, with `\frac{numerator}{denominator}`

Few other minor things to correct:

- Eqn 6: +/- instead of $\pm$
- Eqn 7:
    - MF and $\Delta$M terms need to be better separated
    - $\Delta$M term has a ˘ symbol - checking Dudeney, this should be a minus sign

**Make various things more explicit**

There are various places where things were quite implicit, and I had to work ~hard to understand. Might be just me, but am flagging these so you're aware. I think you could usefully make these more explicit, to:

- help convey (what I think are!) your messages
- help less-expert readers follow subtler points (likely ~obvious to you / other experts, but could help this important work get a wider readership / impact)

I've put some suggestions for text which would likely have helped me in most places.

- Very interfering / word-smith-y - naturally please adjust / ignore as needed!

**Abstract**

**Make importance of this work clearer**   Clarify up-front that effect you've found is of same order as climate signal, so could mask latter / cause misattribution

```
@@ L1 @@
-Long-term change in
+Long-term reduction of ~20 km
```

```
@@ L9 @@
-±25km an altitude of 250km).
+±25 km at an altitude of 250 km, i.e. similar to the long-term climate signal change being sought).
```

Other: clarify this "shown" part of the point being made comes from previous work, not this one

```
@@ L12 @@
-have been
+have previously been
```

**Introduction**

**Effects**   I lost you a little in the discussion of the effects (L44 - L58). As this is ~crucial for this paper, a few overall ideas on making this section easier for non-experts - some specific ones below.

Overall points:

- Multifaceted nature / spatiotemporal scales of processes: Once I realised importance of this list, I had to draw out diagrams for myself to distinguish the processes, how they work, and outcomes. I don't think a diagram required - but would be helpful to flag up a few things readers should consider when reading list. Specifically outlining at outset sentence that (I think!) effects cover a range of timescales. And that a specific effect can encompass a ~complex set of in/direct processes, which can be local or remote (e.g. advected in).

- Clarity on spatiotemporal scales of effects: Optional, but consider whether here / in a separate paragraph / even a table below, for each effect, it's worth briefly discussing timescales (storm -> solar cycle(s)) and spatial extent - just in auroral zone, or more global due to transport?

- Process impact on height: Have you put these effects in order of ~expected impact on height? Ignore this if you can't / haven't. But if so, indicate explicitly!

- List of effects here: yours / someone else's / common knowledge? Can you be more explicit (e.g. "therefore we expect that changes") if to your knowledge you're the first authors to give this enumeration of effects. Helps readers sense-check "is this everything / is something missing?". Else a citation if someone else has listed these before.

- Citations: If ~easy to do, worth putting in an overall reference / specific references to each effect. Your point 2. got me fishing out Schunk & Nagy, and checking block diagram of energy flow in Ch9!

Specific points:

- effect 1: demarcate better the loss rate change covered above, from the height change introduced here

```
@@ L46 @@
-thermosphere which increases the height
+thermosphere, as described above. This also increases the height
```

- effect 2: be bit more specific on solar irradiance process

```
@@ L48 @@
-to the atmosphere, increasing
+to the upper atmosphere (thermosphere, ionosphere, mesosphere), increasing
@@ L49 @@
-of the atmosphere
+of the upper atmosphere, and hence raising of pressure levels
```

- effect 3: Make the composition point clearer. This is where I was glad I'd drawn out diagrams for myself! Helped me see that all other processes were changing the *height* of layer features (e.g. hmF2), whereas this process seemed not to. As written, ~implies this is only directly changing the *density* (e.g. nmF2). I *think* this is your intent - rest of article showing that such changes to (say) F1 densities can contaminate hmF2 reconstructions. I've suggested an extra sentence at end of intro to discuss this point, and tease out this difference. But possibly worth flagging this up already here? E.g. via tweak below? Of course, if the composition changes *also* directly affects true feature height (not just density), then alter tweak! In that case, recommend being as explicit as you can.

```
@@ L49 @@
-greater altitudes
+greater altitudes. Profile changes mainly affect densities,
+but this can indirectly affect height estimates.
```

- effect 5: clarify that this effect acts exclusively on long-term timescales?

- L64: might need rewording for clarity 4 and 5? Think only 5 is discussing contraction. And isn't your point in following sentence and section that most authors considered residual to be 5 alone? Whereas only a few - Jarvis, Bremer, Mikhailov - have pointed out that at least 4 can affect long-term trend too?

If so, something like following?

```
@@ L63 @@
-due to the contraction of the thermosphere assumed to be the dominant part of (4) and (5).
+widely assumed to be dominated by the contraction of the thermosphere (5).
```

**Composition and circulation, effect 3**   Recommend you also consider adding/moving following to this section:

- moving your composition+circulation discussion from ~L431 - L439 to this section
- appending a brief discussion about effect 3

Composition+circulation discussion: I think you could usefully move this to introduction, say just before the effects discussion Highly relevant, and usefully bulks out wind pattern change point, as well as better motivating the seasonal timescale. Currently the intro doesn't give much indication about why results consider seasonal timescale carefully - current timescales are really only ~ diurnal, space weather event, climate. This would help fill out spectrum! Having this here would also let you refer back to this in discussion of Scott et al ~L240.

Brief discussion on 3: I think worth adding something to end of intro:

- while it's very easy to juxtapose with the other effects
- to re/plant seed in readers' minds
- to help introduce following section
- to explicitly tie this in with wider thrust of paper

Something along lines of following - playing around with true/measured as appropriate!

"Absent from most previous literature has been consideration of the ionisation profile changes in (3). This effect may seem irrelevant, as it mainly affects the density of given features in the profile, rather than their true height. However, such density changes can have an indirect effect on estimates of this true height using ionosonde measurements, as discussed in the following section. And hence, as we show in this paper, can contaminate efforts to reconstruct true height estimates to extract the target climate signal from (5)."

**Measuring the ionosphere using ground-based radar**

**Scaling**   You briefly cover inverting (L84), but don't really cover scaling, despite referring to this later (L164). "Scaling" is quite jargon-y, so may not be accessible to all readers.

Once you got to B&D discussion of foF1, I had to do a quick refresher on ionosondes. I ended up needing to go quite deep into INAG links to understand what the issue was.

As you're (necessarily!) going into the weeds in this paper, possibly worth:

- adding a brief description of scaling process
- adding some citations to guides to both scaling and inverting ionosondes?

I ended up looking at the below for scaling. Might usefully supplement Rishbeth & Garriott (which I didn't check)?

Flagging resources like this helpful:

- for any readers wanting to do a deeper dive here
- esp to any you motivate to run with data-rescue inversion efforts!

~Old guides for scaling linked off https://www.sws.bom.gov.au/IPSHosted/INAG/ :

- Wakai, Ohyama and Koizumi, 1987, Manual of Ionogram Scaling, Radio Research Laboratory, Japan, 3rd edition, https://www.sws.bom.gov.au/IPSHosted/INAG/scaling/japanese_manual_v3.pdf
- Piggott & Rawer (eds.), 1972, U.R.S.I. handbook of ionogram interpretation and reduction, 2nd edition, UAG-23A, World Data Center A, https://www.sws.bom.gov.au/IPSHosted/INAG/uag_23a/UAG_23A_indexed.pdf (note high latitude supplement linked from top link too)

**Using ionosonde data to estimate hmF2**

**Motivation**   I think you could usefully make it even more explicit here what the motivation for using ionospheric measurements is, and hence why so many authors have tried. A few suggestions:

```
@@ L97 @@
-effects are altitude dependent which will lead to differing results. Searching
+effects in the upper atmosphere are altitude-dependent, which will lead to
+unhelpfully differing results, depending on the altitudes of the specific
+observations in each study.
+By contrast, searching

@@ L99 @@
-The longevity of ionospheric data series
+Furthermore, of the upper atmosphere regions, the ionosphere is unique as it can
+be observed remotely with relative ease. Due to the ionosphere's importance for
+long-distance radio communication, such observations have been made routinely
+since the early 20th century. The resulting longevity of ionospheric data series

@@ L101 @@
-can be extracted from the data, allowing for all other effects.
+can be extracted from the data, with all other potentially-confounding effects compensated for.
```

```
@@ L102 @@
```
```
-The first published analysis
+This potential has motivated similar re-analysis of ionospheric data,
+seeking evidence of climate-driven trends.
+The first published analysis
```

Be more explicit on what relevance is:

```
@@ L115 @@
```
```
-Kokubunji in Japan with
+Kokubunji in Japan (the same station examined here) with
```

**Ap and am**   L131: Ap index vs am index: Somewhere appropriate (here / a very short new section 2.3 / section 3.4) it would be good to have a brief discussion of this, making it explicit why you use am, rather than the Ap you flag has been used by most previous studies have used Ap!

Took me down a bit of a rabbit hole refreshing myself on am - had been a while! I think I eventually got my answer from Mienville & Berthelier, 1991, The K-derived planetary indices: Description and availability, https://doi.org/10.1029/91RG00994:

- end of their section 5 basically says that better sensitivity and distribution of Km network (cf Kp) makes am more suitable for statistical studies.

If this sort of reasoning governed your collective choice to use am, could you add a brief explicit sentence to this effect, citing Mienville & Berthelier.

- Or Lockwood+ 2019 which you cite already - looks like similar reasoning in there

Say this as if "accurate and painstaking" is required, favouring am over Ap seems very aligned: worth making your reasoning explicit, so others can follow suit!

**Confounding factors**   Make it more explicit that the tempting elision that "remaining trend == climate change" is dangerous? And loop back to your different effects from introduction? E.g.

```
@@ L134 @@
```
```
-Mikhailov and Marin
+When analysing such trends in residuals however, it is important not to assume
+any local environment changes which may underly this can be attributed to
+greenhouse forcing alone.
+The previously-discussed effects altering thermospheric composition (3)
+and wind patterns (4) risk being confounding factors, as they can potentially
+also lead to trends on long timescales.
+Mikhailov and Marin
```

**Clarify scope**   L160: worth being blunter, as think you've been nicely impartial in the actual analysis?

- if you adopt my "in section X" suggestion below, you could leave this as-is, and have summary outcome in there

```
@@ L160 @@
```
```
-to investigate the efficacy of
+is to demonstrate the potential pitfalls in
```

L161: current sentence ~implies the present work is looking into this reconciliation globally. Section 3.4 is a great start, and you've ~convinced me that global trends need to be reinterpreted in light of this paper, to see if reconciliation possible. But given this paper only examines Kokubunji, Chilton, and Stanley, this sentence may need watering down a little. E.g.

```
@@ L161 @@
```
```
-investigate the extent to which this can reconcile the difference
+demonstrate that such effects may have potential for reconciling the differences
```

**Add an outline**  L162: you lost me a little in section 2, trying to keep key bits in my working memory, and interpreting the detail you're exposing in light of this, while we're (necessarily!) knee-deep in complex empirical equations!

I think you could usefully insert a brief outline here at L62, summarising what you'll do & key findings. A la "In Section 2, we do XYZ and show, In Section 3 …".

Would let readers see where you're going, so help follow the thread of your argument better once in there - and flick back to this if they lose thread!

**Estimating hmF2 from empirical formulae**

**Existing parameters**  To make point clearer to those less familiar with ionosondes, something like following:

```
@@ L167 @@
-from existing scaled ionospheric parameters
+from existing standard ionospheric parameters (e.g. foF2, MUF).
+Determining these from an ionogram only required a scaling process (not inversion),
+hence these were routinely calculated from ionograms at the time of measurement.
```

**Minor fluctuations**  L204: I'm not following. Can you try to interpret what B&D / their Paul ref means? Is the point that:

- these are true F1 layers, but hard to pin down, as noisy?
- this is effectively noise (ionospheric variability / instrumental) getting mislabelled as F1?

Good if you could also tweak "more ubiquitous" on L207 to whatever is more appropriate:

- more ubiquituously-recorded
- less contentious

**Shape of profile**  L213: I don't follow - can you clarify what this "well above the limit" profile point means? I assume this supports B&D's argument that their approach means any F1 ionisation present can be ignored. But I'm unclear why. Especially if they were doing this off synthetic ionograms. Is this:

- a "how far are F2 and E separated along x-axis" measurement error type argument?(!)
- a physics-based argument on the retardation between E and F2 exceeding retardation from F1 features above this cutoff foF2/foE ratio?
- other?

**Limitations**  Can you give a physical insight? E.g.

```
@@ L217 @@
-X_E < 1.7, which
+X_E < 1.7, due to a more ionised E layer, which
```

Possibly worth adding foF1? Belabours point from previous sentence, but think OK, as it's a key finding of this paper that this is wrong!

```
@@ L219 @@
-field.
+field. And by definition this assumes F1-related effects are negligible.
```

**Which formula?**  L246: "refinements of the formula":

- a specific one? If yes, can you specify which one
- if general class, can you be clearer, e.g. "refinements of similar formulae"

**What is impact of foE = 0.4 MHz assumption?**  L279: can you indicate what expected impact of this fixed foE assumption is on hmF2? (realise elsewhere you say noise in results - I'm less sure! Here's why)

I assume this acts as a floor - the true foE would likely be lower, but this can't be measured, so it gets floored (ceilinged really!) to 0.4 MHz?

If so, what is this going to do to the "true" hmF2?

From a quick play with B&D eqn in Excel, and guessing some values:

- foE = 0.001 - 0.4
- foF2 = 5 - 7
- M(3000)F2 = 3 I think this floor will impose a very small < 5 km shift downwards on the value of hmF2 which would be calculated with the "true" foE. And this would affect night time values, so the ~300 km mark.

So should have a negligible effect on your plots & analysis? But would be good to have you check this, and to convey any such message.

**The Kokobunji ionosonde data**

**Clarify binning, ionosonde data retrieved**  L282: worth defining & justifying binning better. And enumerating all data retrieved.

Until I looked at code / thought hard about plots, I thought your monthly medians collapsed local time information. Also, it's only later that justification for medians appears. And that you also download F1 values.

How about something like following? Longer, but covers all data, and makes why & how for averaging more explicit up front.

"To estimate hmF2, scaled critical frequency parameters for the E, F1, and F2 layers (foE, foF1, and foF2), as well as the M(3000)F2 factor were downloaded. Monthly averages were then calculated for these data, to protect against outliers caused by short-lived space weather events that are not representative of the data on monthly timescales. Specifically, hourly monthly medians were used - medians calculated across corresponding hours (in local time) within a given month. For a given year, this yields 288 median values (bins of 12 months, and 24 local time hours). Such hourly monthly medians of foE, foF2 and M(3000)F2 were then used to estimate corresponding hourly values of hmF2, the true height of the F2 layer, using..."

**The Middle and Upper Atmosphere (MU) Radar**

**Clarify averaging**

```
@@ L318 @@
-height profiles were then averaged to
+height profiles were then averaged over the four antenna positions to

@@ L327 @@
-data, monthly means were
+data, corresponding hourly monthly means were

@@ L328 @@
-35 years of data used in this study, observations have
+35 years of ISR data used in this study, ISR observations have

@@ L329 @@
-10080 bins (monthly averages for each hour over 35 years).
+10,080 bins (monthly means, in bins for each local time hour and month, over 35 years).
```

I've suggested moving justification below to ionosonde section

```
@@ L329 @@
-parameters, this is to protect against outliers in ionospheric
-concentrations caused by short-lived space weather events that
```

```
-are not representative of the monthly data. For the ISR data,
+parameters, for the ISR data,
```

**Seasonal and diurnal comparison**

**Fig 1 interpretation**   L339: recommend flagging up in text a few extra things which jump out at me from Fig 1:

- The ISR data are noisier.
  - I assume because the campaign nature of ISR measurements means fewer underlying datapoints are available to constrain each hourly monthly mean datapoint?
- At solar min, the ISR data hit a floor at 180 km due to your sporadic E contamination procedure (suggest flag in figure too?)

**Fig 3 interpretation**

```
@@ L351 @@
-averaging over the 35 years
+averaging the aforementioned 10,080 bins over the year axis, for the 35 years
```

L352: you don't explain the white lines - I had to look at your code to double-check interpretation. Can you:

- add an explanation that these are the sza = 90, 100 limits used to isolate the previously-discussed day/terminator/night populations in Fig 2
- make these lines a bit thicker - the 90 one is particularly hard to see
- add lines like this to all subpanels, and all Figures like this
- consider a very brief discussion of results with respect to these?

Re latter, when I hand-draw similar lines on ISR, I see that for both ISR and ionosonde, dawn hmF2 hugs this sza line across all months. But that sza sn't constrain dusk hmF2, esp in winter months. I expect a ~easy Appleton anomaly explanation for diurnal, something else for seasonal. Would be good to have this flagged, and briefly explained by you!

```
@@ L353 @@
-peaks in hmF2 at
+peaks in night-time hmF2 at
```

Flag up midday summer peak

- you concentrate on this difference for ~rest of paper
- so good to flag up here that this is the target you'll concentrate on

```
@@ L354 @@
-a stronger peak
+an unexpected strong peak
```

```
@@ L355 @@
-source of this bias
+source of this midday summer bias
```

**Fig 4 interpretation**   L356: foF2/foF1: flag that this corresponds to xF from earlier eqn 4

L357: foF2/foF1: can you explicitly link back to discussion of eqn 4? Make point that B&D's range is nominally xF > 1.2 (note earlier query). But that here you're seeing it looks like it's more restrictive, that the 1.2 <= x.F <= 1.6 range is problematic?

**Correcting for foF1**   L360: can you be explicit on how you've done this?

Ideally break this out into an equation, as there are a couple of times later where you could usefully refer back to this.

Good to have this made explicit - currently I've got questions/guesses below.

I assume making the assumption that for

- `xF < 1.6`, `hmF2_iono_true ~= hmF2_ISR`
- `xF >= 1.6`, `hmF2_iono_true = hmF2_iono_raw`?

And so you filter ionosonde data to select 1st branch (`xF < 1.6`).

And for those data you apply linear model `hmF2_iono_true = (231*xF) + hmF2_iono_raw - 356`?

And you apply this to the binned hourly monthly median values (i.e. rather than to raw values, pre-binning)?

And for Fig 5, you do this after binning, but before collapsing over the year axis?

**Long-term bias**

L380: reword - this is unclear. My guess:

```
@@ L380 @@
-to correct affected daytime values within the hourly monthly median hmF2 values.
+to correct affected hmF2 values (where foF2/foF1 < 1.6)
```

- If you've followed my suggestion above to add an equation for your model, you could usefully refer to it here

```
@@ L381 @@
-When plotted
+When daytime values are plotted
```

**The impact of foF1 on the long-term drift in estimates of hmF2**

L411: if you followed my suggestion re model equation, here you could say "values, yielding similar models to eqn X, but for each year".

L413: add unit and make the interpretation clearer here!

```
@@ L413 @@
-of ±10% (±25km at 250)
+of up to ±10%(±25km at 250 km, i.e. a little larger than the
+~20 km decrease expected from climate change)
```

```
@@ L420 @@
-composition.
+composition, arising in turn from long-term changes in geomagnetic activity.
```

L440: can you make this point a little clearer?

I'm getting a bit lost on the temporal aspect - annual vs semi-annual. I think this is a red herring, not least as you're showing a clear difference in Fig 9 when you simply present annual averages!

I think the argument is ultimately altitudinal - what I'm (mis)understanding so far:

- near the poles, the neutral upwelling in summer disproportionately suppresses nmF2 (??? not sure why nmF1 less affected?)
- which pulls the yearly average of foF2/foF2 down?
- whereas further from poles, there isn't a strong altitudinal variation by season.

**Conclusions**

L496: latitude only? Does longitude need considering too cf circulation. Or is this ~purely meridional? Ask as ISRs which Bilitza 1979 used all have similar MLON (ie MLT) values.

**Refs**

Good if you can add missing URLs

- Bradley & Dudeney: https://doi.org/10.1016/0021-9169(73)90132-3
- Dudeney: https://nora.nerc.ac.uk/id/eprint/509197
- …

**Possible typos**

```
@@ L22 @@
-peak density of the F2 layer, foF2, would
+peak density of the F2 layer, nmF2, would
```

```
@@ L86 @@
-automatically identify and invert
+automatically scale and invert
```

L92: citepshangguan2019

L96: citepphilipona2018

L175: I think this should be inverted, no? M(3000)F2 = MUF/foF2 definition used by Elias 2017 (eqn 13), Japanese manual (p109), UAG (p23)

```
@@ L175 @@
-The ratio foF2/MUF
+The ratio MUF/foF2
```

L282: concentration vs critical frequency Think this should be critical frequencies, no, given foE and foF2, and formula definitions? So Note "concentration" is used quite often, and on a few other occasions is applied to a frequency, rather than a density/concentration. Realise these are ~equivalent (per eqn on line 68-69), but this doesn't help clarity. Could you do a search for "concentration", and try to rationalise this?

```
@@ L282 @@
-The peak electron concentrations of the E and F2 layers (foE and foF2)
+The critical frequencies of the E and F2 layers (foE and foF2)
```

L296: fragment?

```
@@ L296 @@
-integrated over. In the ionosphere, While the
+integrated over. While the
```

```
@@ L342 @@
-Daytime data (for which the solar zenith angle, sza > 100◦ )
+Daytime data (for which the solar zenith angle, sza < 90◦ )
```

```
@@ L342 @@
-twilight data (90◦ ≤sza≥ 100◦)
+twilight data (90◦ ≤ sza ≤ 100◦)
```

```
@@ L382 @@
-values (figure 7, the correction
+values (figure 7), the correction
```

```
@@ L383 @@
-the remaining offset is not 1:1.
+the remaining gradient is not 1:1.
```

L388: ionosonde values being "generally still lower" and "underestimate of the F2 layer height". Err, are you sure?

- Lower in some bins - equinox night
- But not in others - midsummer night
- cf Fig 2: the two clumps I've flagged above/below trendline

- No idea about "general" (average over all months), and how much there's in it!
- Think this is where my requested difference subpanel could help!

L469-L471: Comparing the before/after gradient values, I think you're currently a factor 10x too large in your reported improvements! Hence following changes

```
@@ L469 @@
-resulted in an improved relationship
+resulted in a slightly improved relationship

@@ L470 @@
-the revised gradient for daytime hmF2 increases by 0.3 to 0.89±0.01
+the revised gradient for daytime hmF2 increases by 0.03 to 0.89±0.01

@@ L471 @@
-the equivalent after correcting for the presence of foF1 increases by 0.3 to 0.92±0.01
+the equivalent after correcting for the presence of foF1 also increases by 0.03 to 0.92±0.01
```

Fig 2 caption

```
-gradient of 0.71pm0.01
+gradient of 0.71±0.01
```

Fig 4 caption

```
-(gradient 231±21, offset −356±31km
+(gradient 231±21, offset −356±31 km)
```

Fig 5 caption

```
-by the ISR (figure 5 lower panel
-).
+by the ISR (figure 4, lower panel).
```

Fig 6 caption

```
-(as identified in 4. It can
+(as identified in Figure 4). It can
```

---

## Author Comment (AC1)

***Review report of the manuscript entitled, "Calibrating estimates of ionospheric long-term change" authored by Christopher Scott, Matthew Wild, Luke Barnard, Bingkun Yu, Tatsuhiro Yokoyama, Michael Lockwood, Cathryn Mitchel, John Coxon, and Andrew Kavanagh.***

*Investigations on the ionospheric trends is an important area of research. Researchers have been using hmF2 and foF2 from ionograms to address this issue. The authors of this work dwell on the corrections for possible uncertainties that may be introduced in the use of these parameters for assessing the long term trends given the fact that there is ambiguity in the rates of changes in the hmF2 in different studies reported in the literature. Thus, the current manuscript attempt to investigate the efficacy of deriving long-term ionospheric trends in hmF2 using empirical formulae and to investigate the extent to which this can reconcile the difference in trends derived from the global network of ionospheric monitoring stations.*

*Thus, the authors suggest using foF2/foF1 ratio to qualify the values of hmF2 for a mid-latitude location. For that they use common data from the MU radar (from 1986-2020) and the ionosonde at Kokubunji (1957 – 2020) to arrive at the frequency rations of F2 to F1 regions to "calibrate" the hmF2 values from ionosonde.*

*This is an interesting effort which needs to be encouraged. There are, however, several loose ends, incomplete information, which need to be addressed before this article can be considered further.*

Many thanks for your consideration of our article and for the helpful comments. We address each of these in turn below.

1. *The formation of F1 layer has seasonal dependence. So, in order to provide an appropriate correction, which is more "accurate", why not use the data from only that season? It will reduce the number of points, however, the values considered for such a season over 35 years may provide a different (better?) correction?*

Yes, the presence of F1 has a seasonal dependence and this does indeed introduce a seasonal bias into the equation. The prominence of the F1 layer in ionograms is controlled by the underlying thermospheric composition which, in addition to the seasonal variation, also undergoes long-term changes driven by geomagnetic activity. The main focus of the current article is to demonstrate that there are long-term biases in hmF2 introduced by the formula and that these will differ by location. Cutting the data by season will not remove these long-term biases although some times and months will be less affected than others.

In response to reviewer #2, we also investigated the influence of the foF2/foE ratio and this too was found to have a similar bias where the ratio fell below ~2.5. This resulted in a similar seasonal bias as for foF2/foF1 since the ratios are both driven below their bias thresholds by the suppression of foF2 during the summer months.

We agree that is important to highlight the presence of a seasonal bias and have added to this section, including plots of how the foF2/foF1 and foF2/foE ratios vary with time of day and season.

2. *Line 20: How much is the role of radiative cooling by $CO_2$ in the thermospheric cooling?*

The work by Roble & Dickinson referred to here investigates the impact of simultaneous doubling of CH4 and CO2 concentrations in the lower atmosphere. As such, the work does not separate out the effects but does note that

'The model uses the complete $CO_2$ radiation code of Dickinson [1984] which includes both LTE and non-LTE $CO_2$ radiational processes and considers hot bands, isotopic bands, Voigt line shapes and radiative transfer as significant for mesospheric cooling. In the lower thermosphere $CO_2$ cooling rates are proportional to the product of atomic oxygen concentrations and the poorly known rate of energy exchange between O and $CO_2$. For our calculations we assume a quenching rate of 10-12 cm3 $s^{-1}$.'

They later add that 'The cooling is caused primarily by enhanced $CO_2$ emissions'

We have added the latter statement to our introduction for clarity.

'Roble & Dickinson (1989) note that the modelled cooling is caused primarily by enhanced $CO_2$ emissions.'

3. *Line nos: 92 and 96, correct the citations.*

Thank you for noticing this. Citation repaired.

4. *Is there any study that describe the relationship between stratospheric cooling with ionosphere? If it is there then provide the reference.*

We do not know of any such study. A search of the literature did not reveal any papers of relevance to this specific area.

5. *The title of section 1.2 can be different as it does not match with the text.*

Yes, this is a valid point. We have changed the section name to 'Investigating long-term trends in the height of the upper atmosphere' which we feel is sufficiently generic that it now covers the contents of the section

6. *Line 126: define M3000F2.*

Thank you for noticing this. As M(3000)F2 is discussed in detail in the next section, we have referred the reader to the subsequent section at this point in the manuscript.

7. *In equation 2, what is M?*

Thanks for spotting this. M represents the M(3000) factor for a general layer. We have added the following text to make this clear;

'where M represents the M(3000) factor of a general layer.'

8. *In eq. 5, define $M_o$.*

Thank you for spotting this. We have added the following text to explain this term;

'where $M_o$ = MUF/foF2.'

9. *Line 236: Insert 'and' before $\Delta M$.*

Done as suggested.

10. *Equation 4 has been used for the estimation of hmF2. According to Bradley and Dudeney, (1973) this relation is not valid for the values of $x_E < 1.7$. Then, how the estimation has been carried out? In addition, what about those ionograms when the signal of E-layer is insignificant even in daytime?*

Our analysis does not calculate hmF2 values where $x_E < 1.7$ irrespective of time of day.

We have added the following text in the section describing the ionosonde data analysis to state this more fully;

'Following the analysis of Jarvis et al (1998) it was initially assumed that foE = 0.4 MHz at night. Where $x_E < 1.7$, no value of hmF2 was calculated.'

11. *Mention the R square values in Figure-2.*

We have added the R-square value (for each of the sub-panels in the revised figure) to the figure caption.

12. *Correct the caption of Figure 2, such as, write +/- at the place of pm, and change the symbol by < for the case of SZA values.*

Done, as suggested.

13. *In Fig. 1, there is a significant spread in the hmF2 values derived from ISR values which is not present in ionosonde derived values. Can author comment on this?*

This spread in ISR data is due to the fact that the monthly means for ISR data contain fewer points than the monthly median ionosonde data. This point was also raised by reviewer #2.

We have added the following text to the caption of figure 1;

'It can be seen that the ISR data are noisier, due to there being relatively fewer data points from which the mean values are calculated'

We have also added a subpanel to figure 3 (and subsequent comparisons) to show the number of years contributing to each value when the ISR data are averaged.

14. *Line 341 – 342: daytime is when SZA < $90^0$. In these sentences it is mentioned incorrectly.*

Thank you. We have corrected this.

15. *5 is incomplete. The text points to the ISR data as well (388 – 389), but it does not exist in the Figure. The figure caption too alludes to this. ISR hmF2 also to be plotted in this figure, similar to the format in Fig. 3.*

Thank you. This figure has been updated to match figure 3 in format.

16. *Lines 411-412: Explain how the percentage error is reconstructed using gradient and offset.*

The text has been changed to describe this more explicity;

'The resulting gradient and offset of each fit were used to derive a modelled height for an arbitrary ISR height of 250 km. The differences between these two values were used to reconstruct the percentage error in hmF2.'

*17. Lines 438- 439: This statement may be substantiated by an appropriate reference.*

We have added a reference to Millward et al, Ionospheric F2 layer seasonal and semi-annual variations, J. Geophys. Res., 101, 5149-5156, 1996.

*18. Lines 462 – 463: In the estimation of range bias, why inospheric profile is integrated only up to hmF2 as topside ionosphere also affects the radio wave propagation?*

The range bias is introduced by the radio wave propagating through an ionised medium. Since it is the bias in the height of the F2 peak we are interested in, the signal delay needs to be integrated along the path of the radio wave between the ground and the F2 peak, accounting for the upwards and downwards path of the signal.

In order to make this point more explicit, we have added the following text to this section;

'Since it is the bias in the height of the F2 peak we are interested in, the signal delay needs to be integrated along the path of the radio wave between the ground and the F2 peak, accounting for the upwards and downwards path of the signal.'

*19. Lines 467-468: can one carryout a simple subtraction like this?*

Yes. The ISR uses time of flight of the radio signal to estimate the height of the F2 layer, assuming that there is no signal delay introduced by interaction with the underlying ionisation. By modelling the delay introduced to the time of flight by the signal interacting with the underlying ionisation and comparing this with the known (modelled) height of the layer, an estimate can be made of this bias over a range of diurnal, seasonal and solar cycle conditions. While this is not an absolute measure of the delay occurring in the real-world data, it is sufficient to estimate the relative change in bias across diurnal, seasonal and solar cycle conditions and to demonstrate that this effect does not affect the conclusions drawn from the analysis using uncorrected ISR data.

We have added the following text to make this clear;

'By modelling the delay introduced to the time of flight by the signal interacting with the underlying ionisation, and comparing this with the known (modelled) height of the layer, an estimate can be made of this bias over a range of diurnal, seasonal and solar cycle conditions. While this is not an absolute measure of the delay occurring in the real-world data, it is sufficient to estimate the relative change in bias across a representative range of conditions.'

*20. One can expect that the neutral wind changes with the magnitude of geomagnetic activity. Can author comment on this aspect?*

This is a very valid point. While we discuss this in the introduction, we have added the following text to the discussion section in order to emphasise this point.

As noted in the introduction, geomagnetic activity may also induce change in global thermospheric circulation with changes in the meridional wind modulating the height of F2-layer. Titheridge (1995) reviews the magnitude of these effects. A poleward wind would move ionisation to lower altitudes, where the loss rate is higher. This would lead to a

decrease in the peak F2 ionisation. Under such circumstances, the foF2/foE ratio could potentially become sufficiently small that, in addition to the genuine change in layer height, the empirical formula would start to underestimate hmF2. More work is needed to deconvolve the relative magnitude of these effects but whether driven by changes in the wind-field or local changes in composition, geomagnetic activity can lead to long-term bias in estimates of hmF2.

21. According to these results, the composition effect that varies with geomagnetic activities, should be taken care of for the long-term study. It is mentioned at the line number 119 that regression model is not significant alone to remove the effect of geomagnetic activities. But, in Fig. 8, am index shows clear proportional behaviour with composition proxy. Then, how do author think that correcting the compositional variations in the hmF2 variation will give better result? Could the authors compare their results with the study of Xu et al., (2004) to validate/assess their method?

We agree that we ought to say more about the comparison between the results of Xu et al (2004) and our analysis. However a direct comparison is not possible since both analyses use different version of the empirical formula used to estimate hmF2. We have added the following qualitative comparison;

When conducting their analysis of the Kokubunji data, Xu et al (2004) used the formula of Bilitza et al (1979). While a direct comparison cannot be made with the current analysis, the variability in long-term trends observed by Xu et al (2004) (a difference between long-term trends in noon and midnight hmF2, with the seasonal variation at these two times being opposite to each other) is consistent with a maximum bias occurring around noon in the summer months. Xu et al (2004) also conclude that geomagnetic activity was not significant in the regression model used to remove the effects of geomagnetic and solar variability. Nevertheless, our analysis indicates that variations in the bias of hmF2 estimates is likely driven by geomagnetic activity.

---

## Author Comment (AC2)

Response to reviewer #2

*Scott et al provide a compelling case that previous efforts to discern climate-driven lowering of the ionosphere have been hampered by a reliance on semi-empirical equations which incorrectly assume any F1 layer present has an insignificant effect.*

*They show F1 layers can cause biases in height estimates which risk masking any climate trends, or causing misattributions.*

*They attribute this to thermospheric composition effects, driven by geomagnetic activity.*

*The manuscript is of very high quality, and only needs a few revisions - should be minor in effort - to exclude some minor remaining doubts about potentially confounding effects, and make their case unarguable. As well as to make some of their reasoning a little more explicit, improving their work's accessibility to a wider audience. A more detailed breakdown of general and specific comments, and technical corrections is attached.*

*Slightly weaker however: foF1 <-> composition link*
*• Physical rationale (introduced ~L364, invoked ~L405) linking presence of foF1 to composition changes (maybe a priori knowledge I lack - make this more explicit if so!)*

Thank you for this insight. We agree that adding some additional information about the influence of thermospheric composition on the presence of the F1 layer would make the paper more stand-alone.

We propose adding the following text to section 1.1;

"The shape of the electron concentration profile is influenced by the composition of the neutral thermosphere, through the loss rate of ionisation. At equilibrium, the electron concentration, N, can be expressed as (Rishbeth & Garriott, 1969);

$$N = \left(\frac{q}{2\beta}\right)\left[1 + \left(1 + \frac{4\beta^2}{\alpha q}\right)^{\frac{1}{2}}\right]$$

where q is the ion production rate (s$^{-1}$), α is the loss rate of molecular ions (s$^{-1}$) and β is the loss rate of atomic ions (s$^{-1}$).
Between the E and F2 layers, there is often an additional layer, the F1 layer, visible on ionograms. This layer forms between 160-200km where the loss rate of ionisation transitions from being dominated by the loss of molecular ions at the lower altitudes in the E layer where;

$$N = N_\alpha = \left(\frac{q}{\alpha}\right)^{\frac{1}{2}}$$

to a loss process dominated by atomic ions at the higher altitudes of the F2 layer where;

$$N = N_\beta = \frac{q}{\beta}$$

Ratcliffe (1956) first suggested that the transition between molecular to atomic loss processes may account for the splitting of the F region into two distinct layers, with the parameter β$^2$/αq determining the shape of the electron distribution with height. Denoting this quantity as G at the level of peak production,

$$G = \frac{\beta^2}{\alpha q} = \frac{N_\alpha^2}{N_\beta^2}$$

Figure 28 in Rishbeth & Garriott, 1969 (reproduced as figure 1 in Scott et al, 2019) presents the vertical profile of electron concentration for a range of values of G. When G is small (<1), the presence of the F1 layer is barely visible in the profile, becoming a much more pronounced inflexion for larger values of G. In this way, the prominence (and thus visibility of the F1 layer on an ionogram) is a function of the ratio of molecular and ion loss rates, with the ion composition itself being controlled by the composition of the neutral thermosphere. F1 layers are a daytime phenomenon, prominent during summer months.

While the presence of F1 layers in historic ionograms is tabulated (in terms of peak frequency foF1 and virtual height, h'F1), these do not record the prominence of the layer, from which a more detailed understanding of the ionospheric and thermospheric composition profiles could be gleaned."

*Figure from R&G presented for reference in this review:*

[Figure]

• *Qualitative support from composition proxy in 8b showing similar trend to 8a Am not familiar with this Wright & Conkright proxy, and I note comment CC1 queries it…*

In addition to the text above which introduces this concept, we have added the following text to make it clear where the idea of using this as a proxy for compositional change came from.

"Similarly, changes in neutral composition affect the peak electron concentration of the F2 layer. At noon, where the F2-layer approaches a steady-state condition, production and loss are in equilibrium. If the loss rate of ionisation is enhanced by the presence of molecular ions, the peak electron concentration of the layer will be reduced. Since the electron concentration is proportional to the peak frequency squared, comparison can be made

between measurements at similar dates but different times in the solar cycle by scaling these values by the ion production rate, q. A good proxy for q is the F10.7cm solar radio flux. Thus

$$N \propto \frac{f^2}{q} \equiv \frac{foF2^2}{F10.7}$$

Using noon values of foF2 to track qualitative changes in thermospheric composition was suggested by Rishbeth et al (1995) with Wright and Conkwright (2001) comparing the efficacy of this simple index (their FFD index, averaged over 5 hours around noon) with other more complex indices derived from the rate of change of the ionosphere at sunrise."

*• Discussion of annual/semi-annual circulation-induced changes to composition in L430-L447 seems to suggest foF2 is more affected than foF1: L441 "foF2 is suppressed… when foF1 is at its peak"*
*So it would be good if the foF1 <-> composition link rationale could be tightened up a little*

Hopefully the above description has given a greater insight into the role of thermospheric composition in splitting of the F-layer and the modulation of the peak density seen in the F2 layer. The occurrence of an F1 layer is most prominent around noon in the summer months while (for Slough/Chilton at least) the F2 peak is weakest during the summer months when the concentration of molecular species at F2 altitudes is greatest.

**Attribution of effect to F1: exclude E as a confounder**
*While the F1 evidence seems good, I'm a bit concerned that the apparent "smoking gun" in the derived parameters means you may not have looked at the raw data, to see if there are other potential causes for the patterns you've discerned.*

*Specific areas of concern:*
*• the domain of validity for B&D model is for xE (= foF2/foE) > 1.7*
*• both xF and xE share a common numerator, so I'd expect them to be somewhat correlated*
*Data-wise:*
*• What happens if you create an equivalent to Fig 4, but order the hmF2 bias by xE?*
*– Is there a similar "break" partway along?*
*– Shouldn't be, if the B&D model for hmF2(xE) is decent along its validity range…*
*– but worth double-checking, in case of any nasty non-linearities?*
*• Likewise does any of your data have xE <= 1.7 (i.e. outside of B&D validity range)?*
*– What happens to bias then?*
*Physically:*
*• Might composition changes affecting foF1 also affect foE?*
*I think it's worth being certain that the attribution to foF1 contamination is rock-solid.*
*Not least as a foE effect might be good news: given foE scaled more frequently than foF1, if any effect can be characterised, might give more data for looking at trends globally using already-manually scaled parameters, without data-rescue-based inversions?*
*See my suggestion below on a few supplementary material plots for probing foF1 attribution.*
*• there may be other ways of excluding potential confounders of foF1 attribution*

This has been an incredibly useful insight, thank you.
We had not tested to see if there was any bias in the foF2/foE ratio, having assumed that this had been considered when formulating the initial empirical relation. As has been stated in the literature however, the empirical relations are honed for a specific station and may therefore not hold for other locations or times.

Repeating the analysis looking at the foF2/foE ratio revealed a very similar (and larger) bias for values below around 2.5 (values below 1.7 are automatically excluded as the form of the equation means that it fails for values below this limit). A robust fit to these data gives a best fit line of y=80 (± 9) + -194 (± 19)

[Figure]

Fitting this bias and correcting for it results in a gradient of 0.97 ± 0.01 when comparing the ISR and ionosonde-derived hmF2 values).

[Figure]

The foF2/foE ratio is also affected by changes in composition through modulation of the peak F-region density, which in turn affects the value of foF2. Since foF2 is suppressed by compositional changes during the day in summer, this effect is coincident with the presence of F1 layer ionisation which may also be contributing to the bias.

[Figure]

Such a correction introduces a significant amount of noise however which completely masks any subsequent effect due to the presence of foF1. As the referee points out, the two

parameters are highly corelated and so, while a comprehensive correction cannot easily be applied to the ionosonde-derived estimates of hmF2 (and such an effort, even if successful, would only be relevant to this geographical location) our conclusion remains that long-term changes in composition lead to biases in the estimate of hmF2 when using the empirical formula. Furthermore, these biases will be different for different locations, as previously demonstrated for Slough and Stanley

We propose addressing this by treating the two effects separately in the paper and discussing the subsequent impact on estimates of long-term change in hmF2.

**Enhance existing plots**

*On ISR subpanel, worth adding a horizontal line / note in caption that there's a 180 km threshold getting imposed*
*to avoid sporadic E layer?*
*• This to help readers spot this in subsequent plots*
*• And consider this when ISR is acting as source of truth*
*Add a 3rd subpanel below current two, plotting delta_hmF2 = hmF2_ISR - hmF2_ionosonde*

Figure 1 amended, as suggested

[Figure]

***Figure 2: break out the day/night/terminator populations***
*Recommend changing this to a 2x2 subpanel plot*
> *• keep current plot as an "all populations" subpanel*
> *• but use other 3 subpanels to plot each day/night/terminator population separately*

*Current plot is good for general intercomparison of the three populations, and reasoning about the "all" fit. But overplotting means it's hard to see & assess any detailed features in each population.*
*E.g. there seem to be ~two distinct clumps in the nighttime population, above/below the main trendline, at low/high ISR hmF2. But as everything is overplotted, it's hard to assess if there's anything similar in the twilight/ day populations, e.g.*
> *• do they have similar clumps?*
> *• if so, do their "upper" clumps also contain ISR hmF2s which hit the 180 km floor?*

*Likewise hard to see scatter cores - is anything interesting lurking in there?*

*Overall makes it harder to assess validity of the linear fits. These look OK, but good to reassure reader before these get used (Fig 5, 8a)*

Thanks for this very useful suggestion. We have revised figure 2 as suggested;

[Figure]

(we have changed the yellow to magenta as the yellow was less clear when these data were plotted on their own). We also have plotted best fit lines for these populations for comparison. In addition to the scatter, the assumption that foE=0.4 MHz when below the detection threshold of the ionosonde introduces a bias in estimates of hmF2 via the foF2/foE ratio. The twilight population has less scatter and a slightly better gradient of 0.56, this is consistent with values of foE that are larger than 0.4 MHz but are nonetheless below the detection threshold of the ionosonde (typically ~ 1 MHz).

***Figure 3: plot the delta_hmF2; bin count / 180 km count***
*These plots are great for probing the seasonal and local time behaviour, but I think they could be made even more revealing.*
*Again I'd recommend inflating this to a 2x2 plot. Keep the current two, but add:*
*• 3: subpanel of delta_hmF2*
*• 4: subpanel of number of year bins [0 - 35] which have contributed to the ISR mean value in current lower panel*
*3: to help readers spot the differences, and quantify these. Very hard to compare colours by eye! Might also reveal more subtle differences.*
*4: to check there's no concerns about quality of ISR averages in any bins I hope that the 48% occupancy you report for the 35 years means there's ~16-17 counts per ISR month-localtime bin. But I think worth exposing bin count to be sure:*
*• as you're treating ISR as ~truth-ier than ionosonde*
*• as ISR is a campaign-based instrument, and campaigns might clump in season-localtime space!*
*• as this ISR climatology seems to be showing a few interesting features (e.g. dips in summertime night hmF2)*
*- worth checking these are robust!*
*– are these summertime night dips likely to be real?*
*– if not, could they be contributing to bias?*

Again, another useful suggestion, thank you.
We have modified the plot as suggested;

[Figure]

The new difference plot shows that the summertime bias is large despite there being above average numbers of ISR data available in these months from which to calculate mean values. In addition, we note that the minimum number of ISR data available in a bin was 14, the maximum number was 31 and the median was 25.

*alternative 4: fraction of bin with ISR hmF2 = 180 km If the ISR bins all turn out to have OK-ish occupancy, flat enough that you don't think there's an issue, I think you can simply report this in text, and use the subpanel to look at other minor concern instead:*
*• The 180 km threshold on ISR will skew the bin mean value*
*• and in worst case (all "true" ISR hmF2 values in bin across years < 180 km) set it to 180*
*This might be affecting some of the blue-er bins. You could quantify this by calculating what fraction of the counts in each bin have values == 180 km.*

We did not consider this option since if the maximum electron density was found to be either the minimum or maximum height within the window considered, the value was set to a null value and not included in subsequent calculations. We will make this clearer in the revised text.

**Figure 5: plot delta_hmF2**
*The daytime summer peak has clearly gone, but it's hard to compare colours here to Fig 3b, to get anything more out of this.*
*Again, recommend adding a subpanel for delta_hmF2 - help readers compare with the previous delta_hmF2, see if there's anything more here.*
*Will also support your discussion of the nighttime hmF2 differences remaining*
*• in L365*
*• in L388 - let you be a bit more nuanced - your "generally lower" point may apply across all months, but doesn't apply in summer months.*

We have added the additional subplot, as suggested;

[Figure]

This indeed reveals that the corrected ionosonde-derived estimates of hmF2 in summer daytime are now closer to those observed by the ISR.

So, what is the cause?

We plotted the ratios as a function of season and time of day.

[Figure]

[Figure]

This shows that both foF2/foE and foF2/foF1 fall below their critical values in the summer daytime. This is consistent with high values of foE and foF1 (whose peak densities are strongly controlled by peak ionisation rates occurring during times of depleted noon foF2 densities due to increased thermospheric molecular content leading to an enhanced loss rate.

Changes in thermospheric composition are therefore leading to a breakdown in the reliability of the empirical formula through the suppression of foF2 and/or the presence of foF1. Since foE and foF1 are highly correlated, it is challenging to deconvolve their relative contributions to the bias seen in the empirical formula, with the correction of the foF2/foE bias also correcting for any bias resulting from the presence of foF1.

As with the foF2/foF1 ratio, long-term changes in thermospheric composition will therefore introduce a long-term bias into any estimates of hmF2. That. We have therefore expanded the discussion section to include how the foF2/foE ratio changes with time.

Uncertainty in ionosonde-derived hmF2

sqrt((($(-(-355 / ( x - 1.4) ^ 2 * M ^ ((2.5 *  x - 3) ^ (-2.35) - 1.6)) - 5.875 * (2.5 *  x - 3) ^ (-2.35 - 1) * (1890 - 355 / ( x - 1.4)) *  M ^ ((2.5 *  x - 3) ^ (-2.35) - 1.6) *  ln( M)) *  \delta x) ^ 2 + ( M ^ ((2.5 *  x - 3) ^ (-2.35) - 1.6 + -1) * (1890 - 355 / ( x - 1.4)) * ((2.5 *  x - 3) ^ (-2.35) - 1.6) *  \delta M) ^ 2)

Revised figures show, as anticipated, smaller uncertainties in ionosonde-derived hmF2 values.

**Check unexpected striations in time series scatter plots**
*Can you double-check your plotting code in Fig 6 (likely applies to Fig 1 too)? And if all is coded OK, explain how the striations in panel a are arising?*

I have checked the code here and think the plots are correct. The striations in figure 6 and 1 are due to the annual ionospheric variability combined with the fact that foF1 is primarily a daytime summer phenomenon. As a result the F2/F1 ratio displays the fraction of the annual variation for which both foF2 and foF1 values are available.

**Flag up more clearly where your findings differ from others'**
*There are a few places throughout intro sections where you flag up ~detailed points from previous literature - often*
*I think as these run counter to what you find.*
*But you don't refer back to these when you present your results / conclusions.*
*Upshot is that it's not easy for the reader to spot these - requires multiple passes.*
*I'd recommend flagging any such disagreements up more explicitly somewhere: either in results as you go, or*
*aggregating somewhere just before/in conclusions.*
*Ones I've spotted (may well be more!)*

Thanks for this general comment, we have endeavoured to be more explicit as to how our results differ from others throughout the paper.

• *L119: Xu not finding significant geomag effect (cf your Fig 8)*

We do not have an explanation for this discrepancy but we note that their use of a different empirical model may result in this difference. We have added the following text;

'This could be attributed to their use of a different empirical model.'

*• L215: B&D claiming F1 insignificant. OK, "claim", and it's ~clear that your whole paper is contesting this general point! But if B&D / derivatives get used as much as L246 implies, I think worth being blunt somewhere*
*"B&D's claim is wrong"!*

Now that the paper includes a bias in the foF2/foE ratio as well as foF2/foF1, it could be that the presence of foF1 is, in itself not the cause of the bias, but instead, an indication that changes in thermospheric composition (which also cause a reduction in foF2 and therefore of the two ratios). We have therefore opted not to make a stronger statement here.

==*• L487: the juxtaposition is a little too subtle for me - think you could make if clearer that that you're ~disagreeing with Jarvis. How about something like following?*==

*@@ L488 @@*
*-explain this. Our results indicate that*
*+explain this. Our results indicate such mechanisms do not need to be invoked, rather that…*

Changed as suggested.

**Minor points**
*L212: "They noted that xE > 1.7 is equivalent to about xF ≈ 1.2"*
*• Where are you getting 1.2 from? Looking at B&D I can't see this.*
*Closest match their sentence below, but I don't think this yields this equivalence. "(In the limit for x = 1.7, f1 = foF2 and we have an entirely linear Fl/F2-layer.)"*

This statement was in fact taken from Dudeney (1974). We have corrected the citation in the text.

**Technical corrections**
**Improve equations: typesetting, M(3000)F2 consistency**
*I found maths in section 2 a little hard to follow due to:*
*• M(3000)F2 factor changing form a lot*
*• equations having inline fraction form*
*Great if you could tighten this up a little.*
**M(3000)F2**
*Not your fault this ratio's name has such a convoluted form. Grr, not a single letter, no subscripts/superscripts,*
*just one long set of inline chars! Guess a fine ionospheric tradition - hmF2 etc - and historic typesetting constraints.*
*But this convoluted form doesn't help readers parse these equations, twig it's just a factor. Even harder as the*
*form in this section changes quite a bit:*
*• Same thing: specific case: M(3000)F2, M3000F2*
*• Not the same thing: general case: M (looks like eqn 2 is for a general layer)*
*Realise that for sake of for sake of clarity/historic continuity, in specific case you'll have to use something like*

*M(3000)F2. But there:*
*• can you use the M(3000)F2 form consistently please - not M3000F2*
*• great if there's anything you can do with typography to make it clearer this is just a factor!*
*– M_{3000}^{F2} say?*
*– ignore me if this won't wash!*

Thanks for pointing this out. I have unified the text to be M(3000)F2 everywhere for the time-being and will raise this with the type-setters.

*Generally, there are various divisions going on in these equations. With the current inline fraction form it's much harder to follow them.*
*Could you change these to non-inline fractions please?*

Another good suggestion. We have changed the form of the equations to use the \frac{}{} form as suggested.

*Few other minor things to correct:*
*• Eqn 6: +/- instead of ±*

Done.

*• Eqn 7:*
*– MF and ΔM terms need to be better separated*

Done, as suggested.

*– ΔM term has a ˘ symbol - checking Dudeney, this should be a minus sign*

Done

**Make various things more explicit**
*There are various places where things were quite implicit, and I had to work ~hard to understand. Might be just me, but am flagging these so you're aware. I think you could usefully make these more explicit, to:*
*• help convey (what I think are!) your messages*
*• help less-expert readers follow subtler points (likely ~obvious to you / other experts, but could help this important work get a wider readership / impact)*
*I've put some suggestions for text which would likely have helped me in most places.*
*• Very interfering / word-smith-y - naturally please adjust / ignore as needed!*

**Abstract**
**Make importance of this work clearer** *Clarify up-front that effect you've found is of same order as climate signal, so could mask latter / cause misattribution*
*@@ L1 @@*
*-Long-term change in*
*+Long-term reduction of ~20 km*

Changed as suggested, thank you.

*@@ L9 @@*

*-±25km an altitude of 250km).*
*+±25 km at an altitude of 250 km, i.e. similar to the long-term climate signal change being sought).*

Changed to 'These are of similar magnitude to layer height changes anticipated as a result of climate change.'

*Other: clarify this "shown" part of the point being made comes from previous work, not this one*
*@@ L12 @@*
*-have been*
*+have previously been*

Changed as suggested, thank you.
**Introduction**
**Effects** *I lost you a little in the discussion of the effects (L44 - L58). As this is ~crucial for this paper, a few overall ideas on making this section easier for non-experts - some specific ones below.*

*Overall points:*
*• Multifaceted nature / spatiotemporal scales of processes: Once I realised importance of this list, I had to draw out diagrams for myself to distinguish the processes, how they work, and outcomes. I don't think a diagram required - but would be helpful to flag up a few things readers should consider when reading list. Specifically outlining at outset sentence that (I think!) effects cover a range of timescales. And that a specific effect can encompass a ~complex set of in/direct processes, which can be local or remote (e.g. advected in).*

We have changed the introduction to this list to state;
'Above a given ionospheric monitoring station therefore, changes to the height of the ionosphere are a superposition of several different effects, occurring across a wide range of temporal scales from long-term ($\sim$ multi-decadal) through solar cycle ($\sim$ decadal) to individual space weather events $\sim$ hours. These can be grouped into several interrelated categories;'

And added a reference to a paper discussing such effects;

'Detailed discussion of contributions to ionospheric long-term change have be presented by Rishbeth and Clilverd (1999)'.

(Henry Rishbeth, Mark Clilverd, Long-term change in the upper atmosphere, *Astronomy & Geophysics*, Volume 40, Issue 3, June 1999, Pages 3.26
3.28, https://doi.org/10.1093/astrog/40.3.3.26)

*• Clarity on spatiotemporal scales of effects: Optional, but consider whether here / in a separate paragraph / even a table below, for each effect, it's worth briefly discussing timescales (storm -> solar cycle(s)) and spatial extent - just in auroral zone, or more global due to transport?*

We hope that including the above reference will provide the reader with a source of such information without having to go into too much detail that is not central to the narrative of this paper.

*• Process impact on height: Have you put these effects in order of ~expected impact on height? Ignore this if you can't / haven't. But if so, indicate explicitly!*

No, this list is not in any particular order in terms of expected impacts on ionospheric height. Again, we would rather not deviate from the central narrative to discuss this so have taken you up on your offer to ignore this.

*• List of effects here: yours / someone else's / common knowledge? Can you be more explicit (e.g. "therefore we expect that changes") if to your knowledge you're the first authors to give this enumeration of effects. Helps readers sense-check "is this everything / is something missing?". Else a citation if someone else has listed these before.*
*• Citations: If ~easy to do, worth putting in an overall reference / specific references to each effect. Your point 2. got me fishing out Schunk & Nagy, and checking block diagram of energy flow in Ch9!*

We hope that both the above points are addressed by the inclusion of the reference to Rishbeth and Clilverd 1999.

*Specific points:*
*• effect 1: demarcate better the loss rate change covered above, from the height change introduced here*
*@@ L46 @@*
*-thermosphere which increases the height*
*+thermosphere, as described above. This also increases the height*

Changed, as suggested.

*• effect 2: be bit more specific on solar irradiance process*
@@ L48 @@
-to the atmosphere, increasing
+to the upper atmosphere (thermosphere, ionosphere, mesosphere), increasing

Changed, as suggested.

@@ L49 @@
-of the atmosphere
+of the upper atmosphere, and hence raising of pressure levels

Changed, as suggested.

*• effect 3: Make the composition point clearer. This is where I was glad I'd drawn out diagrams for myself! Helped me see that all other processes were changing the height of layer features (e.g. hmF2), whereas this process seemed not to. As written, ~implies this is only directly changing the density (e.g. nmF2). I think this is your intent - rest of article showing that such changes to (say) F1 densities can contaminate hmF2 reconstructions. I've suggested an extra sentence at end of intro to discuss this point, and tease out this*

*difference. But possibly worth flagging this up already here? E.g. via tweak below? Of course, if the composition changes also directly affects true feature height (not just density), then alter tweak! In that case, recommend being as explicit as you can.*

*@@ L49 @@*
*-greater altitudes*
*+greater altitudes. Profile changes mainly affect densities,*
*+but this can indirectly affect height estimates.*

The point we were trying to make here is that if you erode the lower edge of the electron concentration profile through changes to loss rate, the lower edge of the peak can be eroded too, resulting in  a new -weaker- peak at a different altitude.

We have added the following text to clarify this point;

'Changes to the shape of the ionisation profile can alter the altitude at which the peak electron concentration is established.'

*• effect 5: clarify that this effect acts exclusively on long-term timescales?*

We have added the sentence
'This latter effect is predicted to alter the height of ionospheric layers over a multi-decadal timescale.'

*• L64: might need rewording for clarity 4 and 5? Think only 5 is discussing contraction. And isn't your point in following sentence and section that most authors considered residual to be 5 alone? Whereas only a few - Jarvis, Bremer, Mikhailov - have pointed out that at least 4 can affect long-term trend too?*
*If so, something like following?*
*@@ L63 @@*
*-due to the contraction of the thermosphere assumed to be the dominant part of (4) and (5).*
*+widely assumed to be dominated by the contraction of the thermosphere (5).*

Changed, as suggested

***Composition and circulation, effect 3** Recommend you also consider adding/moving following to this section:*
*• moving your composition+circulation discussion from ~L431 - L439 to this section*
*• appending a brief discussion about effect 3*

Rather than moving a discussion about an explicit station to this general section, we have added the following text to effect 3;

'Such compositional changes can result from thermospheric circulation both on a seasonal timescale and as a result of space-weather related events increasing the molecular fraction of the upper thermosphere which is then transported equatorward via thermospheric circulation.'

*Composition+circulation discussion: I think you could usefully move this to introduction, say just before the effects discussion Highly relevant, and usefully bulks out wind pattern change*

*point, as well as better motivating the seasonal timescale. Currently the intro doesn't give much indication about why results consider seasonal timescale carefully - current timescales are really only ~ diurnal, space weather event, climate. This would help fill out spectrum! Having this here would also let you refer back to this in discussion of Scott et al ~L240.*

This is a useful observation, thank you. Rather than moving the discussion about two explicit locations to this general section however, we suggest adding the following text to the introduction to introduce the concept of seasonal change;

'Seasonal changes in thermospheric circulation coupled with seasonal changes in geomagnetic activity produce seasonal variability in thermospheric composition that varies with location.'

*Brief discussion on 3: I think worth adding something to end of intro:*

*• while it's very easy to juxtapose with the other effects*
*• to re/plant seed in readers' minds*
*• to help introduce following section*
*• to explicitly tie this in with wider thrust of paper*
*Something along lines of following - playing around with true/measured as appropriate!*
*"Absent from most previous literature has been consideration of the ionisation profile changes in (3). This effect may seem irrelevant, as it mainly affects the density of given features in the profile, rather than their true height. However, such density changes can have an indirect effect on estimates of this true height using ionosonde measurements, as discussed in the following section. And hence, as we show in this paper, can contaminate efforts to reconstruct true height estimates to extract the target climate signal from (5)."*

Thanks for this suggestion. Further investigation suggested by the referee has revealed that the observed bias in hmF2 can be attributed to foF2/foE and foF2/foF1, and that these biases cannot easily be deconvolved. However, both are signatures of compositional change. We have opted to generalise the suggested paragraph slightly to the following;

'Absent from most previous literature has been consideration of the ionisation profile changes in (3). This effect may seem irrelevant, as it mainly affects the electron concentration of given features in the profile, rather than their true height. However, such profile changes can have an indirect effect on estimates of this true height using ionosonde measurements. As discussed in the following section, this can potentially introduce bias into reconstructions of true height estimates needed to extract the target climate signal from (5).'

**Measuring the ionosphere using ground-based radar**
***Scaling** You briefly cover inverting (L84), but don't really cover scaling, despite referring to this later (L164). "Scaling" is quite jargon-y, so may not be accessible to all readers.*
*Once you got to B&D discussion of foF1, I had to do a quick refresher on ionosondes. I ended up needing to go quite deep into INAG links to understand what the issue was.*
*As you're (necessarily!) going into the weeds in this paper, possibly worth:*
*• adding a brief description of scaling process*
*• adding some citations to guides to both scaling and inverting ionosondes?*
*I ended up looking at the below for scaling. Might usefully supplement Rishbeth & Garriott (which I didn't check)?*

*Flagging resources like this helpful:*
*• for any readers wanting to do a deeper dive here*
*• esp to any you motivate to run with data-rescue inversion efforts!*
*~Old guides for scaling linked off https://www.sws.bom.gov.au/IPSHosted/INAG/ :*
*• Wakai, Ohyama and Koizumi, 1987, Manual of Ionogram Scaling, Radio Research Laboratory, Japan, 3ʳᵈ edition,*
*https://www.sws.bom.gov.au/IPSHosted/INAG/scaling/japanese_manual_v3.pdf*
*• Piggott & Rawer (eds.), 1972, U.R.S.I. handbook of ionogram interpretation and reduction, 2nd edition, UAG-23A, World Data Center A,*
*https://www.sws.bom.gov.au/IPSHosted/INAG/uag_23a/UAG_23A_indexed.pdf (note high latitude supplement linked from top link too)*

Thanks for this. Yes, we agree that 'scaling' is jargon and so have added the following text to the section describing the creation of ionograms;

"Tabulating the entire ionospheric height profile was not routinely carried out in the early days of routine ionospheric monitoring. Instead, international standards were established for the identification and recording of key features on each ionogram, such as the peak frequencies of each layer and their virtual heights (Piggot & Rawer, 1978). This task is referred to as ionogram 'scaling' or 'reduction' and was usually carried out by a skilled individual or small team from each ionospheric sounding station to ensure consistency of the data."

We have cited Piggott and Rawer as their work was represents the early consensus reached through the international organisation URSI.

**Using ionosonde data to estimate hmF2**
***Motivation*** *I think you could usefully make it even more explicit here what the motivation for using ionospheric measurements is, and hence why so many authors have tried. A few suggestions:*

*@@ L97 @@*
*-effects are altitude dependent which will lead to differing results. Searching*
*+effects in the upper atmosphere are altitude-dependent, which will lead to*
*+unhelpfully differing results, depending on the altitudes of the specific*
*+observations in each study.*
*+By contrast, searching*

Text changed, as suggested.

*@@ L99 @@*
*-The longevity of ionospheric data series*
*+Furthermore, of the upper atmosphere regions, the ionosphere is unique as it can*
*+be observed remotely with relative ease. Due to the ionosphere's importance for*
*+long-distance radio communication, such observations have been made routinely*
*+since the early 20th century. The resulting longevity of ionospheric data series*

Text changed, as suggested.

*@@ L101 @@*

*-can be extracted from the data, allowing for all other effects.*
*+can be extracted from the data, with all other potentially-confounding effects compensated for.*

Text changed, as suggested

*@@ L102 @@*
*-The first published analysis*
*+This potential has motivated similar re-analysis of ionospheric data,*
*+seeking evidence of climate-driven trends.*
*+The first published analysis*
*Be more explicit on what relevance is:*

Text changed, as suggested

*@@ L115 @@*
*-Kokubunji in Japan with*
*+Kokubunji in Japan (the same station examined here) with*

Text changed, as suggested.

***Ap and am*** *L131: Ap index vs am index: Somewhere appropriate (here / a very short new section 2.3 / section 3.4) it would be good to have a brief discussion of this, making it explicit why you use am, rather than the Ap you flag has been used by most previous studies have used Ap! Took me down a bit of a rabbit hole refreshing myself on am - had been a while! I think I eventually got my answer from Mienville & Berthelier, 1991, The K-derived planetary indices: Description and availability, https://doi.org/10.1029/91RG00994:*
*• end of their section 5 basically says that better sensitivity and distribution of Km network (cf Kp) makes am more suitable for statistical studies.*
*If this sort of reasoning governed your collective choice to use am, could you add a brief explicit sentence to this effect, citing Mienville & Berthelier.*
*• Or Lockwood+ 2019 which you cite already - looks like similar reasoning in there*
*Say this as if "accurate and painstaking" is required, favouring am over Ap seems very aligned: worth making your reasoning explicit, so others can follow suit!*

Again, a very useful point. We have added the following text in section 3.4 to justify our use of the am index when comparing with the bias in the modelled hmF2 values;

"We here choose to use the am index rather than the Ap index used in previous studies. The response patterns of the individual magnetic observatories used to compile such indexes depend strongly on the level of geomagnetic activity. At low activity levels the effect of solar zenith angle on ionospheric conductivity dominates over the effect of station proximity to the midnight-sector auroral oval, whereas the converse applies at high activity levels. It has been shown (Lockwood et al, 2019) that these biases are far smaller for the am index than for Ap"

***Confounding factors*** *Make it more explicit that the tempting elision that "remaining trend == climate change" is dangerous? And loop back to your different effects from introduction? E.g.*
*@@ L134 @@*
*-Mikhailov and Marin*

*+When analysing such trends in residuals however, it is important not to assume*
*+any local environment changes which may underly this can be attributed to*
*+greenhouse forcing alone.*
*+The previously-discussed effects altering thermospheric composition (3)*
*+and wind patterns (4) risk being confounding factors, as they can potentially*
*+also lead to trends on long timescales.*
*+Mikhailov and Marin*

Text changes, as suggested.

***Clarify scope*** *L160: worth being blunter, as think you've been nicely impartial in the actual analysis?*
*• if you adopt my "in section X" suggestion below, you could leave this as-is, and have summary outcome in there*
*@@ L160 @@*
*-to investigate the efficacy of*
*+is to demonstrate the potential pitfalls in*

Text changed as suggested

*L161: current sentence ~implies the present work is looking into this reconciliation globally. Section 3.4 is a great start, and you've ~convinced me that global trends need to be reinterpreted in light of this paper, to see if reconciliation possible. But given this paper only examines Kokubunji, Chilton, and Stanley, this sentence may need watering down a little. E.g.*
*@@ L161 @@*
*-investigate the extent to which this can reconcile the difference*
*+demonstrate that such effects may have potential for reconciling the differences*

Text changed, as suggested.

***Add an outline*** *L162: you lost me a little in section 2, trying to keep key bits in my working memory, and interpreting the detail you're exposing in light of this, while we're (necessarily!) knee-deep in complex empirical equations!*
*I think you could usefully insert a brief outline here at L62, summarising what you'll do & key findings. A la "In Section 2, we do XYZ and show, In Section 3 …".*
*Would let readers see where you're going, so help follow the thread of your argument better once in there - and flick back to this if they lose thread!*

Thanks for this suggestion. We have added the following text;

"Section 2, contains a summary of various methods used to derive $hmF2$ estimates from routinely scaled ionospheric parameters, and the assumptions made in doing so. Thereafter, section 3 will examine the accuracy of such estimates through comparison with ionospheric heights measured by Incoherent Scatter Radar."

***Estimating hmF2 from empirical formulae***
***Existing parameters*** *To make point clearer to those less familiar with ionosondes, something like following:*
*@@ L167 @@*
*-from existing scaled ionospheric parameters*

+*from existing standard ionospheric parameters (e.g. foF2, MUF).*
+*Determining these from an ionogram only required a scaling process (not inversion),*
+*hence these were routinely calculated from ionograms at the time of measurement.*

Text modified, as suggested.

***Minor fluctuations*** *L204: I'm not following. Can you try to interpret what B&D / their Paul ref means? Is the point that:*
*• these are true F1 layers, but hard to pin down, as noisy?*
*• this is effectively noise (ionospheric variability / instrumental) getting mislabelled as F1?*

I think their point was that, while an additional layer had been identified, it did not necessarily represent a major change in electron concentration.

We have addressed this point by adding the following text
(indicating that the presence of an F1 layer did not represent a significant increase in electron concentration)

*Good if you could also tweak "more ubiquitous" on L207 to whatever is more appropriate:*
*• more ubiquituously-recorded*
*• less contentious*

We have changed the text to read 'ubiquitously-recorded' as suggested.

***Shape of profile*** *L213: I don't follow - can you clarify what this "well above the limit" profile point means? I assume this supports B&D's argument that their approach means any F1 ionisation present can be ignored. But I'm unclear why. Especially if they were doing this off synthetic ionograms. Is this:*
*• a "how far are F2 and E separated along x-axis" measurement error type argument?(!)*
*• a physics-based argument on the retardation between E and F2 exceeding retardation from F1 features above this cutoff foF2/foE ratio?*
*• other?*

Thanks for pointing this out. We have corrected the reference to foF2/foF2 ~ 1.2 (this came from Dudeney, 1974) and added text to clarify what was meant by this and how comparison with real data by Bradley and Dudeney led them to the conclusion that the presence of an F1 layer had minimal impact on their estimates of hmF2.

"…sufficiently close to the layer critical frequency to make a significant contribution to the total group delay of the radio pulse. However, Bradley and Dudeney (1973) compared estimates of hmF2 with heights determined from ionogram inversion for a number of different locations, over all seasons and for solar cycle extremes. These results supported their conclusion that the presence of an F1 ledge had minimal effect on hmF2 estimates calculated using their empirical formula."

***Limitations*** *Can you give a physical insight? E.g.*
*@@ L217 @@*
*-$X_E < 1.7$, which*
*+$X_E < 1.7$, due to a more ionised E layer, which*

*Possibly worth adding foF1? Belabours point from previous sentence, but think OK, as it's a key finding of this*
*paper that this is wrong!*

*@@ L219 @@*
*-field.*
*+field. And by definition this assumes F1-related effects are negligible.*

Thanks for this. We have changed this paragraph to read;

While powerful, this method cannot be used when $X_E < 1.7$, which frequently occurs during the daytime summer at mid to high latitudes \citep{dudeney1974} Here the E-layer is strongest due to increased ion production while the F2-layer is often weakened through an enhanced loss rate resulting from a higher fraction of molecular species in the upper thermosphere. In addition, this formula does not account for the effects of the Earth's magnetic field.

***Which formula?** L246: "refinements of the formula":*
*• a specific one? If yes, can you specify which one*
*• if general class, can you be clearer, e.g. "refinements of similar formulae"*

Thank you for pointing out this ambiguity, We have changed this to read 'refinements of similar formulae'

***What is impact of foE = 0.4 MHz assumption?** L279: can you indicate what expected impact of this fixed foE assumption is on hmF2? (realise elsewhere you say noise in results - I'm less sure! Here's why)*
*I assume this acts as a floor - the true foE would likely be lower, but this can't be measured, so it gets floored (ceilinged really!) to 0.4 MHz?*
*If so, what is this going to do to the "true" hmF2?*
*From a quick play with B&D eqn in Excel, and guessing some values:*
*• foE = 0.001 - 0.4*
*• foF2 = 5 - 7*
*• M(3000)F2 = 3 I think this floor will impose a very small < 5 km shift downwards on the value of hmF2 which would be calculated with the "true" foE. And this would affect night time values, so the ~300 km mark. So should have a negligible effect on your plots & analysis? But would be good to have you check this, and to convey any such message.*

The assumption that foE = 0.4 MHz at night appears to generate an underestimate of the derived hmF2 value. In addition, assuming that this value is a constant (when, at the very least it will have a diurnal component) introduced scatter around this underestimate. Residual nighttime E-region ionisation results from cosmic ray and astronomical x-ray sources. Adding to this is the fact that a typical ionosonde is insensitive to frequencies below ~ 1MHz. For a low night-time foF2 value, over-estimating foE will lead to an underestimate of the foF2/foE ratio and an underestimate of hmF2.

The effects of this are now apparent in the revised figure 2 where the populations are plotted separately, as advised. That the night-time estimates result in a lower gradient than the twilight population indicates that the assumed value of 0.4 MHz is an over estimate at night, being more applicable to (though still an over-estimate of) the E-region peak concentration at twilight (at least for this location).

We have changed discussion of figure 2 to read;

"From equation ??, over-estimating foE will lead to an underestimate of the foF2/foE ratio and an underestimate of hmF2. In addition, assuming that the value of foE is a constant (nighttime E-region ionisation results from cosmic ray and astronomical x-ray sources that will vary throughout the night) likely introduces scatter around this underestimate. Added to this is the fact that a typical ionosonde is insensitive to frequencies below ~1MHz.

With reference to figure 2, that the night-time estimates result in a lower gradient than the twilight population indicates that the assumed value of 0.4 MHz is an over estimate at night, being more applicable to (though still an over-estimate of) the E-region peak concentration at twilight (at least for this location)."

*The Kokobunji ionosonde data*
*Clarify binning, ionosonde data retrieved* *L282: worth defining & justifying binning better. And enumerating all data retrieved.*
*Until I looked at code / thought hard about plots, I thought your monthly medians collapsed local time information. Also, it's only later that justification for medians appears. And that you also download F1 values. How about something like following? Longer, but covers all data, and makes why & how for averaging more explicit up front.*

*"To estimate hmF2, scaled critical frequency parameters for the E, F1, and F2 layers (foE, foF1, and foF2), as well as the M(3000)F2 factor were downloaded. Monthly averages were then calculated for these data, to protect against outliers caused by short-lived space weather events that are not representative of the data on monthly timescales. Specifically, hourly monthly medians were used - medians calculated across corresponding hours (in local time) within a given month. For a given year, this yields 288 median values (bins of 12 months, and 24 local time hours). Such hourly monthly medians of foE, foF2 and M(3000)F2 were then used to estimate corresponding hourly values of hmF2, the true height of the F2 layer, using…"*

Thank you for this observation. We have added the text as suggested.

*The Middle and Upper Atmosphere (MU) Radar*
*Clarify averaging*
*@@ L318 @@*
*-height profiles were then averaged to*
*+height profiles were then averaged over the four antenna positions to*

Changed as suggested,

*@@ L327 @@*
*-data, monthly means were*
*+data, corresponding hourly monthly means were*

Changed as suggested.

*@@ L328 @@*
*-35 years of data used in this study, observations have*
*+35 years of ISR data used in this study, ISR observations have*

Changed as suggested.

*@@ L329 @@*
*-10080 bins (monthly averages for each hour over 35 years).*
*+10,080 bins (monthly means, in bins for each local time hour and month, over 35 years).*

Changed as suggested

*I've suggested moving justification below to ionosonde section*
*@@ L329 @@*
*-parameters, this is to protect against outliers in ionospheric*
*-concentrations caused by short-lived space weather events that*

*-are not representative of the monthly data. For the ISR data,*
*+parameters, for the ISR data,*

Changed as suggested

**Seasonal and diurnal comparison**
**Fig 1 interpretation** *L339: recommend flagging up in text a few extra things which jump out*
*at me from Fig 1:*
*• The ISR data are noisier.*
*– I assume because the campaign nature of ISR measurements means fewer underlying*
*datapoints are available to constrain each hourly monthly mean datapoint?*
*• At solar min, the ISR data hit a floor at 180 km due to your sporadic E contamination*
*procedure (suggest flag in figure too?)*

While the code rejects values that are at the extreme ends of the altitude profile window, it is true that this arbitrary floor may lead to an over-estimate of hmF2 at these times. We have added the following text to address the point above.

"From this comparison it can be seen that the ISR data are noisier, due to there being relatively fewer data points from which the mean values are calculated. Also at solar minimum years some values are close to the $180 km$ floor of the altitude window in which the F2 peaks were identified."

And added the following text to the figure caption;

"It can be seen that the ISR data are noisier, due to there being relatively fewer data points from which the mean values are calculated. Also at solar minimum years some values are close to the $180 km$ floor of the altitude window in which the F2 peaks were identified."

**Fig 3 interpretation**
*@@ L351 @@*
*-averaging over the 35 years*
*+averaging the aforementioned 10,080 bins over the year axis, for the 35 years*

Changed, as suggested.

*L352: you don't explain the white lines - I had to look at your code to double-check interpretation. Can you:*
*• add an explanation that these are the sza = 90, 100 limits used to isolate the previously-discussed day/terminator/night populations in Fig 2*

*• make these lines a bit thicker - the 90 one is particularly hard to see*
*• add lines like this to all subpanels, and all Figures like this*
*• consider a very brief discussion of results with respect to these?*

*Re latter, when I hand-draw similar lines on ISR, I see that for both ISR and ionosonde, dawn hmF2 hugs this sza line across all months. But that sza sn't constrain dusk hmF2, esp in winter months. I expect a ~easy Appleton anomaly explanation for diurnal, something else for seasonal. Would be good to have this flagged, and briefly explained by you!*

As suggested, we have thickened the lines slightly (though we did not want them to dominate the plot) and replicated over all such plots.

In addition we have added the following text;

"The dotted and solid white lines overlaid on these plots represents the times at which the solar zenith angle is 90$^{\circ}$ and 100$^{\circ}$ respectively. These values were used to differentiate between the 'daytime', 'twilight' and 'nighttime' populations. It is interesting to note that there is a clear change in hmF2 at the dawn boundary which is less apparent at dusk. This is likely due to variability in the F2 layer at dusk which is 'reset' by the decay of this layer overnight.

We have also added the following text to the figure 3 caption;

'The dotted and solid white lines overlaid on these plots represents the times at which the solar zenith angle is 90$^{\circ}$ and 100$^{\circ}$ respectively'

*@@ L353 @@*
*-peaks in hmF2 at*
*+peaks in night-time hmF2 at*

Changed as suggested

*Flag up midday summer peak*
*• you concentrate on this difference for ~rest of paper*
*• so good to flag up here that this is the target you'll concentrate on*
*@@ L354 @@*
*-a stronger peak*
*+an unexpected strong peak*

Changed as suggested.

*@@ L355 @@*
*-source of this bias*
*+source of this midday summer bias*

Changed as suggested

**Fig 4 interpretation** *L356: foF2/foF1: flag that this corresponds to xF from earlier eqn 4 L357: foF2/foF1: can you explicitly link back to discussion of eqn 4? Make point that B&D's range is nominally xF > 1.2 (note earlier query). But that here you're seeing it looks like it's more restrictive, that the 1.2 <= x.F <= 1.6 range is problematic?*

We have added the following text;

'(the parameter xF intruduced when discussing equation 4)'
and
'It was expected that equation 4 should be valid for values of foF2/foF1 above 1.2'.
Before going on to discuss the observed bias above 1.6.

***Correcting for foF1*** *L360: can you be explicit on how you've done this?*
*Ideally break this out into an equation, as there are a couple of times later where you could usefully refer back to this.*
*Good to have this made explicit - currently I've got questions/guesses below.*

*I assume making the assumption that for*
*• xF < 1.6, hmF2_iono_true ~= hmF2_ISR*
*• xF >= 1.6, hmF2_iono_true = hmF2_iono_raw?*
*And so you filter ionosonde data to select 1st branch (xF < 1.6).*

*And for those data you apply linear model hmF2_iono_true = (231\*xF) + hmF2_iono_raw - 356?*
*And you apply this to the binned hourly monthly median values (i.e. rather than to raw values, pre-binning)?*
*And for Fig 5, you do this after binning, but before collapsing over the year axis?*

Yes, this is what we have done. While we have opted not to create an explicit equation for this result, we have added the following text to make this clearer;

'For each bin in figure 3 where foF2/fof1 <=1.6, the liner fit was used to calculate the bias and these biases were used to correct the ionosonde-derived values of hmF2.'

***Long-term bias***
*L380: reword - this is unclear. My guess:*
*@@ L380 @@*
*-to correct affected daytime values within the hourly monthly median hmF2 values.*
*+to correct affected hmF2 values (where foF2/foF1 < 1.6)*

Thanks for pointing this out. This sentence was changed to read;

'The relationship between the bias in hmF2 and the foF2/foF1 ratio established above was used to correct affected hmF2 values (where foF2/foF1 <= 1.6) before averaging by year.'

*@@ L381 @@*
*-When plotted*
*+When daytime values are plotted*

***The impact of foF1 on the long-term drift in estimates of hmF2***
*L411: if you followed my suggestion re model equation, here you could say "values, yielding similar models to eqn X, but for each year".*

This is not what we have done. In the previous section referred to, we fitted the difference between the ISR and ionosonde derived values of hmF2 as a function of the foF2/foF1 ratio.

Here, we are taking annual subsets and determining the bias between the ionosonde-derived and ISR measurements of hmF2. In this way, we can investigate whether the uncorrected formula introduces any long-term bias into the estimates of hmF2.

*L413: add unit and make the interpretation clearer here!*
*@@ L413 @@*
*-of $\pm$10% ($\pm$25km at 250)*

+of up to $\pm 10\%$($\pm 25$km at 250 km, i.e. a little larger than the
+~20 km decrease expected from climate change)

Thanks for this. As we have been somewhat arbitrary in choosing a height of 250 km at which to determine any annual bias, rather than a qualitative comparison we have added the text;

'This is of the order of decrease expected from climate change'.

*@@ L420 @@*
*-composition.*
*+composition, arising in turn from long-term changes in geomagnetic activity.*

Changed as suggested.

*L440: can you make this point a little clearer?*
*I'm getting a bit lost on the temporal aspect - annual vs semi-annual. I think this is a red herring, not least as you're showing a clear difference in Fig 9 when you simply present annual averages!*
*I think the argument is ultimately altitudinal - what I'm (mis)understanding so far:*
*• near the poles, the neutral upwelling in summer disproportionately suppresses nmF2 (??? not sure why nmF1less affected?)*
*• which pulls the yearly average of foF2/foF2 down?*
*• whereas further from poles, there isn't a strong altitudinal variation by season.*

Whether a station exhibits an annual or a semi-annual variation in foF2 is down to whether the annual variation in composition at F2 altitudes modulates the loss process more than the seasonal change in solar zenith angle affects the ion production rate. If the former, the station exhibits an annual variation (with a minimum foF2 in the summer, where loss rates are enhanced) if the latter, the semi-annual variation dominates. If there are also long-term changes in composition, the annual variation in ionisation will vary over time as each effect waxes and wanes in dominance. Previous studies have shown that changes in the power ratio of the annual/semi-annual variation in foF2 follow geomagnetic activity, through modulation of thermospheric composition. We raise this point here as it corroborates our current result.

Composition change primarily affects the F2 region where the difference between the lifetime of atomic and molecular ions is most apparent, leading to much larger changes in the peak density at equilibrium between production and loss processes.

In order to explain why it is largely the F2 layer that is affected by compositional changes, we have added the text;

'In contrast, the influence of compositional change on the peak concentrations of the E and F1 layers is much smaller, since molecular ions exist in much greater proportions at these altitudes and loss rates are higher due to the comparatively high thermospheric densities.'

**Conclusions**
*L496: latitude only? Does longitude need considering too cf circulation. Or is this ~purely meridional? Ask as ISRs which Bilitza 1979 used all have similar MLON (ie MLT) values.*

Yes, this is a fair point! We have changed 'latitude' to 'location.

*Good if you can add missing URLs*
*• Bradley & Dudeney: https://doi.org/10.1016/0021-9169(73)90132-3*
*• Dudeney: https://nora.nerc.ac.uk/id/eprint/509197*
*• …*

We have added such links where we can

***Possible typos***
*@@ L22 @@*
*-peak density of the F2 layer, foF2, would*
*+peak density of the F2 layer, nmF2, would*

Changed as suggested.

*@@ L86 @@*
*-automatically identify and invert*
*+automatically scale and invert*

Changed as suggested

*L92: citepshangguan2019*
*L96: citepphilipona2018*

Citations corrected.

*L175: I think this should be inverted, no? M(3000)F2 = MUF/foF2 definition used by Elias 2017 (eqn 13), Japanese manual (p109), UAG (p23)*
*@@ L175 @@*
*-The ratio foF2/MUF*
*+The ratio MUF/foF2*

Thanks, changed!

*L282: concentration vs critical frequency Think this should be critical frequencies, no, given foE and foF2, and formula definitions? So Note "concentration" is used quite often, and on a few other occasions is applied to a frequency, rather than a density/concentration. Realise these are ~equivalent (per eqn on line 68-69), but this doesn't help clarity. Could you do a search for "concentration", and try to rationalise this?*

We have checked through the manuscript and altered where appropriate

*@@ L282 @@*
*-The peak electron concentrations of the E and F2 layers (foE and foF2)*
*+The critical frequencies of the E and F2 layers (foE and foF2)*

Changed as suggested.

*L296: fragment?*

*@@ L296 @@*
*-integrated over. In the ionosphere, While the*
*+integrated over. While the*

Changed to '…integrated over. In the ionosphere, while…'

*@@ L342 @@*
*-Daytime data (for which the solar zenith angle, sza > 100° )*
*+Daytime data (for which the solar zenith angle, sza < 90° )*

Corrected as suggested

*@@ L342 @@*
*-twilight data (90° ≤sza≥ 100°)*
*+twilight data (90° ≤ sza ≤ 100°)*

Changed, as suggested

*@@ L382 @@*
*-values (figure 7, the correction*
*+values (figure 7), the correction*

Changed as suggested

*@@ L383 @@*
*-the remaining offset is not 1:1.*
*+the remaining gradient is not 1:1.*

Changed as suggested

L388: ionosonde values being "generally still lower" and "underestimate of the F2 layer height". Err, are you sure?
• Lower in some bins - equinox night
• But not in others - midsummer night
• cf Fig 2: the two clumps I've flagged above/below trendline
16
• No idea about "general" (average over all months), and how much there's in it!
• Think this is where my requested difference subpanel could help!

Yes, we agree that the variability is not as simplistic as this. We have changed the text to read;

'the night-time ionosonde-derived hmF2 values still show more scatter when compared with those measured by the ISR..'

*L469-L471: Comparing the before/after gradient values, I think you're currently a factor 10x too large in your reported improvements! Hence following changes*

Corrected as suggested in an earlier comment.

*@@ L469 @@*
*-resulted in an improved relationship*
*+resulted in a slightly improved relationship*

Changed, as suggested.

*@@ L470 @@*
*-the revised gradient for daytime hmF2 increases by 0.3 to 0.89 $\pm$0.01*
*+the revised gradient for daytime hmF2 increases by 0.03 to 0.89 $\pm$0.01*

Changed as suggested

*@@ L471 @@*
*-the equivalent after correcting for the presence of foF1 increases by 0.3 to 0.92 $\pm$0.01*
*+the equivalent after correcting for the presence of foF1 also increases by 0.03 to 0.92 $\pm$0.01*

Changed as suggested

Fig 2 caption
-gradient of 0.71pm0.01
+gradient of 0.71$\pm$0.01

Changed as suggested

*Fig 4 caption*
*-(gradient 231 $\pm$21, offset $-$356 $\pm$31km*
*+(gradient 231 $\pm$21, offset $-$356 $\pm$31 km)*

Changed as suggested

*Fig 5 caption*
*-by the ISR (figure 5 lower panel-).*
*+by the ISR (figure 4, lower panel).*

Now that the figures all contain the ISR data, we have changed this to self-reference the same figure;

'by the ISR (top right panel)'

*Fig 6 caption*
*-(as identified in 4. It can*
*+(as identified in Figure 4). It can*

Revised figure captions no longer contain this text.

---

## Author Comment (AC3)

Response to reviewer Claudia Borries

*This manuscript written by C. Scott et al. addresses the limitations of using the height of the F2-layer maximum electron density (hmF2), which is a derived quantity from ionosonde data, for studying long-term changes in the ionosphere. This is an important topic because there have been already many publications using hmF2 for long-term studies and this work helps evaluating these results and using hmF2 more careful in future. From my point of view the manuscript is written excellently. It contains a very good overview of the state of the art at the beginning and the applied analysis and presentation of the results is adequate and well understandable. The authors detect a relation between the occurrence and strength of the F1 layer and the accuracy of the hmF2 layer (derived with one of the common approaches), which has not been described before. The results are discussed with respect to numerous related studies and the conclusions are logically derived from the results. The manuscript also contains some relevant results and discussions on the potential impact of climate change on the ionosphere. I evaluate the manuscript very good and I have some questions and remarks which may be considered before publication.*

*Questions and remarks:*
• Section 3.5 provides corrections for the time delay in the ISR data. Why has this correction not been applied earlier in the study?

Thank you for this comment. This is a valid point and one we debated before we presented the work. We decided to present to analysis without the ISR data range correction (and simply quote the impacts on the various correlations when applied) because accounting for this correction required modelling the height of the F2 layer using the International Reference Ionosphere. We wanted to demonstrate that the result was not influenced by some implicit assumption in the model. The differences between the two analyses (with and without range correction applied) are small and the basic conclusions we draw from the analysis are the same in both cases. The equivalent plots for figures 4 and 7 in the revised manuscript but using the range-corrected data are shown below.

[Figure]

[Figure]

For transparency we would be happy make the range-corrected plots available as supplementary information.

We have modified the conclusions of this section to read;

'The results showed similar biases, with the coefficients of the linear fit exhibiting slight changes (gradient 79.6, offset -196.5 for the foF2/foE correction and a gradient of 46.6 with offset of -74.1 for the foF|2/foF1 correction). While it is not appropriate to apply these corrections to the individual points in the 35 year time series (since these have not been rangecorrected), as these are linear fits, the small changes to the coefficients will result in similarly small changes to the corrected values. The underlying conclusions concerning the impact of the foF2/foE and foF2/foF1 ratios on the empirical hmF2 formula are unaffected. The results showed similar biases, with the coefficients of the linear fit exhibiting slight changes (gradient 79.6, offset -196.5 for foF2/foE correction and a gradient of 46.6 with an offset of -74.1 for the foF|2/foF1 correction). While it is not appropriate to apply these corrections to the individual points in the 35 year time series (since these have not been range-corrected), as these are linear fits, the small changes to the coefficients will result in similarly small changes to the corrected values. The underlying conclusions concerning the impact of the foF2/foE and foF2/foF1 ratios on the empirical hmF2 formula are unaffected.'

• *line 482: Geomagnetic activity correlates to Joule heating driven by solar wind. How do the authors evaluate the potential that changes in thermosphere due to greenhouse effects may affect the magnitude of geomagnetic activity?*

This is an interesting question but one that is rather tangential to the subject covered in the current paper which concerns the calibration of ionosonde-based estimates of F2 layer heights (via an empirical formula) with those measured directly by Incoherent Scatter. While there may indeed be a modulation of the joule heating efficiency if the composition of the thermosphere is changed, the resulting changes in ionospheric layer height would be expected to affect both measures of F2 layer height and so would not introduce any bias into the calibration.

• *The authors use the ratio of foF2 and F10.7 as a composition proxy and refer to Wright and Conkright (2001). Such a proxy sounds very favourable, but reading Wright and Conkright (2001) I cannot see a justification that the ration of foF2 and F10.7 is a proxy for thermosphere composition. Wright and Conkright (2001) worked with a sunrise extrapolation index SRCC, which is related indirectly to foF2. Wright and Conkright (2001) correlate the ratio log(SRCC/F10.7) with log(O/N2) and find a rather moderate correlation. The authors describe in their conclusion that they intended to provide a composition index, but the morphology of the proposed one differs from that of [O/N2]. If the ratio [foF2/10.7] is used by Scott et al. as a proxy for thermosphere composition, they need to provide better justification.*
Thank you for pointing this out. This was also queried by reviewer #2. We have added the following text to describe this in more detail;

"Similarly, changes in neutral composition affect the peak electron concentration of the F2 layer. At noon, where the F2-layer approaches a steady-state condition, production and loss are in equilibrium. If the loss rate of ionisation is enhanced by the presence of molecular ions, the peak electron concentration of the layer will be reduced. Since the electron concentration is proportional to the peak frequency squared, comparison can be made between measurements at similar dates but different times in the solar cycle by scaling these values by the ion production rate, q. A good proxy for q is the F10.7cm solar radio flux. Thus

$$N \propto \frac{f^2}{q} \equiv \frac{foF2^2}{F10.7}$$

Using noon values of foF2 to track qualitative changes in thermospheric composition was suggested by Rishbeth et al (1995) with Wright and Conkwright (2001) comparing the efficacy of this simple index (their FFD index, averaged over 5 hours around noon) with other more complex indices derived from the rate of change of the ionosphere at sunrise."

Minor issues

• line 75 "Data from a such …"

Corrected. Thank you.

• lines 92 and 96 error in citing

Corrected, thank you.

• line 243 "… function of The Earth …"

Corrected. Thank you

• line 296 "In the ionosphere, While the …"

Corrected, thank you.

• line 371 "… in to the …"

This text has been changed in response to reviewer #2.

• line 389 "… formulae tends introduces …"

This text has been changed in response to reviewer #2.

• line 413 "(\pm 25 km at 250)". Add "km altitude" after the 250

Changed as suggested.

• line 416: Using am index is very reasonable. However, since it is not yet very popular to use, I recommend adding some justification, why this is used instead of the more frequently used kp/ ap indices.

Thanks for this useful comment. In order to do this we have introduced the following text;

"We here choose to use the am index rather than the Ap index used in previous studies. The response patterns of the individual magnetic observatories used to compile such indexes depend strongly on the level of geomagnetic activity. At low activity levels the effect of solar zenith angle on ionospheric conductivity dominates over the effect of station proximity to the midnight-sector auroral oval, whereas the converse applies at high activity levels. It has been shown (Lockwood et al, 2019) that these biases are far smaller for the am index than for Ap"

• line 417: "… between the two, …" it is not immediately clear what are the two parameters that are correlated. Accordingly, it is not clear for the correlation values in lines 423 and 426, too.

Text expanded in all cases to explicitly state that the correlation is between this parameter and the model error.

• line 430: What is meant with "longitude sector near to the geomagnetic pole (\approx 48-50N)"? How can a longitude sector be close to the pole and why does it have latitude (north) coordinates?

Thank you for pointing out this ambiguity. We have expanded the text to read;

"… is a mid-latitude station in a geographic longitude sector near to the geomagnetic pole (at a geomagnetic latitude during this epoch of ~48-50 N).

- line 438: I get confused with the description. Chilton does not have a semiannual variation in foF2?

Thank you for pointing out this ambiguity in the text, We have restructured this paragraph to read;

"In contrast, Stanley in the Falkland Islands (at a geomagnetic latitude of ~35- 39 S during this epoch) is a station that is far enough from the magnetic pole that compositional changes between equinox and winter months are relatively small compared with the associated change in solar zenith angle, resulting in a semiannual variation in foF2 (Millward et al,1996)."

- line 438-439 "such as is seen": Where is it seen? Is there a figure or paper?

This point has also been addressed in the above restructuring

- line 437: What means far enough away? Does it just need to be outside the auroral oval or even further away?

The distance varies depending on geomagnetic activity levels. We have added the following text to support this qualitative statement.

"The relative magnitudes of the annual and semi-annual variations at a given station vary depending on geomagnetic activity, resulting in the long-term trends identified by Scott, Stamper and Rishbeth, (2014)."

- line 440 "at these stations": here the two station are addressed and in the second part of the sentence only Chilton. This is confusing.
Thank you for pointing this out. We have split the discussion of Chilton into a separate sentence;

"Such differences are also likely to influence the relative values of the foF2/foE and foF2/foF1 ratios at these stations. For example, the ratios at Slough/Chilton will be lower during the summer when compositional change suppresses foF2 while foE and foF1 are at their peak."

---

## Author Response (AR2)

Response to reviewers 'Calibrating Estimates of Ionospheric long-term change' by C. J. Scott et al.

We thank the reviewers for their consideration of the revised manuscript and note that the manuscript has now been accepted without further changes requested.